# Effects of climate anomalies on warm-season low flows in Switzerland

Marius G. Floriancic[1,2], Wouter R. Berghuijs[2], Tobias Jonas[3], James W. Kirchner[2,4], Peter Molnar[1]

[1] Institute of Environmental Engineering, ETH Zurich, 8093 Zürich, Switzerland

[2] Department of Environmental Systems Science, ETH Zurich, 8092 Zürich, Switzerland

[3] WSL Institute for Snow and Avalanche Research SLF, 7260 Davos Dorf, Switzerland

[4] Swiss Federal Research Institute WSL, 8903 Birmensdorf, Switzerland

Corresponding author: Marius G. Floriancic (floriancic@ifu.baug.ethz.ch)

**Keywords:** low flow, hydrological drought, precipitation, evapotranspiration

**Short summary**: Low river flows affect societies and ecosystems. Here we study how precipitation and potential evapotranspiration shape annual warm-season low flows across a network of 380 Swiss catchments. Low flows in these rivers
typically result from below-average precipitation and above-average potential evapotranspiration. The lowest low flows result from long periods of the combined effects of both drivers.

**Abstract.** Switzerland has faced extended periods of low river flows in recent years (2003, 2011, 2015, and 2018), with major economic and environmental consequences. Understanding the origins of events like these is important for water resources
management. In this work we provide data illustrating the individual and joint contributions of precipitation and evapotranspiration to low flows in both typical and dry years. To quantify how weather drives low flows, we explore how deviations from mean seasonal climate conditions (i.e., climate anomalies) of precipitation and potential evapotranspiration correlate with the occurrence and magnitude of annual 7-day lowest flows ($Q_{min}$) during the warm season (May through November) across 380 Swiss catchments from 2000 through 2018. Most warm-season low flows followed periods of
below-average precipitation and above-average potential evapotranspiration, and the lowest low flows resulted from both of these drivers acting together. Low-flow timing was spatially variable across Switzerland in all years, including the driest (2003, 2011, 2015, and 2018). Low flows in these driest years were associated with much longer-lasting climate anomalies than the ≤2-month anomalies which preceded typical warm-season low flows in other years. We found that snow water equivalent and winter precipitation totals only slightly influenced the magnitude and timing of warm-season low flows in low-elevation
catchments across Switzerland. Our results provide insight into how precipitation and potential evapotranspiration jointly shape warm-season low flows across Switzerland, and potentially aid in assessing low-flow risks in similar mountain regions using seasonal weather forecasts.

## 1. Introduction

In recent decades, Europe has experienced several severe droughts (Van Lanen et al., 2016). Their impacts, such as dry river reaches and high water temperatures, have a range of adverse effects on society and river ecology (e.g., Poff et al., 1997; Bradford & Heinonen, 2008; Price et al., 2011; Rolls et al., 2012; van Vliet et al., 2012). Severe low flows in the years 2003, 2011, 2015 and 2018 led to substantial economic losses by limiting water availability for households, industry, irrigation and hydropower, as well as impacting river transportation (Stahl et al., 2016; Munich Re, 2019). Such effects are expected to become more severe and frequent as water demand rises, and as droughts are anticipated to increase in frequency and intensity in the future (e.g., De Stefano et al., 2012; Wada et al., 2013), leading to calls for improved understanding and management of droughts and their effects on low flows across Europe (e.g., Seneviratne et al., 2012a; Van Lanen et al., 2016; WMO, 2008).

In temperate climates, annual low flows typically occur in two distinct seasons, i.e. during late summer and autumn in warmer regions and during winter in colder regions (Fiala et al., 2010; Smakhtin, 2001). This typical low-flow seasonality has been reported for many regions of the world, including, for example, Austria (Laaha & Blöschl, 2006; Van Loon & Laaha, 2015), the Rhine river basin (Demirel et al., 2013; Tongal et al., 2013), and North America (Cooper et al., 2018; Dierauer et al., 2018; Wang, 2019). Switzerland also has two low-flow seasons, where the distinction between warm-season low flows and winter low flows is strongly connected to elevation (Wehren et al., 2010; Weingartner & Aschwanden, 1992). Low flows tend to occur in late summer and early autumn (August through October) in low-elevation Swiss catchments, and during the winter (January through March) in high-elevation catchments.

Catchment properties shape low flows by controlling the storage and release of water (e.g., Stoelzle et al., 2014; Van Lanen et al., 2013; Van Loon & Laaha, 2015; Staudinger et al., 2017), but the landscape itself does not cause low flows. Instead, the drivers of low flows are meteorological conditions that dry out catchments (e.g., Fleig et al., 2006; Haslinger et al., 2014 ; Smakhtin, 2001). Warm-season low flows are typically caused by sustained periods of high evapotranspiration and low precipitation, whereas winter low flows often follow sustained periods of sub-freezing temperatures (e.g.; Laaha et al., 2013; Van Loon, 2015). The duration of these anomalous weather conditions is critical in shaping the annual lowest flows. Their timing varies between years and is largely driven by climate seasonality. In this paper we refer to weather conditions that deviate from the seasonal norm as 'climate anomalies' regardless of the magnitude of this departure.

The two main climatic factors controlling water storage and release in a catchment are precipitation and temperature (through its influence on snow processes and evapotranspiration). Therefore precipitation (P) and potential evapotranspiration ($E_p$) anomalies are expected to be important drivers of warm-season low flows across Switzerland. Precipitation controls the amount of water that is available for runoff in a catchment, and sustained periods with little precipitation will inevitably reduce storage and thereby limit streamflow. Because there is a time lag between low precipitation and low streamflow, meteorological droughts (i.e., precipitation deficits) result in hydrological droughts and/or low flows if they persist for long enough (e.g., Peters et al., 2006; Tallaksen & Van Lanen, 2004; Van Loon, 2015). In Switzerland, there is limited precipitation seasonality, but precipitation can still vary substantially within seasons or from year to year. However, precipitation is expected to become increasingly seasonal with changing climatic conditions in the future, with less precipitation during summer and more precipitation in winter. In addition, anticipated changes in snowfall and snow packs may also alter river flows (CH2018, 2018).

High temperatures can be an indicator of high $E_p$, and thus high potential for depletion of soil moisture storage, reducing aquifer recharge and streamflow (e.g., Jaeger & Seneviratne, 2011; Vidal et al., 2010). Temperature extremes can be amplified

when low soil moisture limits evapotranspiration, leading to lower relative humidity and higher air temperatures, which further increase $E_p$ (Granger, 1989). Furthermore, vegetation decreases the amount of water available for streamflow by increasing transpiration during periods of high vapor pressure deficits. Although these mechanisms are known, the effects of evapotranspiration on river low flows have received relatively little attention compared to precipitation effects. Seneviratne et al. (2012) reported that low flows across Switzerland in 2003 more likely resulted from excess evapotranspiration than from spring precipitation deficits, and Teuling et al. (2013) have documented the depletion of soil water storage by high evapotranspiration during past European low flows. Woodhouse et al. (2016) reported that temperatures rather than precipitation explained the interannual streamflow variations of the Colorado river. More recently Cooper et al. (2018) reported that summer low flows in the maritime Western US are largely driven by summer $E_p$, rather than by winter precipitation or snow water equivalent. Mastrotheodoros et al. (2020) modeled how increasing evapotranspiration strongly reduced streamflow across the European Alps during the summer of 2003. Future $E_p$ is projected to increase along with increases in incoming longwave radiation (Roderick et al., 2014), with uncertain consequences for future low flows. In Switzerland, temperatures are expected to rise even quicker than the global average in the next decades (CH2018, 2018), potentially influencing low-flow dynamics.

Future climate changes will also affect low flows in mountain regions by altering snow accumulation and melt. Multiple studies have examined how winter precipitation and snow water equivalent affect summer low flows in high-elevation catchments. For example, Godsey et al. (2014) found that shrinking snowpacks in the Sierra Nevada of California led to smaller low flows in the following summers. Jenicek et al. (2016) reported that maximum snow accumulation strongly affected summer low flows across several Swiss mountainous catchments. Dierauer et al. (2018) found that warmer winters with less snow accumulation led to lower summer low flows in mountainous catchments of the Western United States. Recently, Wang (2019) reported that climate warming might increase aquifer conductivity and thereby streamflow in cold region watersheds. Future climate warming in both warm and cold seasons will most likely impact summer low flows through different mechanisms. In summer, higher temperatures increase potential evapotranspiration, whereas in winter they reduce snowpacks (e.g., Déry et al., 2009; Diffenbaugh et al., 2015; Musselman et al., 2017).

The effects of precipitation, temperature and evapotranspiration on low flows have been investigated for individual events or individual catchments and regions in the literature. Previous studies have largely focused on how signatures of low flows (averaged across many events) relate to catchment and climate characteristics (e.g., Fangmann & Haberlandt, 2019; Hannaford, 2015; Laaha & Blöschl, 2006; Van Loon & Laaha, 2015). To our knowledge, however, no studies have systematically assessed the direct impact of temperature and precipitation during periods immediately preceding individual annual low-flow events across many catchments in a topographically diverse region.

Here we explore how precipitation and $E_p$ deviations from their seasonal norms (here termed "climate anomalies") jointly shape the occurrence and magnitude of annual warm-season low flows across a network of 380 Swiss catchments. Because the annual lowest flow is an atypical flow condition, we expect it to follow atypical weather conditions, rather than reflecting climate seasonality alone. Therefore, we hypothesize that annual lowest flows will typically occur after anomalous weather conditions, that is, weather conditions that deviate from the seasonal average. Understanding how anomalous weather drives low flows may help to reveal the processes at work, and also support low-flow forecasting. Switzerland is an interesting study region because gauging and climate data are available from a dense station network spanning a wide range of elevations, climates, and topographies. We investigate (a) how precipitation and $E_p$ anomalies separately and jointly shape the occurrence

and magnitude of warm-season low flows across Switzerland, (b) which durations of these anomalies have the strongest impact on low-flow occurrence and magnitude, both in typical and in exceptionally dry years, and (c) how winter precipitation and snow packs influence the magnitude and timing of warm-season low flows. Understanding these connections is important for anticipating how streamflows are likely to respond as the exceptionally dry years of today are expected to become more typical in a future warmer climate.

## 2. Data and methodology

### 2.1. Streamflow and climate data

We compiled daily streamflows for 380 gauging stations across Switzerland for a 19-year period (2000-2018), using data collected by the Swiss Federal Office of the Environment (FOEN) and the Swiss Cantonal authorities. This data set excludes catchments with obvious anthropogenic influences on the hydrograph, e.g., from major dams or hydropeaking operations. Low flows were defined as the lowest 7-day average streamflow for each year ($Q_{min}$). We calculated the magnitude and timing of $Q_{min}$ in each catchment for each year from 2000 to 2018. Not all catchments had continuous data for all 19 years; in total we could calculate low-flow magnitude and timing for 6237 station-years. This data set included years when the lowest annual flows were much higher than typical low flows (e.g., in especially wet years and years without distinct dry periods). We removed all annual low flows above the threshold of 2.5 mm d$^{-1}$, which is the 25$^{th}$ percentile of daily discharges across all catchments, because flows above this threshold cannot be considered truly low flows. This resulted in the removal of approximately 2% of all low flows, leaving a total of 6124 station-years for our analysis. We split the dataset of annual low flows into cold-season low flows occurring between December and April and warm-season low flows occurring between May and November. In total, we observed 2122 cold-season low flows and 4002 warm-season low flows across the 380 catchments within the 19-year time period.

We determined catchment area and mean catchment elevation for each gauging station based on a 2-m DEM (SwissAlti3D 2016, Swisstopo), using functions provided in the ArcGIS "Spatial Analyst" toolbox. The catchments range in size from 1 to 519 km$^2$, vary in mean elevation from 309 to 2930 m, and have diverse landcovers and climates. Daily gridded precipitation and temperature data (~2x2 km cells; Meteoswiss products "RhiresD" and "TabsD") were used to derive catchment-averaged weather and climate conditions. Daily potential evapotranspiration ($E_p$) was estimated following the method of Hargreaves & Samani (1985). A gridded dataset of snow water equivalent (SWE) on March 1$^{st}$ of each year was used to estimate catchment-average SWE. The SWE product was based on data from 320 Swiss snow monitoring stations that were assimilated into a distributed snow cover model (Magnusson et al., 2014; Griessinger et al., 2016). We use SWE on March 1$^{st}$ instead of April 1$^{st}$ because our focus is on warm-season low flows in lower-elevation catchments, most of which have no substantial snow left by April 1$^{st}$ (Steger et al., 2013; Winstral et al., 2019; Lüthi et al., 2019).

### 2.2. Anomalies of climate variables

To infer climate conditions preceding annual low flows, we selected the annual 7-day minimum streamflow events ($Q_{min}$) in each catchment for each year from 2000 to 2018. We then calculated precipitation and potential evapotranspiration for time windows of different lengths prior to each annual low flow. We hypothesize that severe low flows will usually follow periods in which precipitation and potential evapotranspiration significantly deviate from their seasonal averages. Thus, we define climate anomalies as deviations of precipitation and potential evapotranspiration from their climatic norms, defined as their long-term averages on the same day of the year. For example, we quantify precipitation anomalies (in mm) by:

$$\sum_{t=d_l-d_t}^{d_l} (P(t) - \bar{P}(t)) \qquad\qquad Eq.(1)$$

where $P(t)$ is daily precipitation (mm) at day $t$, $\bar{P}(t)$ is the climatic mean precipitation on day $t$ averaged across all of the years of record, $d_t$ is the time period over which anomalies are calculated for each annual low flow, and $d_l$ is the day of the low flow. We vary the time period $d_t$ from one week to half a year (7, 14, 30, 60, 90, 120, 182 days), with the endpoint always being the date of the low flow. For example, the 30-day precipitation anomaly for a low flow that happened on 30 September 2018 is

calculated using the sum of precipitation from 1st to 30th September 2018 minus the mean of precipitation for all 1st to 30th September periods from 2000 to 2018. We calculate $E_p$ anomalies in the same way.

### 2.3. Statistical tests and quantification of process importance

We first report the spatial distribution of the timing of the annual lowest flows across Switzerland for 2000 until 2018. We

then show the magnitude of 30-day climate anomalies before each annual low flow as a function of elevation. The mechanisms involved in generating annual cold-season low flows and warm-season low flows are different, thus we split our dataset into cold-season and warm-season low flows. From this point on, we report results only for warm-season low flows. To quantify the relationship between the magnitudes of climate anomalies and the magnitudes of warm-season annual low flows, we use Spearman rank correlation coefficients ($r_S$) as a robust estimator (Legates & McCabe, 1999). We report these rank correlations

across all catchments in a histogram. To test the regional significance of the $r_S$ coefficients we use the sign test.

We assess the impact of the length of climate anomalies preceding the annual warm-season low flows by comparing the magnitude of P and $E_p$ anomalies for the different time windows (7, 14, 30, 60, 90, 120, 182 days) between the four driest years and the more typical years. We also correlate the magnitude of warm-season $Q_{min}$ with the number of days that P and $E_p$

exceed certain thresholds. The threshold that defines low precipitation is the 20th percentile of 10-day running averages of precipitation over the entire period of record. Similarly, the threshold that defines high $E_p$ is the 80th percentile of 10-day running averages of $E_p$ over the entire period of record. We report the distribution of rank correlations calculated for each catchment based on the 19 years of data in histograms. The magnitudes of the annual low flows are shown as boxplots for each individual year. The horizontal line in the boxplots indicates the median, the box represents the interquartile range, and the

whiskers extend to 1.5 times the interquartile range above and below the box; the dots are outliers.

To quantify the individual and joint importance of the magnitude of P and $E_p$ anomalies, we first calculated the bivariate Spearman rank correlation between the individual anomalies and $Q_{min}$ for the different time windows (30, 60, 90, 120, 182 days) for all years (2000-2018) and for the years with the lowest low flows (2003, 2011, 2015 and 2018). For this analysis we

reduced the original dataset to only those catchments where at least 5 years of $Q_{min}$ data were available, as suggested in WMO (2008). In a next step we used the joint anomalies of P and $E_p$ for all durations 30, 60, 90, 120, 182 days to predict $Q_{min}$ with a multivariate stepwise generalized linear model (GLM) fitted by minimizing RMSE. We then computed the fraction of the GLM's $R^2$ attributable to the individual precipitation and $E_p$ anomalies for each duration, to assess the relative contribution of each anomaly to the prediction of $Q_{min}$. We compared the results for all years to those for the lowest-flow years (2003, 2011,

2015 & 2018) to assess whether the relations between climate anomalies and $Q_{min}$ differed during the driest years.

To test how warm-season low flows are influenced by precipitation and snow processes in the preceding winter, we calculated the Spearman rank correlations between the total December-March precipitation sum and the following warm-season $Q_{min}$,

and between SWE on 1$^{st}$ March and the following warm-season $Q_{min}$. We again report these rank correlations across multiple catchments in histograms, and test the significance of these distributions of correlations by the sign test.

Finally, we assess whether the correlations we obtained between P and $E_p$ anomalies and warm-season $Q_{min}$ are influenced by the extent of human impact in each catchment. We quantify human impact by the fraction of human affected land cover in each catchment. As a proxy for human activity we use the Corine landcover dataset (CLC, 2018) and calculate the fraction of catchment area with "Artificial surfaces". We then show histograms of the rank correlations between P, $E_p$, and $Q_{min}$ in the 20% of catchments with the most human-influenced land use, and the 20% of catchments with the least human-influenced land use, compared to the distribution across all catchments. We assessed the significance of the differences between the obtained distributions by the Student t-Test.

## 3. Results

### 3.1. Spatial patterns of low-flow timing

During the dry years of 2003, 2011, 2015, and 2018, low-flow conditions occurred across large parts of Europe (Laaha et al., 2017; Van Lanen et al., 2016). Annual low flows did not occur simultaneously across Switzerland, but instead occurred primarily during winter in the Alpine regions and summer and autumn in the Swiss Plateau (Fig. 1). In addition, within these two sub-regions, the timing of low flows was still spatially variable, indicating that annual low flows may be surprisingly asynchronous across Switzerland, even in unusually dry years. Within the Swiss Plateau, low-flow timing is more spatially consistent during some years without severe low flows (e.g., 2009, 2013, 2016), than during others (e.g., 2000, 2002, 2004, 2010, 2017).

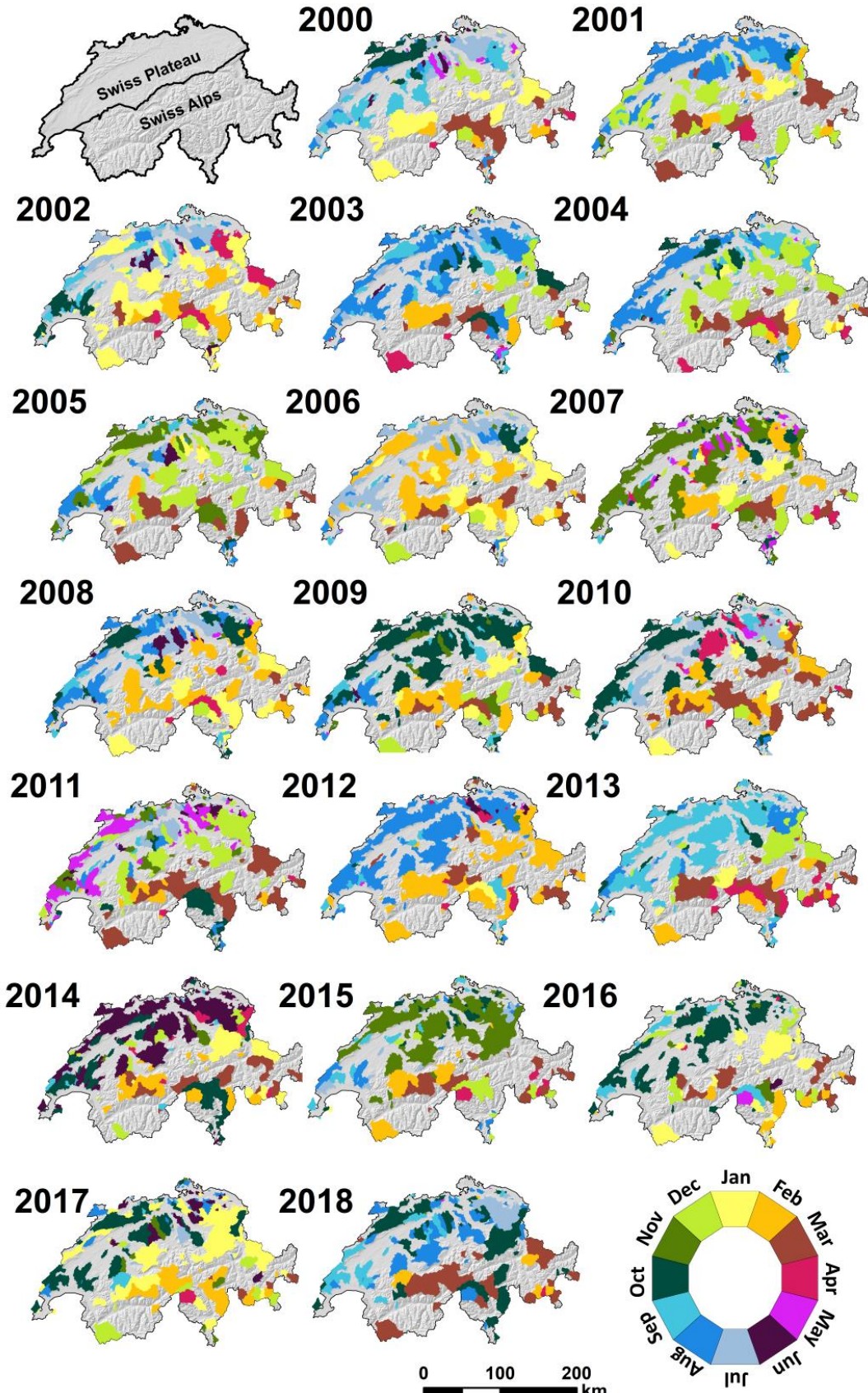

215

**Figure 1: The timing of occurrence of annual low flows across Switzerland for the years 2000 to 2018 in the two main regions: the Swiss Plateau and Swiss Alps (roughly the northern and southern halves of the country, respectively). Low-flow timing tended to be spatially heterogeneous, even in years when large parts of Europe simultaneously experienced severe low flows (2003, 2011, 2015, 2018).**

## 3.2. Climate anomalies control low-flow timing and magnitude

The occurrence of low flows is linked to periods of below-average P and above-average $E_p$ (Fig. 2a&b). However, distinct site-to-site differences exist: at elevations below approximately 1500 m asl, almost all annual low flows occur after periods of anomalously high potential evapotranspiration and anomalously low precipitation (Fig. 2a&b). At higher elevations, by contrast, $E_p$ anomalies have no systematic effect and precipitation anomalies become less important with increasing elevation. This reduced importance of anomalies at these higher elevations is probably because low flows here result primarily from freezing temperatures (or periods of snow accumulation), rather than precipitation or $E_p$ patterns. Low flows at higher elevations occur during the winter months when there is a lack of liquid water inputs to catchments, due to precipitation mostly accumulating as snow, and little snowmelt. These processes are mainly driven by sustained below-zero temperatures. Thus, the main determining factor in winter low flows at high elevations (or in cold environments) will likely be the length of the snow accumulation period, rather than what the exact temperatures were, or how much precipitation occurred.

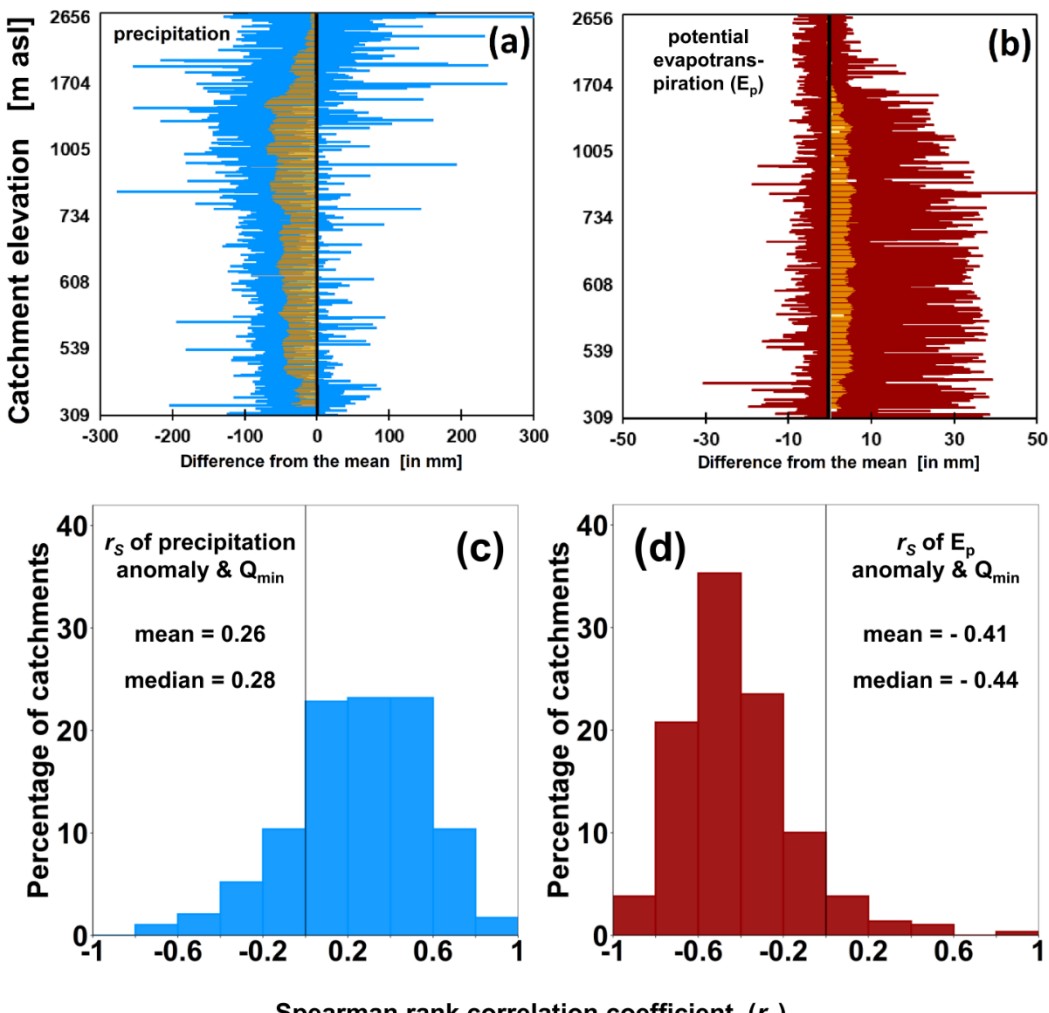

**Figure 2: Altitudinal variation in 30-day anomalies of precipitation (a) and potential evapotranspiration (b) preceding (warm and cold season) annual low flows from 2000 through 2018. Blue and red horizontal bars indicate the range between the minima and maxima of these anomalies at each catchment across the 19 years of this study. Yellow bars show moving averages of these climate anomalies for bins of 10 catchments ordered by elevation. Note that the elevation scale is not linear. Low flows are associated with below-average precipitation (a) and above-average potential**

**evapotranspiration (b). Histograms of rank correlations between anomalies of precipitation (c) and potential evapotranspiration (d) and low-flow magnitudes for warm-season (May through November) low flows across Swiss catchments. Results for cold-season low flows can be found in the supplementary Fig. S2.**

More severe climate anomalies lead to lower low flows (Fig. 2c&d). Spearman rank correlations of magnitudes of the climate anomalies to magnitudes of $Q_{min}$ (shown for the months May through November) indicate that lower precipitation in the 30 days prior to $Q_{min}$ usually results in smaller $Q_{min}$ (median $r_S$=0.28). Similarly, higher potential evapotranspiration usually results in smaller $Q_{min}$ (median $r_S$=-0.44). This indicates that the magnitudes of both precipitation and $E_p$ anomalies affect low-flow magnitudes (p-values < 0.001 according to the sign test), but with substantial site-to-site variability. The $r_S$ between 30-day climate anomalies and $Q_{min}$ does not show distinct spatial patterns across Switzerland (see supplementary Fig.S3). The $r_S$ between the 30-day precipitation anomaly and $Q_{min}$ is not correlated with mean catchment elevation ($R^2 = 0.08$), and the $r_S$ between the 30-day $E_p$ anomaly and $Q_{min}$ is weakly correlated with mean catchment elevation ($R^2 = 0.33$).

### 3.3. Combined effects of climate anomalies on warm-season low flows

The results shown in Fig. 2 indicate that both P and $E_p$ can affect low flows. However, most low flows are not caused by only one driver, but instead result from the combined effects of below-average P and above-average $E_p$ during the same time period. Warm-season low flows usually follow periods of below-average precipitation and above-average potential evapotranspiration (72.2% of low flows fall in the top left quadrant of Fig. 3a). Less than a quarter of the annual low flows occur after periods of below-average precipitation and below-average potential evapotranspiration (20.5% lower left quadrant – Fig. 3a). Only very few annual low flows (7.3%) occur after periods of above-average precipitation. Thus, precipitation anomalies appear to be the most important driver for warm-season low flows in Switzerland, and potentially also in other regions with distinct warm-season low flows. While potential evapotranspiration appears to be less important than precipitation, more than 70% of low flows are caused by a combination of both drivers. The combined effect of above-average $E_p$ thus more than triples the chance of an annual low flow (compared to when precipitation is below average, but there is below-average $E_p$).

In particular, the most severe low flows occur through the combined effects of low precipitation and high potential evapotranspiration. For example, 96% of low flows during the most severe low-flow year (2003, shown by green markers in Fig. 3a) follow periods of both below-average precipitation and above-average potential evapotranspiration. This behavior is not unique to the 2003 event, but was also observed for other years with severe annual low flows such as 2011, 2015 and 2018 (Fig. 3b&c).

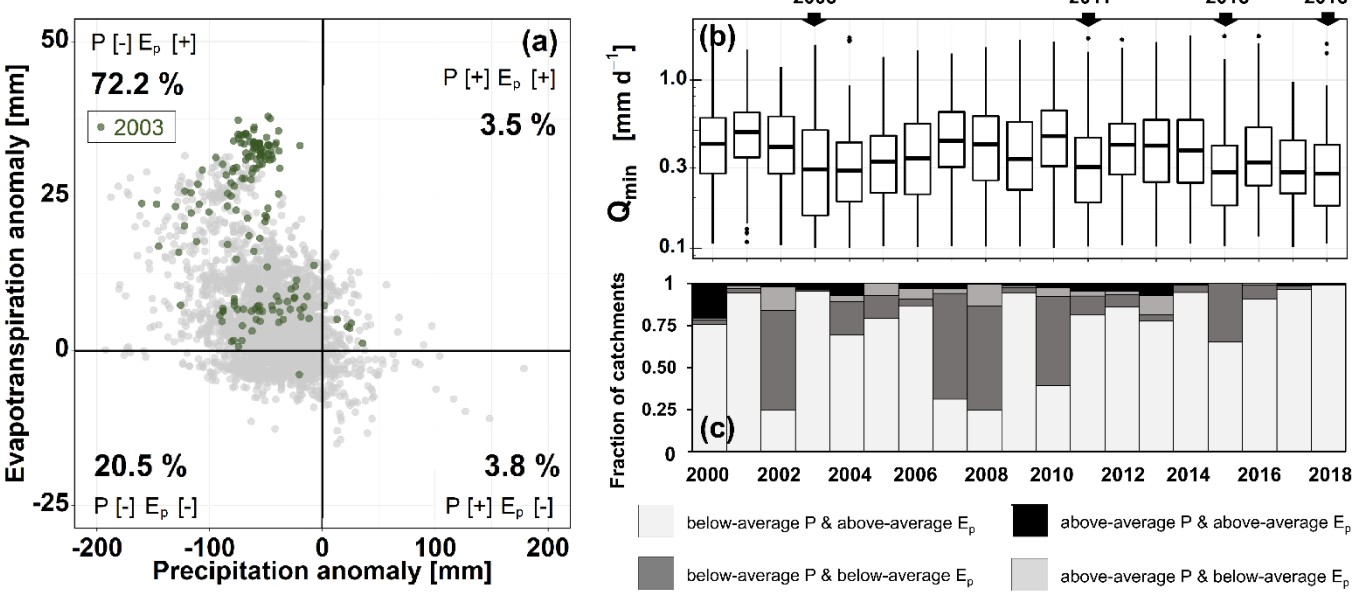

Figure 3: Anomalies in precipitation and potential evapotranspiration 30 days prior to each annual warm-season (May through November) low-flow period in each catchment (grey dots); annual cold-season low flows were excluded (a). The most severe low-flow year during the study (2003) is highlighted in green. Almost all (92.7%) annual low flows occurred following below-average precipitation (the left half of the figure), and 72.2% of all low flows occurred following a combination of below-average precipitation and above-average potential evapotranspiration (the upper left quadrant of the figure). Boxplots of warm-season 7-day minimum flows for the Swiss study catchments (b) and the catchment distribution of the signs of precipitation and evapotranspiration anomalies that preceded these low flows (c). The most severe low-flow years (2003, 2011, 2015, and 2018) were characterized by negative precipitation anomalies and positive $E_p$ anomalies for the large majority of catchments, as indicated by the light grey bars in (c).

## 3.4. Duration of climate anomalies

The magnitudes of low flows are also related to the durations of the preceding precipitation and evapotranspiration anomalies. Longer periods of below-threshold P and above-threshold $E_p$ tend to lead to lower low flows in most of our catchments (Fig. 4). The duration of high $E_p$ is more strongly correlated with low-flow magnitudes than the duration of low precipitation is (mean Spearman correlations $r_S$ of -0.27 and -0.11 respectively; median $r_S$ values differ from 0 at $p<0.001$ by sign test; Fig. 4). The weaker correlation with the duration of below-threshold precipitation probably arises because precipitation is more erratic through the years than $E_p$. A single precipitation event may exceed the precipitation threshold (according to the criterion outlined in Sect. 2.3), but be insufficient to end the low flow in the stream. Low-flow magnitudes are less strongly correlated with the duration of below-threshold precipitation than with the intensity of 30-day precipitation anomalies (compare Fig. 4 with Fig. 1; mean $r_S$ of -0.11 and 0.26, respectively). Similarly, low-flow magnitudes are less strongly correlated with the duration of above-threshold $E_p$ than with the intensity of 30-day $E_p$ anomalies (compare Fig. 4 with Fig. 1; mean $r_S$ of -0.27 and -0.41, respectively).

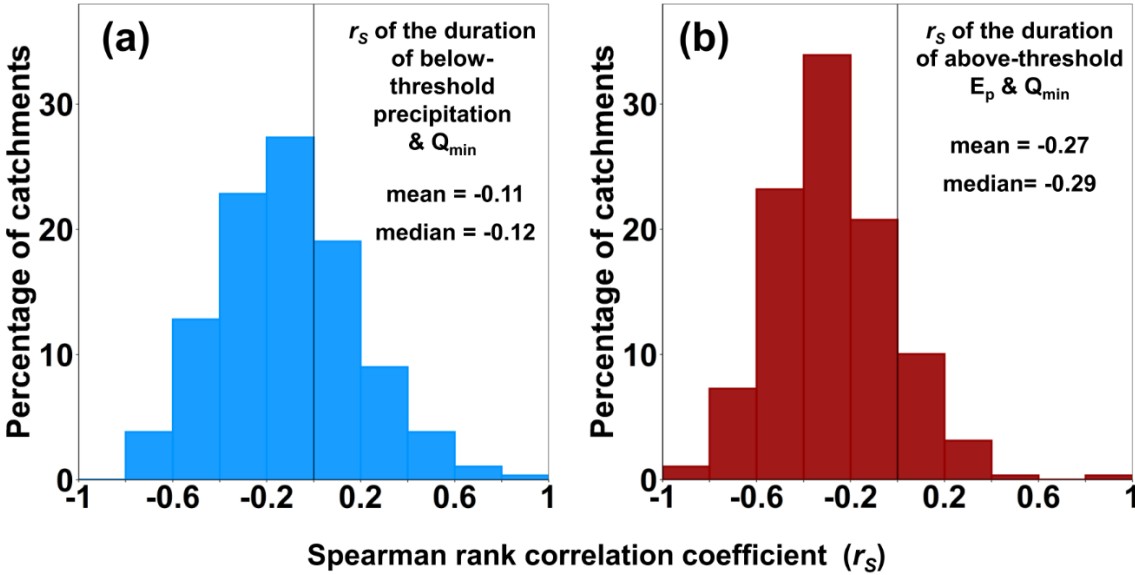

**Figure 4: Histograms of rank correlations between the magnitudes of warm-season low flows and the lengths of the preceding intervals with below-threshold precipitation (a) or above-threshold $E_p$ (b). Longer periods of high $E_p$ are associated with lower low flows, whereas a weaker association is seen between lower low flows and longer periods with low precipitation.**

Summing P and $E_p$ anomalies over time windows ranging from one week to half a year indicates that most low flows can be well explained by anomalies of up to 60 days (Fig. 5h). This is because in the typical Swiss climate, precipitation and $E_p$ anomalies usually last for 60 days or less. This is depicted by the grey cloud of points in Fig. 5, as well as the mean anomalies (indicated by the dotted lines in Figs. 5a-g) which remain approximately stable for periods exceeding 60 days. Thus, while longer precipitation and $E_p$ anomalies would lead to lower flows, most low flows result from anomalies of up to 60 days. This is because most anomalies peak at around that 60-day time scale, which is also indicated by the means of the precipitation and $E_p$ anomalies as functions of timescale (dashed lines in Figs. 5h and 5i).

The severe low flows in 2003, 2011, 2015 and 2018, however, are associated with P and $E_p$ anomalies that grow for much longer, and thus become much larger, than the roughly 60-day anomalies that are typical in this climate (colored symbols in Fig. 5). Long periods of above-average $E_p$ appear to be an important factor for these severe low flows; the colored points in Figs. 5e-g expand more on the y-axis than the x-axis for timescales >60 days. Thus, severe low flows result from longer-lasting (and thus larger) P and $E_p$ anomalies, whereas more typical low flows result from climate anomalies that end after roughly 60 days, as illustrated by Figs. 5h&i.

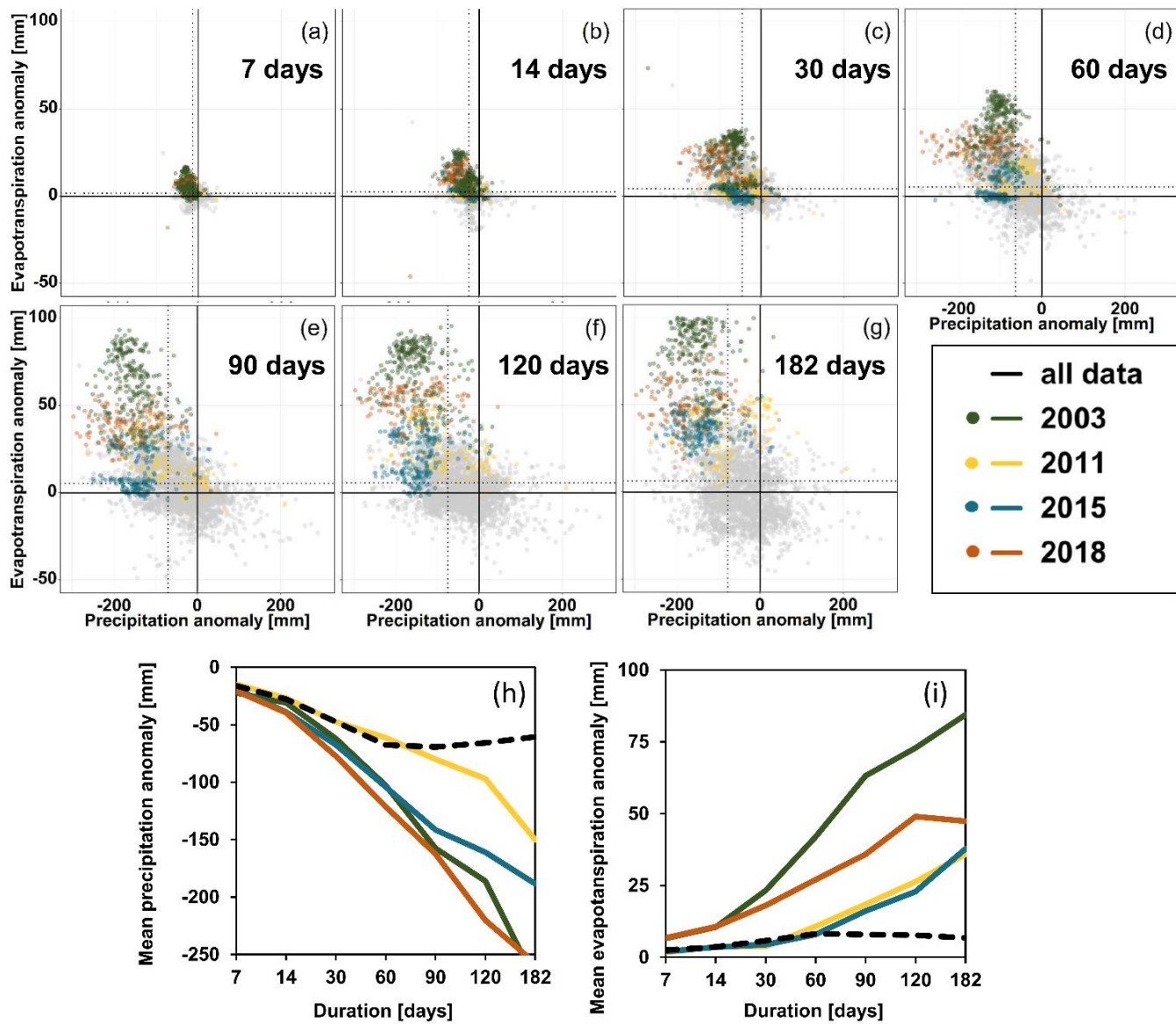

**Figure 5: Cumulative anomalies of precipitation and potential evapotranspiration over 7, 14, 30, 60, 90, 120 and 182 days prior to every annual warm-season low flow in each catchment (a-g), and the evolution of the mean anomalies over the different time windows (h & i). Each grey dot represents the combination of precipitation and $E_p$ anomalies before one low-flow event at one site. Low-flow anomalies in the most severe low-flow years are indicated by different colors (2003 in green, 2011 in yellow, 2015 in cyan and 2018 in orange). The dotted lines indicate the mean precipitation and $E_p$ anomalies. The mean anomalies (dotted lines in all panels) clearly increase within the first 60 days prior to low flows, but show no clear trend over longer time windows. During the most severe low-flow years, however, the mean anomalies continue to increase across all of the time windows examined here. In particular, the $E_p$ anomalies during the severe low-flow years grow well beyond the range that is observed during more typical years.**

## 3.5. The relative importance of P and $E_p$ anomalies for warm-season low-flow magnitudes

We further assessed the relative importance of each of the climate drivers and their duration in predicting the magnitude of annual low flows by calculating the bivariate Spearman rank correlation between each climate driver and $Q_{min}$ as one value for all stations and years together (Fig. 6). The results also include the site-to-site variability in $Q_{min}$; thus the overall $r_S$ correlations are weaker than those shown in Fig. 1c&d. Typical low flows across all years of the observation period (2000-2018) are more

strongly correlated to precipitation anomalies than to $E_p$ anomalies (see also Fig. 1), and this correlation becomes slightly stronger at longer durations. However, during the driest years of our dataset (2003, 2011, 2015 and 2018), the correlation between precipitation anomalies and $Q_{min}$ drops to roughly zero, suggesting that under these extreme conditions low precipitation alone cannot explain the variation in annual low-flow magnitudes. Instead, in these dry years, $E_p$ anomalies retain their predictive power for $Q_{min}$, suggesting a relatively more important role of $E_p$ in dry years.

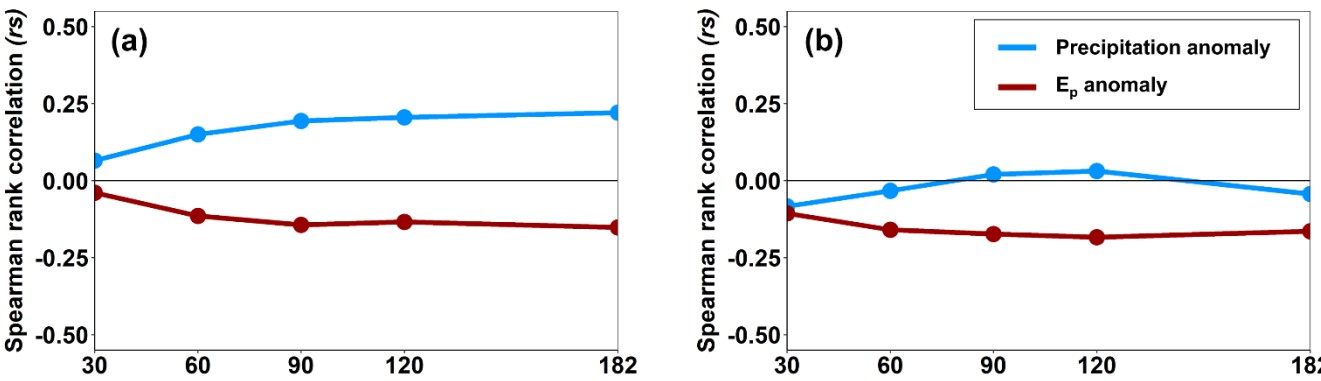

**Figure 6: Bivariate Spearman rank correlation coefficients between precipitation (blue) and $E_p$ anomalies (red) and $Q_{min}$ of warm-season (May to November) low flows for durations of 30 to 182 days, across all stations and years. The overall explanatory power of the climate anomalies in a bivariate regression framework is low, although precipitation anomalies are slightly better correlated to $Q_{min}$ than $E_p$ anomalies in the whole dataset (a). In the four driest years (b) the overall explanatory power of precipitation anomalies is much smaller, and the explanatory power of $E_p$ anomalies is slightly greater, than in all years combined.**

To quantify how much of the maximum predictive power lies in individual anomalies, we first used a multivariate stepwise generalized linear model (GLM) to predict $Q_{min}$ as a function of all precipitation and $E_p$ anomalies for all durations of 30, 60, 90, 120 and 182 days. In Fig. 7 we show the fraction of the model $R^2$ explained by individual P and $E_p$ anomalies for the different durations. Across all stations and years of the observation period (2000-2018), warm season $Q_{min}$ is best predicted by precipitation anomalies with increasing duration (Fig. 7a), which shows the cumulative effect of low precipitation. However, in the years with the lowest annual warm season low flows (2003, 2011, 2015 and 2018) the picture reverses, and instead $E_p$ explains most of the variability in $Q_{min}$. This is true across a wider range of durations, starting even at 30 days. Thus, although precipitation anomalies are a good predictor for typical low flows, low-flow magnitudes in the driest years are more strongly related to $E_p$ anomalies when precipitation is also very low.

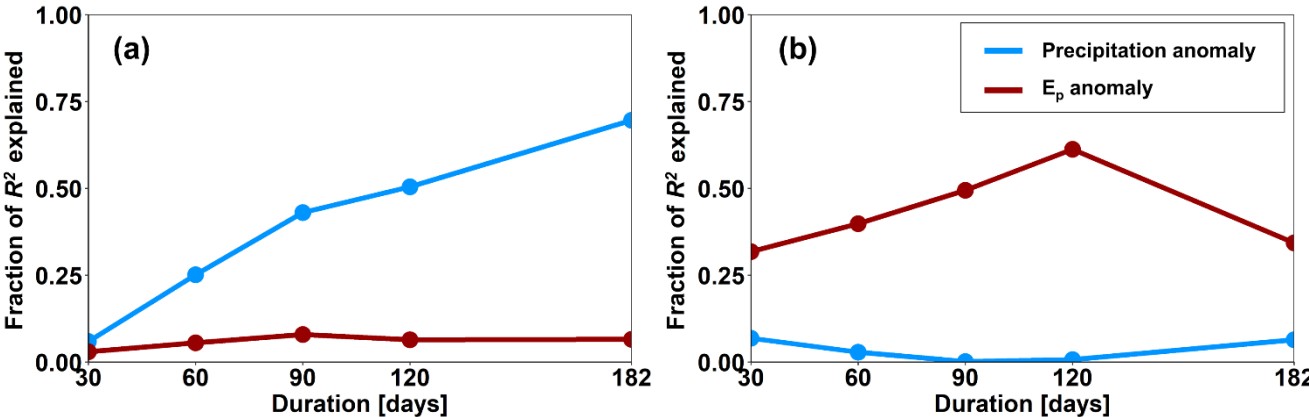

**Figure 7: The fraction of multivariate $R^2$ (calculated by a stepwise generalized linear regression model that explains warm-season low-flow magnitudes using all climate variables and durations) that can be explained by a precipitation (blue) or $E_p$ (red) anomaly of the specified duration. Precipitation anomalies explain most of the variation in $Q_{min}$ when looking at all stations and all years (a). However, precipitation anomalies are not good predictors for low flows that occurred in the driest years (2003, 2011, 2015, 2018), when $E_p$ anomalies are instead much better predictors of $Q_{min}$ variability (b).**

### 3.6. The impact of winter precipitation and snow on warm-season low flows

Previous studies indicate that winter snowpack and snowfall can influence the timing and magnitude of summer low flows in some regions (e.g., Dierauer et al., 2018; Jenicek et al., 2016; Godsey et al., 2014). If this holds true for our study catchments, more winter precipitation (December through March), or higher SWE on March 1st, should lead to larger and later warm-season low flows. To test for this effect, we calculated Spearman rank correlations between winter precipitation totals and subsequent warm-season low-flow magnitudes and timing. The correlations between winter precipitation and the magnitude and timing of $Q_{min}$ (mean absolute $r_S < 0.11$ for both; grey bars in Fig. 8) are weaker than those between low-flow magnitudes and climate anomalies directly preceding low flows (Figs. 1c&d), and they do not vary systematically with elevation. We also calculated the Spearman rank correlations between March 1st SWE and subsequent low-flow magnitudes and timing, and found no strong relationship (Fig. 8, green bars; mean absolute $r_S < 0.17$ for both).

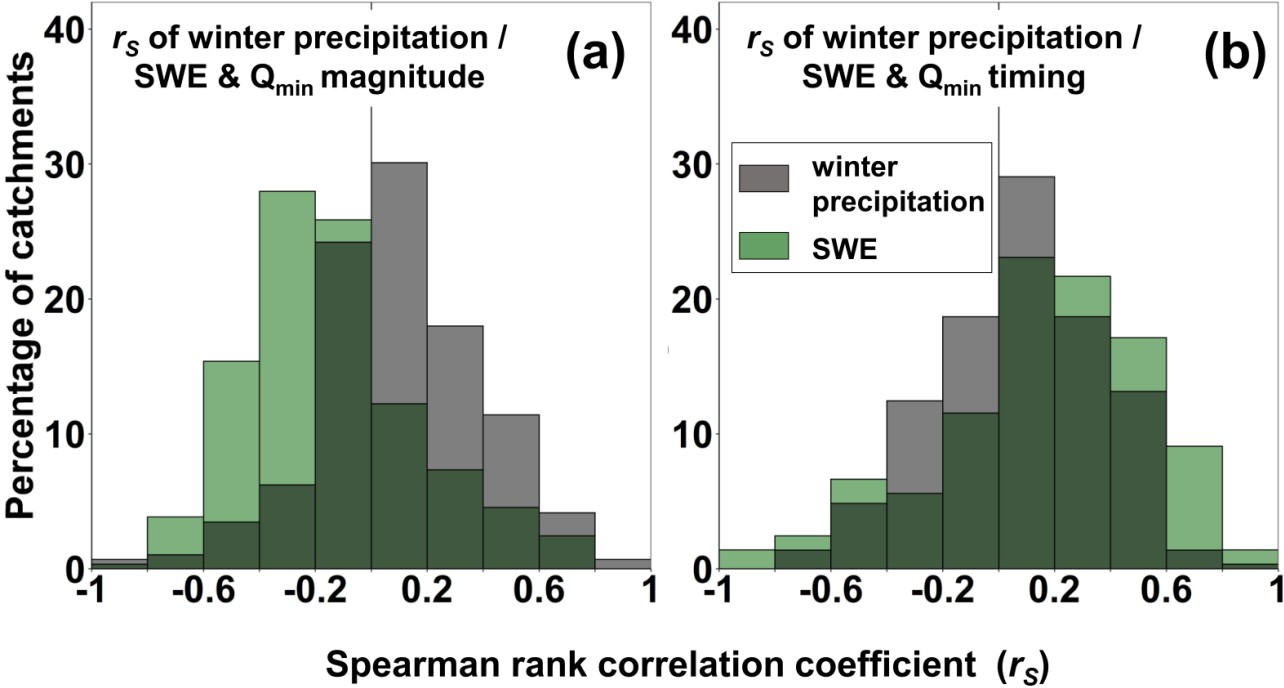

**Figure 8: Histograms of the rank correlations between winter precipitation (December through March; grey bars) and March 1st snow water equivalent (SWE; green bars) and the magnitude (a) and timing (b) of warm-season low flows. Winter precipitation is weakly associated with higher, and later, warm-season low flows, as indicated by the positive rs for the majority of catchments; however, overall correlations are weak, with considerable site-to-site variability.**

## 4. Discussion

### 4.1 Climate anomalies control low-flow timing and magnitude

Anomalies of precipitation and potential evapotranspiration affect the magnitude of low flows, but their influence decreases with elevation (Fig. 2a&b). This pattern is probably not unique to Switzerland, and we expect precipitation and $E_p$ anomalies to also be relatively unimportant in other cold regions where low flows primarily occur in winter (e.g., Dierauer et al., 2018; Laaha & Blöschl, 2006; Van Loon et al., 2015; Wang, 2019), driven by extended freezing periods. However, warm-season low flows are more common globally (e.g., Dettinger & Diaz, 2000; Eisner et al., 2017), suggesting that summer climate anomalies are likely to be important not only for the lower-elevation catchments in Switzerland, but also across many other regions of the world.

We found that the combined effects of P and $E_p$ anomalies shape the occurrence and magnitude of low flows, with the more extreme low flows being driven by longer-duration anomalies. Typical warm-season low flows result from climate anomalies of up to 60 days (Fig.5). In Switzerland, typical low flows result from relatively short climate anomalies, probably because precipitation does not have a strong seasonal signature. In climates that typically have frequent precipitation events, short periods (e.g., one to two months) with less precipitation than normal will most likely precede the annual low flow. Similarly short $E_p$ deviations from the norm often precede typical annual low flows. In the years with the lowest low flows (2003, 2011, 2015 and 2018), the durations of climate anomalies were significantly longer, and especially the impacts of $E_p$ anomalies were larger (Figure 5). This highlights precipitation and evapotranspiration as combined drivers of severe low flows, consistent with findings in several experimental catchments during the 2003 low-flow year (Teuling et al., 2013). Our results suggest that the

magnitude and duration of these precipitation and $E_p$ anomalies are generally important controls on low flows in a large, diverse sample of mesoscale catchments across Switzerland. These compound effects of $E_p$ and precipitation anomalies might also be important for low flows across larger regions (e.g., Stahl et al., 2010), as the climate conditions in Switzerland are comparable to those in other densely populated regions of the world. However, we only analyze these processes on timescales of up to half a year, so long-term memory effects in low-flow generation may not be fully captured by this approach.

The pronounced effect of $E_p$ in the years with the lowest low flows might also reflect the coupling of P and $E_p$ during dry and warm periods. Low precipitation and high air temperature might lead to soil moisture depletion, forcing plants to reduce transpiration. Lower latent heat fluxes and greater sensible heat fluxes from the surface increase air temperature and thus increase $E_p$ while reducing actual evapotranspiration. This complementary relationship between evapotranspiration and $E_p$ can amplify the apparent effect of $E_p$ during (extended) dry periods (Granger, 1989). Conversely, in locations where transpiration is not limited by water availability (e.g., at higher elevations), high temperatures and larger vapor pressure deficits (i.e., high $E_p$) may drive increases in transpiration rates, accelerating the depletion of catchment water stores and thereby reducing runoff. For example, Mastrotheodoros et al. (2020) showed how increased evapotranspiration at higher elevations systematically amplified runoff deficits during severe low flows in 2003 across the European Alps. These processes are especially relevant in view of potential future climatic changes. In Switzerland, climate change is expected to increase temperatures by more than the global average, resulting in warmer summers with less warm-season precipitation (CH2018, 2018). Similar trends are also expected in many other regions. This highlights the effects of water removal through evapotranspiration, especially during extended dry periods, which are expected to become more severe with changing climate conditions.

A small fraction of all warm-season low flows in the period 2000 to 2018 followed periods of above-average precipitation and below-average $E_p$ (4% in lower right quadrant of Fig. 3a). These anomalies are expected to lead to above-average flow conditions, but can nonetheless lead to annual low flows for at least two reasons. First, these low flows occur in years that are relatively wet, with relatively high annual low flows (Fig. 3b). Second, flow conditions in most Swiss catchments are highly seasonal (Wehren et al., 2010; Weingartner & Aschwanden, 1992), meaning that the seasonality of the flow regime can, in some years, outweigh the effects of shorter-term weather.

## 4.2 The influence of winter precipitation and snow on warm-season low flows

Previous work in several Swiss catchments has suggested that the snow-water equivalent (SWE) accumulated in the winter snowpack strongly affects summer low-flow magnitudes (Jenicek et al., 2016). Our more complete dataset of Swiss catchments indicates that winter precipitation and SWE (on March 1[st]) are only weakly related to the magnitude and timing of the following warm-season low flows. In addition, these weak correlations did not significantly increase at higher-elevation catchments, suggesting that even at the higher-elevation sites, SWE is not a major control on warm-season low flows. We caution, however, that this analysis excludes many of the highest-elevation catchments, in which the annual low flow occurs during the winter. Thus the discrepancy between our results and those of Jenicek et al., 2016 probably arises from differences between our respective definitions of low flows. We studied annual 7-day minima, and included only the annual low flows that occur between May and November (thus excluding many high-elevation sites where annual low flows occur in the winter instead), whereas Jenicek et al. (2016) studied 7-day summer minima regardless of whether they are annual minima. Thus, winter precipitation and SWE do affect summer streamflow in Alpine catchments (Jenicek et al., 2016), but our results suggest that for most of the rest of Switzerland, projected changes in winter snowpacks (e.g., Harpold et al., 2017; Mote et al., 2018) might only slightly affect the magnitude and timing of annual low flows that occur during the warm season.

### 4.3 Human impacts on warm-season low-flow statistics

Almost every catchment in Switzerland, and elsewhere where dense gauging data exist, is to some extent affected by human activity (e.g., Grill et al., 2019, Lehner et al., 2011). This could be through, for example, water management operations, water abstractions, hydropower operations, and wastewater treatment plant return flows. Especially in Central Europe, almost no pristine catchments exist and quantitative information capturing all potential human influences on streamflow at catchment scale is unavailable. As described in the methods, we removed any catchments with any obvious anthropogenic influences on streamflow (e.g., from hydropeaking or dams), however some regulation effects may still be present in the dataset.

To assess the impact of human influence on the results, we recalculated Fig. 2c & d for the 20% of catchments with the largest fraction of human-affected land use, and the 20% of catchments with the smallest fraction of human-affected land use. We thereby tested whether the relationships between the 30-day anomalies of precipitation and $E_p$ and the magnitude of warm-season $Q_{min}$ are significantly different in catchments with a lot of human activity compared to catchments with relatively little human activity (Fig. 9). The results were broadly similar with no significant differences between the strongly affected and weakly affected catchments ($p>0.2$ by Student's t-test).

The consistency of the results may be due to the fact that, although human water use during low flows will change their absolute magnitudes (and thus may affect site-to-site differences in low flows, which are not considered here), it may have a smaller effect on their relative magnitudes from year to year at any given site. Thus human influences may not greatly alter the rankings of annual low flows throughout the observation period; drier years are still expected to have lower low flows and wetter years are still expected to have higher low flows, largely independent of human influences. Therefore the Spearman rank correlation coefficient is likely to be a relatively robust index for assessing the effects of climate anomalies on the timing and magnitude of annual low flows. Recent studies across US catchments have also found limited effects of human influence on low flows compared to climate drivers (Ferrazzi et al., 2019; Sadri et al., 2016). Nevertheless, the unexplained variance in our established relationships suggests that human-induced shifts in the $Q_{min}$ ranking may have an effect on low-flow behaviors in some catchments.

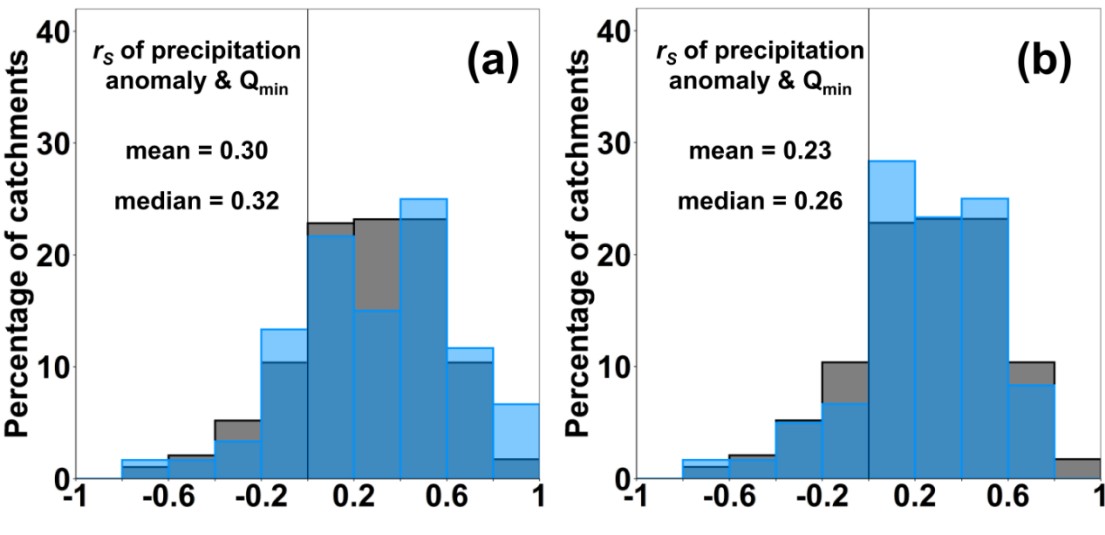

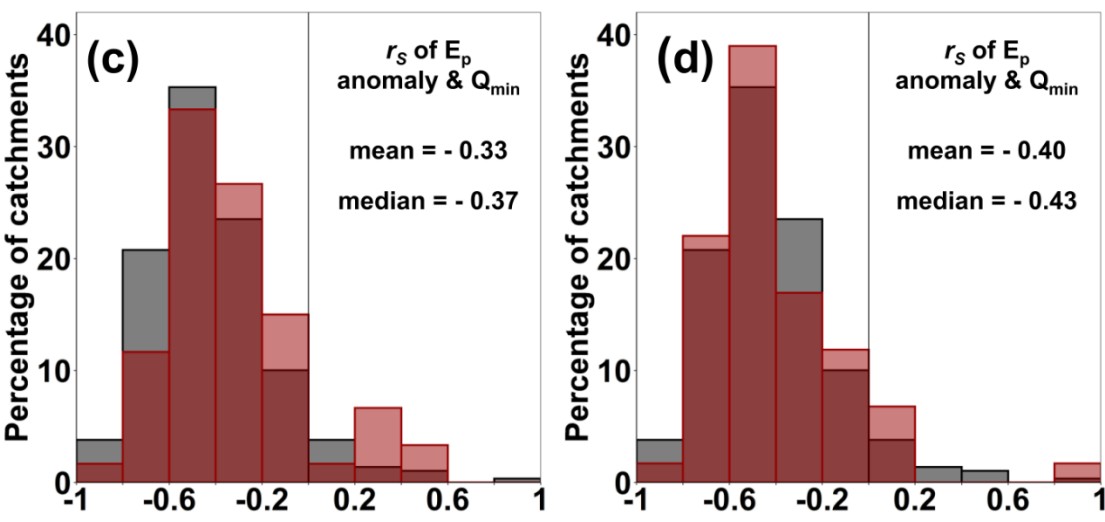

**Figure 9: Histograms of rank correlations between low-flow magnitudes and anomalies of precipitation (a & b) and potential evapotranspiration (c & d) for warm-season low flows across Swiss catchments. The left side (a & c) shows the distributions for the 20% of catchments with the least human impact (blue & red) on top of the distributions for all data (grey). The right side (b & d) shows the distributions for the 20% of catchments with the most human impact (blue & red), again plotted on top of the distributions for all data (grey). The observed distributions of correlations between the 30-day climate anomalies and the magnitudes of low flows are similar in catchments with the most and the least human activity.**

**4.4 Broader implications**

Our overall results are largely consistent with previously discussed drivers of low flows (e.g., Teuling et al., 2013; Woodhouse et al., 2016; Hannaford, 2015). Our work builds upon past research by studying a large dataset which shows the variability and consistency in low-flow/climate relationships among many catchments. We also quantify the effect of the duration of climate anomalies and analyze the interplay of P and $E_p$ as drivers. Our work thereby emphasizes how both

precipitation and $E_p$ anomalies are important drivers of low flows, especially during severe low flows. This is in line with a growing literature on severe events arising from the interplay of multiple drivers (e.g., Zscheischler et al., 2018). Our study also highlights that the relevant properties of low-flow drivers are multidimensional: their magnitudes, timings, and durations all matter. For example, in a lower-elevation catchment, a precipitation anomaly in spring will not have the same impact as a similar anomaly in autumn. Likewise, periods of above-average $E_p$ will have different implications for streamflow in May than

they would in September. Thus, antecedent catchment conditions matter. It is not sufficient to look at climate anomalies alone as drivers of low flows, since they may have different implications at different times of the year. Although our study is based on a network of Swiss catchments, we expect our findings to be more broadly applicable to climatically similar regions as well. We see similar patterns in low-flow seasonality in other regions of the world (e.g., Laaha & Blöschl, 2006; Demirel et al., 2013; Dettinger & Diaz, 2000), suggesting that the effects of climate anomalies in these other regions may also be largely

similar. For example, the severe summer low flows in California in recent years have been driven by below-average precipitation magnified by above-average temperatures and thus potential evapotranspiration (Diffenbaugh et al., 2015). Van Loon et al. (2015) and Van Loon & Laaha (2015) reported similar driving mechanisms for low flows in Austria and Norway. Thus, our approach for assessing the effects of multiple dimensions of climate impacts (i.e., timing, duration and magnitude) on low flows could potentially be used to derive insight into low flows in other regions.


## 5. Conclusions

Annual low flows in Switzerland typically occur in two distinct seasons: in winter at higher elevations due to sub-freezing temperatures, and in summer and autumn at lower elevations, following periods of above-average potential evapotranspiration and below-average precipitation (Figs. 2a&b). The magnitudes of these climate anomalies strongly affect the magnitudes of

annual low flows across our network of catchments (Figs. 2c&d). Almost all (about 92%) of our catchments' annual low flows follow periods of unusually low precipitation, and many (about 70%) also follow periods of unusually high potential evapotranspiration (Fig. 3a). Thus, most low flows arise from the combined effects of precipitation and $E_p$ anomalies. Severe low flows, such as in the years 2003, 2011, 2015 and 2018, almost exclusively occurred after anomalies in both precipitation and $E_p$ (Fig. 3a). During these especially dry years, low flows occurred simultaneously across large parts of Europe, but their

timing was highly variable across Switzerland (Fig. 1). Longer periods of below-threshold precipitation and above-threshold $E_p$ generally led to lower low flows (Fig. 4). Anomalies preceding low flows typically acted over timescales of up to 60 days, while precipitation and $E_p$ anomalies in unusually dry years (2003, 2011, 2015 and 2018) grew for much longer, and thus became much larger (Fig 5). Long periods of above-average $E_p$ appear to be especially important drivers of the most severe low flows (Fig. 5). Typical low flows were mainly driven by precipitation anomalies; however, the low flows in the driest

years (2003, 2011, 2015 and 2018) were more related to $E_p$ anomalies (Fig. 6 & 7). Total winter precipitation (and SWE) affected the magnitude and timing of warm-season low flows (Fig. 8), but was less important than the climate anomalies in the month prior to the low-flow period (Figs. 1c&d). Our results describe how the timing, magnitude and duration of precipitation and $E_p$ anomalies drive warm-season low flows across Switzerland. In combination with seasonal weather forecasts, these results could help in predicting and managing low flows.


**Data availability:** The data that support the findings of this study are available in the ETH library open-access repository. Discharge time series can be obtained from FOEN (Swiss Federal Office of the Environment) and Swiss Cantonal Authorities; meteorological data can be obtained from MeteoSwiss, geodata from Swisstopo (Swiss Federal Office of Topography). Contact information for these agencies is provided in the Supplementary Material.


**Author contribution:** MF, WB, PM designed the study; MF performed the analyses and wrote the first draft, all authors discussed the results and edited the manuscript.

**Competing interests:** The authors declare that they have no competing interests.


**Acknowledgements:** The project was funded by the Swiss Federal Office of the Environment (FOEN). We thank FOEN and the Cantons Aargau, Basel Landschaft, Bern, Geneva, Graubünden, Luzern, Neuchatel, Schaffhausen, Solothurn, St.Gallen, Thurgau, Ticino, Vaud and Zürich for providing discharge data, the Swiss Federal Office of Meteorology (Meteoswiss) for providing gridded climate data and the Swiss Federal Office of Topography (Swisstopo) for providing geodata. We thank

Anna Costa for additional analyses during the revision process, as well as the editor Kerstin Stahl, the four anonymous reviewers and Ryan Teuling and his students for their comments that greatly improved the original manuscript.

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
