# Peer review of "Effects of climate anomalies on warm-season low flows in Switzerland"

_Hydrology and Earth System Sciences, 2019_

## Referee Comment (RC1) · Anonymous Referee #1 · 14 Oct 2019

Floriancic et al. explore how anomalies in precipitation and potential evapotranspiration shape the occurrence and magnitude of annual low flows across 380 Swiss catchments. The varying time period for the precip and PET anomaly calculation, with the end point being the day of the low flow, is a novel method for completing the joint analysis of climate drivers on annual low flows. I found the conclusions to be well-supported by the data. I particularly like how Figure 6 illustrates the role of long periods of PET in development of extreme low flows. The paper is well-written, and the methods are clearly outlined.

I find this manuscript to be a significant contribution to the field, and I recommend it for publication in HESS. I have the following few minor/technical comments that should be easy to address:

[Figure]

Abstract L28: Most of the paper focuses on warm-season low flows, and the authors posit that precip and PET anomalies are "relatively unimportant" on winter low flows. Based on this statement, I suggest removing the reference to winter low flows on this line, since they are not jointly shaped by precip and PET.

L100: change "There were years whose lowest" to "There were years when the lowest"

L140-142: Sentence starting with "However," incorrect figure reference at end of sentence – should be Fig. 2a&b.

Figure 4: Suggest changing the color-scheme to something that is color-blind friendly.

L308-311: Based on the winter precipitation versus annual low flow analysis completed in this study, I don't think this statement is sufficiently supported. As stated earlier in the paragraph, winter precipitation does not always accurately represent SWE. With such a range of catchment elevations (and thus climate conditions), a more detailed analysis would be needed to determine the impact of SWE on summer low flows.

L314: "most work has discussed individual drivers" – statement suggests that some work has analyzed multiple drivers of low flows, but no studies are referenced here. Section should reference the relevant studies listed in the introduction on L68-70.

L319-321: I struggled to directly relate these broader implications statements to the results. How will the impacts be different between spring and autumn? What are the different implications of PET anomalies in May versus September? These implications are likely obvious to the authors, but on the first read through – I did not make the connection.

L350: Data availability – Rather than making the data available "upon request", I would encourage the authors to provide open access to the compiled data used in their analysis (streamflow, catchment-averaged weather and climate conditions, PET, etc.) through an archiving medium such as figshare.com. While not essential, it would be beneficial.

---

## Referee Comment (RC2) · Anonymous Referee #2 · 15 Oct 2019

In this work, the authors assess how anomalies in precipitation and potential evapotranspiration shape occurrence and magnitude of annual low flows for 380 Swiss catchments comparing preceding precipitation and evapotranspiration for different periods to the annual minimum flow. After an initial analysis of annual low flows, the authors decided to focus on summer low flows for the rest of the analysis. The paper is clearly outlined and easy to read.

I agree with reviewer 1 that I particularly like Figure 6 illustrating the increasing role of PET during development of extreme low flows.

However, different to reviewer 1 I have some major comments that I find important to address before publishing:

**Summer and winter low flows**
After an initial analysis of annual low flows, the authors decided to focus on summer low flows for the rest of the analysis. This analysis takes relatively much of the full paper both in text and in Figures (1, 2 and 4). While I agree that it is important to differ between summer and winter low flows particularly when analyzing the drivers, I do not see that the results of general occurrence are new (e.g. Smakhtin 2001 and references therein, Fiala et al. 2010 and basically all runoff regime literature for Switzerland in particular e.g., Weingartner and Aschwanden, 1992) nor that they deserve this weight in the article. I suggest to minimize this to the introduction refereeing to the relevant references and remove Figure 1 and focus in Figure 2 on only the summer events or split in to summer and winter low flows at the beginning and then assess both for winter how snow (or precipitation and temperature) shapes low flow occurrence and magnitude and for summer how precipitation and evapotranspiration shape low flow occurrence and magnitude.

**More focus on the shaping**
Instead of counting and presenting summer and winter low flows representing the annual low flow, I think it would be more interesting to try to add on the shaping of the low flows caused by precipitation and evapotranspiration. For instance, in addition to correlation between precipitation and PET separately on extreme or less extreme low flows it would be interesting to look at their combined effect. And then how much of the combined effect could be attributed to precipitation and to PET. This would allow a better relative quantification and really add to the values of this study.

**Choice of summer months**
The authors focus at extreme low flows and the preceding conditions and chose 2003, 2011, 2015 and 2018 as the relevant years. The drought in 2011 was finished for most catchments before July, I urge the authors to look into the data and if so, adjust the analysis by either treating 2011 differently, i.e. not considering it a summer but a spring low flow and or change the analysis period for all years that was defined by the

authors Jul-Nov.

**Terminology**

- There is a seamless transition from "drought" to "low flow" and mixed use of "droughts" and "deficits" while the citations support both. I find this mix critical since already deficit (even when regularly/seasonally occurring) causes low flows but meteorological droughts are larger deficits than normal, i.e. the regularly/ seasonally occurring deficit. Please, revise the introduction to distinguish clearly between these. This would help the reader in the analysis that follows.

- Likewise, the authors use often only the term "low flow" when actually referring to "extreme low flow". This can result in wrong statements (e.g. L72 "low flows are exceptional flow conditions" or L186 "triples the chance of an annual low flow"). And I would ask the authors to revise and correct the usage throughout the manuscript.

**Mixed results and discussion section**
In my opinion results and discussion should be separated. This allows to focus on the results. Only then what we can learn from the results and where we might to be a bit more careful, then also relate and compare to what was done elsewhere and where limitations and possibilities lay. In the present form the manuscript mixes these aspects and is more difficult to search for s specific result /argument this way.

**Minor comments**
L25-27 Remove this sentence for the abstract
L40/41 is "landscape" only including surface features? Maybe use rather catchment properties".
L72 this statement is not correct they occur every year. Make clear that it is about extreme low flows here!

[Figure]

L92-94 a summary table (maybe only in the supplementary material) would be helpful
L121 1200m asl, why this threshold? Why not a range?
L130 remove "However, this remains to be tested".
L147-150 For these low flows, people are usually prepared. Here it would be inter-esting how much more extreme are others. If it was due to a lack of precipitation, the signal should be visible in spring melt.
L154 "suggesting . . ." could be also formulated that lower elevation Swiss catchments could be representative sample for global summer low flow? (and maybe not even global but for humid regions with seasons?); This part would better fit in the introduc-tion or methodology/catchment section.
L161 altitudinal variation in 30-day anomalies: could that be influenced by catchment size? A large catchment might not react on such an anomaly a small catchment not anymore if the driving anomaly is at the beginning of the period.
L174 "substantial site to site variability" can this be quantified?
L181 can these 8L189-196 These results are not surprising (the authors refer even to studies that found the same) but nicely illustrated and supported by the data. However, it would add to the value of this study to quantify the contribution of precipitation and evapotranspiration.
L189-196 Consider also to compare to Stahl et al. 2010
L196-197 delete sentence
L203-204 it is possible to avoid that by the study design (see also my major comment on seasonal split)
L222-228 that depends on how one looks at the drought: is the same scale as in the references used? The effect was also found in low flows but maybe not in the metric "annual low flow", again distinguish between drought and low flows (see also major comment on terminology above) Figure 4: Looking at the figure makes me wonder how/if the regulated catchments might influence the pattern presented. Could that be picked up in the discussion?
L241 How brief since the study is about anomalies?

Figure 6: the last two sentences of the caption should go rather in the main text; add a legend to the lower plot of the figure.

L285 -311 The influence of snow might change also depending on how wet the catchment was before snowfall and on the distribution of snowfall /melt during winter not only on the pure amount of snow that fell.

L324-326 In the California snow mattered a lot for the droughts/ low flows during summer!

L331-333 I would drop that sentence

**Technical comments**

While I find that active voice generally a good choice, I would avoid starting every sentence with "we" (e.g. 2.1 but also elsewhere), please revise.

L22 "dry years saw" please rephrase
L28 redundant, delete either "could" or "potentially"
L44 "(PET)" -> ", PET"
L44 remove "should" and "usually"
L46 remove "made"; split sentence: "...to a catchment. Hence a sustained..."
L49 Make a new paragraph
L64 "smaller" = "lower"?
L66 "comes" -> "occurs"; remove "the" before summer and before winter
L79 "useful" = "suitable"?
L94 "quantified" -> "estimated"
L100 "years whose lowest annual flows were much" -> "years with lowest annual flows much"
L103 "low flows" -> "extreme low flows"
L224 remove "the" before summer and before winter
L238 "more strongly" -> "higher" (also in L244)

[Figure]

L259-262 rephrase to make more concise

**References**
Fiala, T., Ouarda, T. B., Hladný, J. (2010). Evolution of low flows in the Czech Republic. Journal of Hydrology, 393(3-4), 206-218.
Smakhtin, V. U. (2001). Low flow hydrology: a review. Journal of hydrology, 240(3-4), 147-186.
Stahl, K., Hisdal, H., Hannaford, J., Tallaksen, L., Van Lanen, H., Sauquet, E., ... Jordar, J. (2010). Streamflow trends in Europe: evidence from a dataset of near-natural catchments. Hydrology and Earth System Sciences, 14, p-2367.
Weingartner, R., Aschwanden, H. (1992). Discharge regime–the basis for the estimation of average flows. Hydrological Atlas of Switzerland, Plate, 5, 26.

---

## Referee Comment (RC3) · Anonymous Referee #3 · 15 Oct 2019

General comment

The objective of presented study is to investigate how precipitation (both summer and winter) and PET anomalies influence low flows across Switzerland both in typical and exceptionally dry years. In my opinion, authors provided detailed and important insight into climatic drivers controlling low flows based on data assessment from 380 catchments in Switzerland. In general, I found the results interesting, although the methods used are not novel. I found the main contribution in assessing a large number of catchments which may help us to better understand why catchments sometimes behaves differently, which are main controls and thus what may happen in the future in a warming climate. Thanks to a large number of catchments covering different elevations, I think the results can by generalized to other regions, at least to those located in similar climates. In this respect, the results have an international value and may be very useful for hydrological community. Therefore, the results are important and certainly appropriate for HESS. However, I have some comments listed below, which need to be addressed before I can recommend the manuscript for publication. These comments are mainly related to methods and results interpretation. I hope that these comments will help authors to improve the manuscript.

Major comments

Authors used winter precipitation to show how winter and snow conditions are important for summer low flows. Although this is an important aspect especially for higher elevation catchments, I am not sure to which degree authors were able to capture the snow effect by selecting just winter precipitation as a single variable. The winter precipitation does not tell us whether the precipitation is falling as rain or snow. This is, in my opinion, very important since snow contributes to runoff much later than rain and thus influence the seasonality of groundwater recharge and potentially summer low flows. Therefore, I am not sure whether the winter precipitation could correctly capture this issue well enough to make any general conclusion. Using some snow-related metrics (snowfall fraction, snowfall water equivalent, annual maximum SWE or similar) would be perhaps better to show whether there is (or is not) any relation. Therefore, I would be careful with interpretation going towards the role of snow. I do not see much evidence in authors results to make some conclusion, although several previous studies quantified this effect at different elevations.

I am not fully convinced that assessing both winter and summer low flows is a good approach since the meteorological drivers are different for both of low flows types (see e.g. Harpold et al. (2017) for general overview). I found the mixing of both types throughout the manuscript sometimes a bit confusing. Authors did first analysis (Fig. 1 and Fig. 2) using annual low flows from all catchments (regardless whether they were summer or winter) and later they decided to further analyse only summer low flows. Although I would maybe prefer to focus only on summer low flows in the study, I

accept the authors' decision to make first some results related to both winter and summer low flows together and later focus just on summer low flows. However, I am a bit confused how authors exactly proceeded to select catchments and years for summer low flows analysis (but maybe I only missed something). First, it seems that authors analysed all seasonal low flows occurred in the warm period for all study years (even in case that annual low flow occurred in winter). However, later (L306-307) it seems that authors completely excluded catchments/years in case that annual low flow occurred during winter. The latter approach could result in excluding many of the highest elevation catchments from the analysis (and thus it might lead to the conclusion that winter precipitation is not an important signature to influence summer low flows as noted in the previous comment). Therefore, please clarify how you proceeded. I would think that the first mentioned approach is more appropriate and should be used in the analysis (especially in case you are focusing on the role of snow or winter precipitation in addition to role of previous precipitation and PET).

Regarding to the comment above, I think that mixing the summer and winter low flows in Fig. 2 (top panels) is not a suitable approach since the climatic controls are different for both type of low flows. As it is now, you are losing a lot of information, especially in higher elevation catchments, because you are trying to describe (mostly) winter low flows in these catchments using variables, which are not much relevant. Therefore, I would suggest to make the Fig. 2 just for June (or July) to November low flows. Then you would see, whether the precipitation and PET are important drivers for summer low flows even at highest elevations or whether the figure would suggest that there might by also something else (e.g. snow from preceding winter). In case you decide to keep Fig. 2 as is, please consider to split it into two figure, since the mixing of annual and summer low flows in one figure (top panels vs. bottom panels) is, in my opinion, confusing.

Authors calculated preceding PET as one of the main climate drivers. However, physically correct way is to use actual evapotranspiration (AET) instead of PET. I am aware,

that calculating AET would not be such easy. Nevertheless, the relation between PET and AET is not always straightforward since higher PET do not necessarily mean higher AET (especially in lower elevation catchments with lower precipitation, higher water demand and thus lower water availability). Therefore, I would appreciate more discussion related to PET vs. AET interactions.

Minor comments

One of the conclusions is that low flows are controlled by either low precipitation or high PET or combination of both. This is not surprising since there are not many other options (at least for summer low flows in near-natural catchments). Therefore, I would rather highlight implications which arose from results, but which are not such trivial. In this respect, I would recommend to slightly reformulate the respective part of abstract, short summary and perhaps also hypotheses (line 75) to better highlight the novelty of your work.

Authors used winter precipitation, but this signature is not mentioned and explained in the methods section.

I suggest to create a map showing the location of catchments. I think that just a simple map of Switzerland with shaded DEM with points showing the position of catchments outlets would help those readers not familiar with Swiss hydrology. Maybe, catchment points might be coloured according to catchment elevation (or something similar).

It is a bit questionable to describe the previous precipitation just using the sum of precipitation from the defined preceding period. The reason is that the importance of precipitation for the low flow at the specific date changes when going back in time (precipitation closer to the day with the low flow is more important than that occurred earlier). Did you also consider applying some kind of continuous precipitation index, e.g. current precipitation index CPI (Smakhtin and Masse, 2000)? I would appreciate more discussion on this issue.

[Figure]

L101: Please, add a brief information why you remove "unusually high annual low flows". I see the point, but it would be good to clarify it.

L147-149: Could you please add some references regarding this statement? Just to avoid speculations, since you cannot prove this based on your results.

Fig. 2: Here it is nicely shown that PET anomalies are relatively less important compared to precipitation anomalies (inter-annual variations up to 40 mm for PET, but up to -200 mm for precipitation).

L201-202: Maybe I missed something, but I do not see the described effect (wet years) from Fig. 3b

Fig. 3a: Why there are still some years when low flows are preceded by above-average precipitation and below-average PET (bottom-right quadrant)? Would the figure looks similar also for other than 30 days time-windows (e.g. 60 and longer)? You suggested some explanation in lines 201-205, but could you be more specific?

Fig. 4: This figure shows both summer and winter low flows. However, earlier you stated that only summer low flows are analysed starting from Fig. 3 (L155-156). Therefore, I would consider putting this figure as a Fig. 3 to be consistent. Additionally, I would maybe change the colour scale by using "cold" colours for cold months and "warm" colours for warm months.

L237: Please, specify the thresholds for "below-threshold" precipitation and "above-threshold" PET (maybe in methods as well).

L286-287: This hypothesis would be correct only in case that most of winter precipitation would occur as snow. Therefore, not higher winter precipitation in general, but higher snowfall (snow storages) should lead to larger and later summer low flows (but only at high elevations with high snow storages). Please, consider reformulation. In this respect, you correctly pointed to the fact that winter precipitation sums do not accurately represents SWE (L299).

L325. I would maybe add "winter" to describe the precipitation in California. In contrast to humid catchments in Central Europe, the previous summer droughts in California were mostly driven by lack of winter precipitation (and snowpack).

Technical corrections

L95: Please, add the reference to RhiresD and TabsD products

L106: I would not use the term "long-term" for time period of 18 years. Instead, I would directly specify from which time period the average has been calculated.

L142: I think you wanted to refer to Fig. 2a&b

Figure captions: A lot of text in figure captions (Fig. 2, 3, 5 and 6) is related to figure interpretation rather than figure description. In my opinion, these parts would better fit directly to the main text.

L177: "Our previous results ...". I would remove "previous" since it implies something you did in some previous study. Alternatively, be more specific instead (e.g. "results shown in figure/section no. ...").

Fig. 6: Please, consider larger axis captions to increase the readability.

References

Harpold, A., Dettinger, M. and Rajagopal, S.: Defining Snow Drought and Why It Matters, Eos (Washington. DC)., doi:10.1029/2017EO068775, 2017.

Smakhtin, V. Y. and Masse, B.: Continuous daily hydrograph simulation using duration curves of a precipitation index, Hydrol. Process., 14(6), 1083–1100, doi:10.1002/(SICI)1099-1085(20000430)14:6<1083::AID-HYP998>3.0.CO;2-2, 2000.

---

## Author Comment (AC1) · 15 Oct 2019

**We thank Anonymous Referee 1 for the positive and constructive feedback. We appreciate the suggested corrections and will address them in our revised version. Below we list our response (in bold) to the reviewer's comment***(in italics).*

*Floriancic et al. explore how anomalies in precipitation and potential evapotranspiration shape the occurrence and magnitude of annual low flows across 380 Swiss catchments. The varying time period for the precip and PET anomaly calculation, with the end point being the day of the low flow, is a novel method for completing the joint analysis of climate drivers on annual low flows. I found the conclusions to be well-supported by the data. I particularly like how Figure 6 illustrates the role of long periods of PET*

[Figure]

*in development of extreme low flows. The paper is well-written, and the methods are clearly outlined. I find this manuscript to be a significant contribution to the field, and I recommend it for publication in HESS. I have the following few minor/technical comments that should be easy to address:*

**Thank you.**

*L100: change "There were years whose lowest" to "There were years when the lowest"*

**We will correct this.**

*L140-142: Sentence starting with "However," incorrect figure reference at end of sentence – should be Fig. 2ab.*

**We will correct this.**

*Figure 4: Suggest changing the color-scheme to something that is color-blind friendly.*

**We will evaluate the color options for Figure 4 to improve its visibility and make it colorblind friendly.**

*L308-311: Based on the winter precipitation versus annual low flow analysis completed in this study, I don't think this statement is sufficiently supported. As stated earlier in the paragraph, winter precipitation does not always accurately represent SWE. With such a range of catchment elevations (and thus climate conditions), a more detailed analysis would be needed to determine the impact of SWE on summer low flows.*

**We agree that the role of SWE in summer low flows cannot be directly inferred with the available dataset. In the revised manuscript we will point out that we compare the amount of previous winter precipitation (rather than SWE) to magnitude and timing of summer low flows.**

*L314: "most work has discussed individual drivers" – statement suggests that some work has analyzed multiple drivers of low flows, but no studies are referenced here. Section should reference the relevant studies listed in the introduction on L68-70.*

**In the revised version, we will list the appropriate citations here.**

*L319-321: I struggled to directly relate these broader implications statements to the results. How will the impacts be different between spring and autumn? What are the different implications of PET anomalies in May versus September? These implications are likely obvious to the authors, but on the first read through – I did not make the connection.*

**In the revised version, we will emphasize how antecedent conditions with regard to soil moisture state and subsurface water availability, and the water demand by vegetation in a catchment, matter. It is not sufficient to look at (the combination of) anomalies only, as the same combinations may occur throughout the year with different results, depending on soil moisture and vegetation state. Therefore, it is also necessary to include the timing of these anomalies together with general climatology of a basin.**

*L350: Data availability – Rather than making the data available "upon request", I would encourage the authors to provide open access to the compiled data used in their analysis (streamflow, catchment-averaged weather and climate conditions, PET, etc.) through an archiving medium such as figshare.com. While not essential, it would be beneficial.*

**We will publish the dataset in the "open access" ETH library collection. However, unfortunately we cannot supply the full daily mean streamflow dataset as they are only available at the Swiss cantonal authorities and the Swiss Federal Office of the Environment upon request. Nevertheless, our dataset will include the date of low flow occurrence (2000 – 2018) and the magnitude of the annual lowest flow (Qmin), and we will include contact information for the relevant organizations where the streamflow time series can be obtained.**

---

## Referee Comment (RC4) · Anonymous Referee #4 · 16 Oct 2019

**General comments:**

In this paper, the authors assess to what extend precipitation and PET anomalies trigger summer low flow events in Switzerland. The assessment employs Spearman correlations between low flow magnitude and climate anomalies in P and PET aggregated over varying lead times before the peak of the low flow event. The correlations are overall weak, but still indicate that most low flows arise from the compound effects of precipitation and PET anomalies, with longer and larger anomalies related to more extreme low flow events, as one would expect. The assessment of lead times before the peak of the event is not new (e.g. Fangmann and Haberlandt, 2019 on a monthly time scale), but indeed appropriate to assess the genesis of events.

While the paper is generally easy to follow, the paper appears to suffer from weakly

formulated research question and consequently from a limited scope of the study. The results remain superficial and do not provide sufficiently new insights in low flow generation in Switzerland. I therefore cannot recommend the paper for publication in its present form. I will provide detailed feedback below, which I hope to be useful for the authors for further elaborating the paper.

**Science question**

Science questions (or hypotheses) of this paper (Line 73-75) are formulated in a way that everybody would immediately agree: There is little doubt that low flows will typically occur after anomalous weather conditions, and that most extreme low flows will be associated with the most extreme weather conditions. This leads directly into a quite superficial analysis and weak conclusions. I urge the authors to sharpen the science questions and, accordingly, the study design, in order to gain more significant insights in how precipitation and evaporation together generate low flow events in Switzerland. I agree with Referee 2 that the focus of the paper should be much more on the interplay of the two meteorological drivers. And their relative importance for events with different time of occurrence within the summer/fall low flow season.

**Methods**

The paper also suffers from weakly defined analyses. The methods section does not provide all necessary methodological details; they pop-up in a mixed results and discussion section. This makes analysis rather ad-hoc and hampers a well-structured assessment of the research question. I strongly advocate organizing the paper into clearly separated methods, results and discussion sections to foster a transparent, in-depth assessment.

In the following I review the used methods found in the results section.

In Section 3.2, the purpose of this "first correlation analysis" is not clearly defined (ref. also to the vague section title). The section assesses the correlation of 30-

day-anomalies. For what purpose the time window has been chosen, and what may analysing 30 days before the event tell us has not been indicated.

Section 3.4 Duration of climatic anomalies – The analysis of durations of anomalies before the peak of the event is largely depending on very short interruptions of the climate anomaly that have no effect on streamflows. Some pooling would be necessary to filter out disturbances in this type of analysis. The second analysis based on various time windows is more robust, and the most insightful analysis of the study.

Section 3.5 The role of winter precipitation is not a pertinent research question, it is well-known that in an Alpine environment it is rather snow-storage than accumulated precipitation that shapes summer low flow with respect to timing and magnitude. Analysing winter precipitation (instead of snow storage and snow melt) has not the potential to lead new insights in the low flow generation process in Switzerland.

**Specific comments:**

L 41: It is not either climate, or catchment, but the combined effect of meteorological drivers and catchment functioning that determines streamflow. L68: Contradiction to "the effects of evapotranspiration on low-flow occurrence and magnitude have received relatively little study" (two paragraphs above)

L72 Sentence does not make much sense.

L73 ff: Please revise hypothesis (better formulate them as science question(s) and objectives of the study. Avoid duplication of the overall aim into one objective (currently objective a).

L114: One sentence methodology, apart from the definition of the anomaly measures, is definitely too short.

Section 3.1: This is prior knowledge and should go into the introduction

L222: Statement is not true. What the cited papers say is that large parts of Europe

were affected by the drought events of 2003 and 2015. But papers also show how different timing and magnitude of events were across Europe.

L226: ditto

L239: Citation needed. What do you mean by erratic?

L285: No, snowpack is not the same as precipitation sum, snowmelt is precipitation redistributed over time.

L300 ff: "if SWE is important, we expect to see stronger correlations between winter precipitation and summer low flows at higher elevations – see my previous comment (L222). The following analyses are wrongly motivated and results overinterpreted.

L327: Remove sentence, as the paper does not represent a novel methodological framework

**References:**

Fangmann, A. and Haberlandt, U.: Statistical approaches for assessment of climate change impacts on low flows: temporal aspects, Hydrology and Earth System Sciences 23 (2019), 23, 447–463, 2019.

---

## Author Comment (AC2) · 4 Nov 2019

**We thank Referee 2 for the constructive feedback. We appreciate the suggested improvements and will address them in our revised version. Below we list our response (in bold) to the reviewer's comment** *(in italic).*

*In this work, the authors assess how anomalies in precipitation and potential evapotranspiration shape occurrence and magnitude of annual low flows for 380 Swiss catchments comparing preceding precipitation and evapotranspiration for different periods to the annual minimum flow. After an initial analysis of annual low flows, the authors decided to focus on summer low flows for the rest of the analysis. The paper is clearly outlined and easy to read. I agree with reviewer 1 that I particularly like Figure 6 illus-*

*trating the increasing role of PET during development of extreme low flows. However, different to reviewer 1 I have some major comments that I find important to address before publishing:*

**Thank you. Below we address the comments point-by-point.**

*Major comments:*
*Summer and winter low flows*
*After an initial analysis of annual low flows, the authors decided to focus on summer low flows for the rest of the analysis. This analysis takes relatively much of the full paper both in text and in Figures (1, 2 and 4). While I agree that it is important to differ between summer and winter low flows particularly when analyzing the drivers, I do not see that the results of general occurrence are new (e.g. Smakhtin 2001 and references therein, Fiala et al. 2010 and basically all runoff regime literature for Switzerland in particular e.g., Weingartner and Aschwanden, 1992) nor that they deserve this weight in the article. I suggest to minimize this to the introduction refereeing to the relevant references and remove Figure 1 and focus in Figure 2 on only the summer events or split in to summer and winter low flows at the beginning and then assess both for winter how snow (or precipitation and temperature) shapes low flow occurrence and magnitude and for summer how precipitation and evapotranspiration shape low flow occurrence and magnitude*

**We will improve the introduction of summer vs. winter low flows and include further literature as outlined by Referee 2. While we agree that there are contributions pointing out the general occurrence of winter and summer low flows, we consider it valuable to show that the occurrence of summer vs. winter low flows can be related to elevation (1200m, Fig. 1) and that the differences of summer vs. winter low flows are also detectable when analyzing the climate anomalies (Fig. 2a and b). We furthermore use Fig. 2a and b to argue why we relate summer low**

**flows to precipitation / PET anomalies, and expect these relations to hold for a wide range of low-elevation catchments**

*More focus on the shaping*
*Instead of counting and presenting summer and winter low flows representing the annual low flow, I think it would be more interesting to try to add on the shaping of the low flows caused by precipitation and evapotranspiration. For instance, in addition to correlation between precipitation and PET separately on extreme or less extreme low flows it would be interesting to look at their combined effect. And then how much of the combined effect could be attributed to precipitation and to PET. This would allow a better relative quantification and really add to the values of this study.*

**We emphasize that Fig. 3 and Fig. 6 are studying the combined effects of P and PET on low flows already. In the revision we will try to make the partitioning of the effects of the two drivers more explicit, for example by multivariate regression between the low-flow magnitude (anomaly) and the two climatic driver anomalies. We agree with the reviewer that it is interesting to quantify the relative roles of P and PET for low flow generation.**

*Choice of summer months*
*The authors focus at extreme low flows and the preceding conditions and chose 2003, 2011, 2015 and 2018 as the relevant years. The drought in 2011 was finished for most catchments before July, I urge the authors to look into the data and if so, adjust the analysis by either treating 2011 differently, i.e. not considering it a summer but a spring low flow and or change the analysis period for all years that was defined by the authors Jul-Nov.*

**We agree that the drought in 2011 is predominantly occurring in spring throughout the Swiss catchments (i.e. only 143 of 380 catchments have their low flows in**

July through November in 2011). We choose the 6-month period July-November because most low flows occur during this period. However, in some cases low flows do occur outside of this window (e.g., 2011). When we calculate the statistics of the main figures that use the 6 month window (e.g. Fig. 2 cd, 3, etc.) the results do not change significantly because our chosen period captures almost all low flows. In the revised version we will more clearly acknowledge that not all low flows are captured by this window, but also that this choice does not affect the results significantly.

*Terminology*
*There is a seamless transition from "drought" to "low flow" and mixed use of "droughts" and "deficits" while the citations support both. I find this mix critical since already deficit (even when regularly/seasonally occurring) causes low flows but meteorological droughts are larger deficits than normal, i.e. the regularly/seasonally occurring deficit. Please, revise the introduction to distinguish clearly between these. This would help the reader in the analysis that follows.*

In the revised version we will better distinguish between drought, deficit and low flow. In short, we will minimize the use of the term (and references to) drought, since low flows are not necessarily droughts.

*Likewise, the authors use often only the term "low flow" when actually referring to "extreme low flow". This can result in wrong statements (e.g. L72 "low flows are exceptional flow conditions" or L186 "triples the chance of an annual low flow"). And I would ask the authors to revise and correct the usage throughout the manuscript.*

In the revision we will be clearer when we refer to a (typical) low flow versus an extreme low flow. In addition, we will be clearer what we mean when we use

"extreme", since not being explicit about this can lead to misinterpretation (e.g. L72).

*Mixed results and discussion section*
*In my opinion results and discussion should be separated. This allows to focus on the results. Only then what we can learn from the results and where we might to be a bit more careful, then also relate and compare to what was done elsewhere and where limitations and possibilities lay. In the present form the manuscript mixes these aspects and is more difficult to search for s specific result /argument this way.*

**We understand that separating discussion and results may in many cases be a good choice. However, we tried both options in preparing the manuscript and found the chosen option to work best, because it is easiest for the reader to see the connection between the results and their interpretation. In the revision, we will go through the entire manuscript to ensure it is as clear as possible.**

*Minor comments:*
*L25-27 Remove this sentence for the abstract*

**This sentence may sound trivial but is needed for the logical flow of argument in the sentence following it. Therefore, we prefer to keep this sentence. We will consider reformulating the abstract to improve its clarity.**

*L40/41 is "landscape" only including surface features? Maybe use rather catchment properties".*

**We use landscape because we thought it would make clear that it does not include all catchment properties (such as its climatic conditions). Landscapes**

extend into the subsurface; this is implied in our statement in line 40, but we will now explicitly add this in the revision.

*L72 this statement is not correct they occur every year. Make clear that it is about extreme low flows here!*

**In the revision we will choose more precise wording which reflects that the low flows we are studying are "annual extremes" and not every annual extreme is necessarily an extreme in the long-term record.**

*L92-94 a summary table (maybe only in the supplementary material) would be helpful*

**We will include a table in the supplementary material; the data for all catchments will be made available through the "open access" platform of the ETH library.**

*L121 1200m asl, why this threshold? Why not a range?*

**We use a threshold to split the dataset into two groups (below 1200m asl and above 1200m asl). This threshold accurately reflects what type of low flow (seasonality) is expected within this dataset. We do not see how using a range would improve this observation.**

*L130 remove "However, this remains to be tested".*

**OK.**

*L147-150 For these low flows, people are usually prepared. Here it would be interesting how much more extreme are others. If it was due to a lack of precipitation, the signal should be visible in spring melt.*

We agree that such winter precipitation deficits can have effects on flows later in the snowmelt season, and will likely be visible in the data. However, the aim of this paper was not to explore all these hydrological connections, but rather focus only on the climatic conditions leading to the lowest flow in the year. This remains an interesting suggestion for further research.

*L154 "suggesting. . ." could be also formulated that lower elevation Swiss catchments could be representative sample for global summer low flow? (and maybe not even global but for humid regions with seasons?); This part would better fit in the introduction or methodology/catchment section.*

**We put this statement here because we discuss our results and their implication here. Making this statement in the introduction is leapfrogging ahead, because we have not characterized the seasonality of Swiss low flows at that stage.**

*L161 altitudinal variation in 30-day anomalies: could that be influenced by catchment size? A large catchment might not react on such an anomaly a small catchment not anymore if the driving anomaly is at the beginning of the period.*

**We tested if catchment size affected the altitudinal variation. While such effects can be expected, no clear signal was found, probably because the catchments are relatively small (< 519km$^2$, with a median of 74km$^2$). We will discuss this in the revised paper.**

*L174 "substantial site to site variability" can this be quantified?*

**To clarify what this variability refers to, we now explicitly refer to Fig. 2c and 2d for the reader to look at the spread. We also add the range in the text.**

*L181 can these 8L189-196 These results are not surprising (the authors refer even to studies that found the same) but nicely illustrated and supported by the data. However, it would add to the value of this study to quantify the contribution of precipitation and evapotranspiration.*

**We agree that these results are maybe not surprising, as they have been shown for individual cases (as referenced earlier in the paper). Our work improves past studies by (a) providing a large dataset which shows the variability and consistency in low flow-climate relations among basins; (b) quantifying the effect of duration of the climatic anomalies required to generate the extreme low flow events; and (c) separating the effects of precipitation and PET. We believe this provides a more robust picture of otherwise intuitive relations.**

*L189-196 Consider also to compare to Stahl et al. 2010*

**We will put our results in context of the findings of Stahl et al. 2010.**

*L196-197 delete sentence*

**It is unclear to us why this sentence should be deleted.**

*L203-204 it is possible to avoid that by the study design (see also my major comment on seasonal split)*

**We agree that it is possible to select data such that only the drier years are kept. However, that is not the purpose of our study. We rather discuss that not all annual low flows are created equal.**

[Figure]

*L222-228 that depends on how one looks at the drought: is the same scale as in the references used? The effect was also found in low flows but maybe not in the metric "annual low flow", again distinguish between drought and low flows (see also major comment on terminology above) Figure 4: Looking at the figure makes me wonder how/if the regulated catchments might influence the pattern presented. Could that be picked up in the discussion?*

**We will change this in the text, to clearly distinguish between "droughts" and "low flows". We will discuss the influence of flow regulation on low flow timing in the revised manuscript.**

*L241 How brief since the study is about anomalies?*

**We will remove this.**

*Technical comments*

*While I find that active voice generally a good choice, I would avoid starting every sentence with "we" (e.g. 2.1 but also elsewhere), please revise.*

**We will consider how to reduce the prevalence of sentences beginning with "we" (although they are usually the most compact, clear and direct way of expressing things).**

*L22 "dry years saw" please rephrase*

**We will change this.**

*L28 redundant, delete either "could" or "potentially"*

**We will delete "could".**

*L44 "(PET)" -> ", PET"*

**We will change this.**

*L44 remove "should" and "usually"*

**We will delete "should" and "usually".**

*L46 remove "made"; split sentence: ": : :to a catchment. Hence a sustained: : :"*

**We will change it accordingly.**

*L49 Make a new paragraph*

**Ok.**

*L64 "smaller" = "lower"?*

**We will change this.**

*L66 "comes" -> "occurs"; remove "the" before summer and before winter*

**We will change this.**

*L79 "useful" = "suitable"?*

**We will change this.**

*L94 "quantified" -> "estimated"*

**Ok.**

*L100 "years whose lowest annual flows were much" -> "years with lowest annual flows much"*

**We will change this.**

*L103 "low flows" -> "extreme low flows"*

**We will change this.**

*L224 remove "the" before summer and before winter*

**We will change this.**

*L238 "more strongly" -> "higher" (also in L244)*

**We will change this.**

*L259-262 rephrase to make more concise*

**We will rephrase this sentence.**

---

## Author Comment (AC3) · 4 Nov 2019

**We thank Anonymous Referee 4 for the feedback. Below we list our response response (in bold) to the reviewer's comment** *(in italic)*.

*In this paper, the authors assess to what extend precipitation and PET anomalies trigger summer low flow events in Switzerland. The assessment employs Spearman correlations between low flow magnitude and climate anomalies in P and PET aggregated over varying lead times before the peak of the low flow event. The correlations are overall weak, but still indicate that most low flows arise from the compound effects of precipitation and PET anomalies, with longer and larger anomalies related to more extreme low flow events, as one would expect. The assessment of lead times before the*

[Figure]

*peak of the event is not new (e.g. Fangmann and Haberlandt, 2019 on a monthly time scale), but indeed appropriate to assess the genesis of events.*

**Indeed, drivers of low flows have been studied before (as reflected by the citations, including Fangmann and Haberlandt, 2019), and the result that both PET and P are important for low flows may not be a big surprise. However, the paper provides insight into the durations, magnitudes, and timings of the anomalies that drive low flows, and how these vary across hundreds of catchments situated in diverse landscape conditions. These more detailed insights about low-flow generation reveal aspects that cannot just be derived from intuition. In addition, the aspects of low flows that we discuss are not captured by previous studies, because they study other regions, and/or they study different aspects of low flows. Therefore, we believe that the provided results (and data) may be useful for the hydrological community.**

*While the paper is generally easy to follow, the paper appears to suffer from weakly formulated research question and consequently from a limited scope of the study. The results remain superficial and do not provide sufficiently new insights in low flow generation in Switzerland. I therefore cannot recommend the paper for publication in its present form. I will provide detailed feedback below, which I hope to be useful for the authors for further elaborating the paper.*

**In the revised version we will sharpen the research questions, to make them more specific and clearer. Since the detailed comments on this issue are discussed below, we refer to our detailed responses there on how this will be done.**

*Weakly formulated research questions and and consequently from a limited scope of the study. The results remain superficial and do not provide sufficiently new insights*

*in low flow generation in Switzerland. I therefore cannot recommend the paper for publication in its present form.*

**Below we respond to the detailed comments that refer to this concern.**

*Science question*
*Science questions (or hypotheses) of this paper (Line 73-75) are formulated in a way that everybody would immediately agree: There is little doubt that low flows will typically occur after anomalous weather conditions, and that most extreme low flows will be associated with the most extreme weather conditions. This leads directly into a quite superficial analysis and weak conclusions. I urge the authors to sharpen the science questions and, accordingly, the study design, in order to gain more significant insights in how precipitation and evaporation together generate low flow events in Switzerland. I agree with Referee 2 that the focus of the paper should be much more on the interplay of the two meteorological drivers. And their relative importance for events with different time of occurrence within the summer/fall low flow season.*

**We point out that lines 73-75 do not represent our science questions. However, we will rephrase the text to make this distinction between "introductory text" and "science questions" clearer.**
**In the revised version we will sharpen the actual science "questions" (listed in lines 81-85). For example: "We investigate (i) to what extent low flows are driven by precipitation anomalies, PET anomalies, or their combined effects, (ii) what magnitudes of climate anomalies are leading to low flows, (iii) what durations of climate anomalies are typical for low flows, (iv) how these climate anomalies vary across the Swiss landscape, (v) how these climate anomalies vary with the severity of the low flow event."**
**None of the above questions is answered in previous studies for our study re-**

**gion. In addition, we will update all of the results and discussion paragraphs to better present the results and their implications.**

*Methods*
*The paper also suffers from weakly defined analyses. The methods section does not provide all necessary methodological details; they pop-up in a mixed results and discussion section. This makes analysis rather ad-hoc and hampers a well-structured assessment of the research question. I strongly advocate organizing the paper into clearly separated methods, results and discussion sections to foster a transparent, in-depth assessment.*

**Apologies for the confusion. In the revised version we will ensure all methodological aspects are already explicitly mentioned in the methods section.**

*In the following I review the used methods found in the results section.*
*In Section 3.2, the purpose of this "first correlation analysis" is not clearly defined (ref. also to the vague section title). The section assesses the correlation of 30-day-anomalies. For what purpose the time window has been chosen, and what may analysing 30 days before the event tell us has not been indicated.*

**The purpose of section 3.2 is to reveal what magnitude of climate anomalies are typical for low flows, how this varies between P and PET anomalies, how this varies with elevation, and whether P or PET appears to be more important. In the revised version we will introduce these purposes more clearly, both in the methods section, and the results section. We will also try to provide a more quantitative perspective on the P and PET partitioning.**
**The purpose of choosing a 30-day window is to reflect that low flows are generated during a prolonged period of anomalous climate. We show the results for**

[Figure]

**30 days, but emphasize that other time-windows (from 1 week to 120 days) yield broadly consistent results. We could provide such in supplementary materials. We choose 30 days as the result to present, because 30 days (as later shown) is the time window which explains most typical low flows (Section 3.4). These changes will also lead to updates in the text of manuscript that address these additional analyses and explanations.**

*Section 3.4 Duration of climatic anomalies – The analysis of durations of anomalies before the peak of the event is largely depending on very short interruptions of the climate anomaly that have no effect on streamflows. Some pooling would be necessary to filter out disturbances in this type of analysis. The second analysis based on various time windows is more robust, and the most insightful analysis of the study.*

**We acknowledge that short (irrelevant) interruptions may affect the determined length of a climate anomaly. To address this we do multiple things. First, we use a 10 day moving average of time series to filter out short duration interruptions. Second, for the revised version, we plan to calculate these results also using other time windows to test their sensitivity. Third, these limitations and the sensitivity of the results will be discussed in the revised version.**

*Section 3.5 The role of winter precipitation is not a pertinent research question, it is well-known that in an Alpine environment it is rather snow-storage than accumulated precipitation that shapes summer low flow with respect to timing and magnitude. Analysing winter precipitation (instead of snow storage and snow melt) has not the potential to lead new insights in the low flow generation process in Switzerland.*

**As pointed out by previous reviews, winter precipitation is not always a robust proxy for winter snowpacks. We will change our discussion of the role of winter**

**precipitation for summer low flows accordingly. We would like to emphasize that it remains largely unquantified how winter conditions (either snow specifically, or both snow and winter rain) affect low flows across the Alps. This is clearly important information as we all agree that in many Alpine landscapes winter conditions can shape summer low flows. In addition, in the revision, we will quantify the effect of solid vs liquid precipitation (e.g. by using a temperature threshold) and discuss if this better explains the low flow behaviors.**

*Specific comments:*
*L 41: It is not either climate, or catchment, but the combined effect of meteorological drivers and catchment functioning that determines streamflow.*

**We obviously agree that both climate and the catchment itself shape low flows (as we tried to convey in the original text). We will see how we can rephrase this to avoid confusion.**

*L68: Contradiction to "the effects of evapotranspiration on low-flow occurrence and magnitude have received relatively little study" (two paragraphs above).*

**In the revised version we will rephrase this to make clear what aspects of ET have not received much attention, rather than to make the generic statement we currently have.**

*L72 Sentence does not make much sense.*

**In this sentence we aim to explain why focusing on climate anomalies makes sense. We will consider how to rephrase to avoid confusion.**

*L73 ff: Please revise hypothesis (better formulate them as science question(s) and objectives of the study. Avoid duplication of the overall aim into one objective (currently objective a).*

**As stated earlier, this is not the hypothesis we test in the paper. We now realize that this confusion can arise (probably because we used the word "hypothesize" in this sentence). We will reformulate this statement to reduce the chance of this confusion.**

*L114: One sentence methodology, apart from the definition of the anomaly measures, is definitely too short.*

**In the revised paper we will add some text that explains the rationale of this analysis.**

*Section 3.1: This is prior knowledge and should go into the introduction*

**We agree that some of the aspects in Section 3.1 can already be stated in the introduction. We, however, like to still repeat some of these aspects to put the Swiss results into context of other studies. We also emphasize that the presented results in section 3.1 (e.g. the 1200m split) are not part of existing literature and should therefore not be presented in the introduction.**

*L222: Statement is not true. What the cited papers say is that large parts of Europe were affected by the drought events of 2003 and 2015. But papers also show how different timing and magnitude of events were across Europe.*

We now realize that we oversimplified the spatial coherence reported in previous studies. We change the interpretation of our results accordingly, by not stating that Switzerland is necessarily in contrast with other regions of Europe. However, we would like to point out that the spatial gradients in low flow timing in Switzerland appear stronger than in some other parts of Europe. We will reformulate to clarify this in the revised manuscript.

*L226: ditto*

**See response to previous point.**

*L239: Citation needed. What do you mean by erratic?*

**By "erratic", we mean that daily precipitation is more irregular in time (compared to PET). We now use the word "irregular" to be clearer. We are unsure where a citation is needed in line 239?**

*L285: No, snowpack is not the same as precipitation sum, snowmelt is precipitation redistributed over time.*

**We forgot to add an additional line of logic in our statement that connects winter precipitation as a (weak) proxy for snow for our study region. We will add this to avoid confusion. In addition, we will be careful with stating implications of the results as we earlier discussed that winter P and snow are not identical.**

*L300 ff: "if SWE is important, we expect to see stronger correlations between winter precipitation and summer low flows at higher elevations – see my previous comment (L222). The following analyses are wrongly motivated and results overinterpreted.*

**We will revise the text to reflect that winter P and snow (pack/fall) are not identical.**

*L327: Remove sentence, as the paper does not represent a novel methodological Framework*

**We agree that no real framework is provided, and will change this statement accordingly.**

---

## Author Comment (AC4) · 4 Nov 2019

**We thank Anonymous Referee 3 for the detailed, constructive feedback. We appreciate the suggested corrections and will address them in our revised version. Below we list our response (in bold) to the reviewer's comment** *(in italic).*

*The objective of presented study is to investigate how precipitation (both summer and winter) and PET anomalies influence low flows across Switzerland both in typical and exceptionally dry years. In my opinion, authors provided detailed and important insight into climatic drivers controlling low flows based on data assessment from 380 catchments in Switzerland. In general, I found the results interesting, although the methods used are not novel. I found the main contribution in assessing a large number of catch-*

[Figure]

*ments which may help us to better understand why catchments sometimes behaves differently, which are main controls and thus what may happen in the future in a warming climate. Thanks to a large number of catchments covering different elevations, I think the results can by generalized to other regions, at least to those located in similar climates. In this respect, the results have an international value and may be very useful for hydrological community. Therefore, the results are important and certainly appropriate for HESS. However, I have some comments listed below, which need to be addressed before I can recommend the manuscript for publication. These comments are mainly related to methods and results interpretation. I hope that these comments will help authors to improve the manuscript.*

**Thank you.**

*Major comments:*
*Authors used winter precipitation to show how winter and snow conditions are important for summer low flows. Although this is an important aspect especially for higher elevation catchments, I am not sure to which degree authors were able to capture the snow effect by selecting just winter precipitation as a single variable. The winter precipitation does not tell us whether the precipitation is falling as rain or snow. This is, in my opinion, very important since snow contributes to runoff much later than rain and thus influence the seasonality of groundwater recharge and potentially summer low flows. Therefore, I am not sure whether the winter precipitation could correctly capture this issue well enough to make any general conclusion. Using some snow-related metrics (snowfall fraction, snowfall water equivalent, annual maximum SWE or similar) would be perhaps better to show whether there is (or is not) any relation. Therefore, I would be careful with interpretation going towards the role of snow. I do not see much evidence in authors results to make some conclusion, although several previous studies quantified this effect at different elevations.*

**We agree that winter precipitation is not an ideal proxy for snow. In the revised version we will more explicitly acknowledge that winter precipitation does not fully represent snow. In addition we will quantify the effect of solid vs liquid precipitation (e.g. by using a temperature threshold) and discuss if this better explains the low flow behaviors.**

*I am not fully convinced that assessing both winter and summer low flows is a good approach since the meteorological drivers are different for both of low flows types (see e.g. Harpold et al. (2017) for general overview). I found the mixing of both types throughout the manuscript sometimes a bit confusing. Authors did first analysis (Fig. 1 and Fig. 2) using annual low flows from all catchments (regardless whether they were summer or winter) and later they decided to further analyse only summer low flows. Although I would maybe prefer to focus only on summer low flows in the study, I accept the authors' decision to make first some results related to both winter and summer low flows together and later focus just on summer low flows. However, I am a bit confused how authors exactly proceeded to select catchments and years for summer low flows analysis (but maybe I only missed something). First, it seems that authors analysed all seasonal low flows occurred in the warm period for all study years (even in case that annual low flow occurred in winter). However, later (L306-307) it seems that authors completely excluded catchments/years in case that annual low flow occurred during winter. The latter approach could result in excluding many of the highest elevation catchments from the analysis (and thus it might lead to the conclusion that winter precipitation is not an important signature to influence summer low flows as noted in the previous comment). Therefore, please clarify how you proceeded. I would think that the first mentioned approach is more appropriate and should be used in the analysis (especially in case you are focusing on the role of snow or winter precipitation in addition to role of previous precipitation and PET).*

**In the revised version we will more clearly state what low flows are used to pro-**

**duce the results. We believe it is valuable to show both winter and summer low flows, because these are the actual lowest flows that occur in these catchments. Since summer and winter low flows are indeed generated by different drivers, we have to sometimes use a subset of all low flows to do meaningful analyses. In short, when analyzing summer low flows, we selected all low flows that occurred in July through November. We will better emphasize when and why we make this selection choice.**

*Regarding to the comment above, I think that mixing the summer and winter low flows in Fig. 2 (top panels) is not a suitable approach since the climatic controls are different for both type of low flows. As it is now, you are losing a lot of information, especially in higher elevation catchments, because you are trying to describe (mostly) winter low flows in these catchments using variables, which are not much relevant. Therefore, I would suggest to make the Fig. 2 just for June (or July) to November low flows. Then you would see, whether the precipitation and PET are important drivers for summer low flows even at highest elevations or whether the figure would suggest that there might by also something else (e.g. snow from preceding winter). In case you decide to keep Fig. 2 as is, please consider to split it into two figure, since the mixing of annual and summer low flows in one figure (top panels vs. bottom panels) is, in my opinion, confusing.*

**The purpose of Fig. 2a and 2b is to show that the importance of P and PET as drivers for annual lowest flows systematically changes with elevation. This is in our opinion a useful result that we also want to show in the revised version of this manuscript. We now realize that using a subset of these data for Fig. 2c and 2d may confuse the reader. Therefore, we will follow your suggestion of splitting the figure into two separate figures.**

*Authors calculated preceding PET as one of the main climate drivers. However, physi-*

*cally correct way is to use actual evapotranspiration (AET) instead of PET. I am aware, that calculating AET would not be such easy. Nevertheless, the relation between PET and AET is not always straightforward since higher PET do not necessarily mean higher AET (especially in lower elevation catchments with lower precipitation, higher water demand and thus lower water availability). Therefore, I would appreciate more discussion related to PET vs. AET interactions.*

**Obviously, we agree that AET is the physical process by which water leaves the catchment, that may lead to low flows (and could thus be considered a driver). However, the purpose of our paper is to infer the climatic drivers of low flows. AET is not a climate driver of low flow, it is the outcome of how climate interacts with the soil and vegetation in the catchment. Therefore, we choose (high) PET as a driver because PET is the climatic condition that drives AET (and subsequently low flows). In the revised version we will better emphasize our choice of PET as a driver over AET as a driver, and we discuss its limitations especially regarding the complementary relationship between AET and PET. Using AET would furthermore require an additional soil water balance model which adds uncertainty to the analysis.**

*Minor comments:*
*One of the conclusions is that low flows are controlled by either low precipitation or high PET or combination of both. This is not surprising since there are not many other options (at least for summer low flows in near-natural catchments). Therefore, I would rather highlight implications which arose from results, but which are not such trivial. In this respect, I would recommend to slightly reformulate the respective part of abstract, short summary and perhaps also hypotheses (line 75) to better highlight the novelty of your work.*

**Indeed, it is no surprise that PET and P drive most low flows. However, the**

purpose of our manuscript is to show to what extent, and which characteristics of P and PET drive low flows, and how these vary spatially. These more detailed pictures of low flow drivers are nontrivial and we will try to better express their value in the revised paper.

*Authors used winter precipitation, but this signature is not mentioned and explained in the methods section.*

In the revised version, we will add the description of how we obtained winter precipitation also in the methods (rather than just in the later stages of the paper).

*I suggest to create a map showing the location of catchments. I think that just a simple map of Switzerland with shaded DEM with points showing the position of catchments outlets would help those readers not familiar with Swiss hydrology. Maybe, catchment points might be coloured according to catchment elevation (or something similar).*

We can add such a map. We will defer to the editor's advice on whether such a map is best included in the main paper or the supplementary material.

*It is a bit questionable to describe the previous precipitation just using the sum of precipitation from the defined preceding period. The reason is that the importance of precipitation for the low flow at the specific date changes when going back in time (precipitation closer to the day with the low flow is more important than that occurred earlier). Did you also consider applying some kind of continuous precipitation index, e.g. current precipitation index CPI (Smakhtin and Masse, 2000)? I would appreciate more discussion on this issue.*

We choose a time-window to reflect that low flows are typically not generated instantly, but are generated over longer time spans. We agree that precipitation during times closer to the actual low flow will probably often impact the flow more than precipitation during a longer time prior to a low flow. Alternative metrics such as CPI may account for this fact (to some extent) and their merit will therefore be discussed in the revised paper. However, precipitation in the periods immediately preceding low flows often does not significantly refill groundwater stores, and thus may have very little impact on the low flows themselves (they only result in a short peak in the hydrograph). Such effects are not captured by CPI.

*L101: Please, add a brief information why you remove "unusually high annual low flows". I see the point, but it would be good to clarify it.*

**We will explain this in more detail in the revised version of the manuscript.**

*L147-149: Could you please add some references regarding this statement? Just to avoid speculations, since you cannot prove this based on your results.*

**We will add the appropriate references here.**

*Fig. 2: Here it is nicely shown that PET anomalies are relatively less important compared to precipitation anomalies (inter-annual variations up to 40 mm for PET, but up to -200 mm for precipitation).*

**Thank you.**

*L201-202: Maybe I missed something, but I do not see the described effect (wet years) from Fig. 3b*

**Wet years refers to years with higher low flows (Fig. 3b). We now realize this may be unclear to the reader and therefore we will make this clearer in the revised manuscript.**

*Fig. 3a: Why there are still some years when low flows are preceded by above-average precipitation and below-average PET (bottom-right quadrant)? Would the figure looks similar also for other than 30 days time-windows (e.g. 60 and longer)? You suggested some explanation in lines 201-205, but could you be more specific?*

**We will extend our description of why it makes sense that above-average P and below average PET anomalies are observed when (i) the seasonality of the flow regime outweighs the effects of shorter-term weather, and (ii) when very wet years occur (with high low flows).**

*Fig. 4: This figure shows both summer and winter low flows. However, earlier you stated that only summer low flows are analysed starting from Fig. 3 (L155-156). Therefore, could consider putting this figure as a Fig. 3 to be consistent. Additionally, I would maybe change the colour scale by using "cold" colours for cold months and "warm" colours for warm months.*

**As stated before, in the revised version we will better emphasize when and why we use summer vs. all low flows in various parts of the paper. We will evaluate if changing the color scheme makes the figure clearer.**

*L237: Please, specify the thresholds for "below-threshold" precipitation and "above-threshold" PET (maybe in methods as well).*

**We specified these thresholds in lines 251-252. In the revision we will also include this description in the main text (around line 237).**

*L286-287: This hypothesis would be correct only in case that most of winter precipitation would occur as snow. Therefore, not higher winter precipitation in general, but higher snowfall (snow storages) should lead to larger and later summer low flows (but only at high elevations with high snow storages). Please, consider reformulation. In this respect, you correctly pointed to the fact that winter precipitation sums do not accurately represents SWE (L299).*

**We refer to our earlier more detailed comment on how we address this limitation.**

*L325. I would maybe add "winter" to describe the precipitation in California. In contrast to humid catchments in Central Europe, the previous summer droughts in California were mostly driven by lack of winter precipitation (and snowpack).*

**We will add "winter".**

*Technical corrections*
*L95: Please, add the reference to RhiresD and TabsD products*

**We will add a Meteoswiss reference.**

*L106: I would not use the term "long-term" for time period of 18 years. Instead, I would directly specify from which time period the average has been calculated.*

**We will change this.**

*L142: I think you wanted to refer to Fig. 2ab*

**Thank you, we will change this.**

*Figure captions: A lot of text in figure captions (Fig. 2, 3, 5 and 6) is related to figure interpretation rather than figure description. In my opinion, these parts would better fit directly to the main text*

**We tried to include the main message of the figure in every caption. We believe this is informative to the reader.**

*L177: "Our previous results ...". I would remove "previous" since it implies something you did in some previous study. Alternatively, be more specific instead (e.g. "results shown in figure/section no. ...").*

**We will change this.**

*Fig. 6: Please, consider larger axis captions to increase the readability.*

**OK.**

---

## Short Comment (SC1) · 10 Nov 2019

This review was prepared as part of graduate program course work at Wageningen University, and has been produced under my supervision. The review, in a formal review format as requested by the course, has been posted because of its good quality, and likely usefulness to the authors and editor. This review was not solicited by the journal.

Please also note the supplement to this comment:
https://www.hydrol-earth-syst-sci-discuss.net/hess-2019-448/hess-2019-448-SC1-supplement.pdf

[Figure]

[Figure]

**Supplement:**

**Review on: The effect of climatic anomalies on low flows in Switzerland**

Authors of the reviewed paper:
M. G. Floriancic, W. R. Berghuijs,
J. W. Kirchner, P. Molnar
Journal: Hydrology and Earth System Sciences (HESS)

Reviewed by: Robert Lubben

For: Interdisciplinary Topics in earth and environment
Lecturer: Ryan Teuling

Date: 24-10-2019

**Short summary:**

The reviewed paper is about the effects of climatic anomalies on low flow occurrence in 380 swiss catchments for the period 2000-2018. The low flows are defined by the annual 7-day lowest flows. The anomalies in precipitation and evapotranspiration are calculated for several time periods before the annual low flow occurrence (7 up to 182 days). With this data two hypotheses are tested 1) low flow occurs after anomalous weather conditions and 2) that the most extreme flows will be associated with the most extreme weather anomalies. The results of the study are that the low flows mostly occur after anomalous precipitation and evapotranspiration events. Most of the low flows (92%) are influenced by below average precipitation and 70% is influenced by above average evapotranspiration. Also, the extreme weather anomalies, or meteorological droughts, tend to generate extremely low flows. Winter precipitation, as SWE, was less important for the low flow seasonality than the climatic anomalies in this study.

**General comments:**

Overall, this paper is well written and of high quality. No textual errors could be found. The two hypotheses and their origins are stated well. The need for having an answer on the hypotheses has a good relation with former research. The conclusions clearly give an answer to the hypotheses and the discussion about the effects of SWE on low flow seasonality is helpful for placing the results of this study in context and linking this study to former research. The figures of the results help to give a better understanding and can easily be linked with the corresponding text.

The overall structure of the paper really shows the reader how the research is conducted and which methods are used. The methodology clear and can (almost) be reproduced with the given description. However, the method for estimating potential evapotranspiration needs a bit more clarification, see major comments. The data that has been used for the analysis is of high quality and has a high spatial density. The estimation of low flows is a good method that is widely used in low flow analysis and it identifies the occurrence of a low flow in such a way that errors in measurements and can be filtered out (Smakhtin, 2000). The precipitation and evapotranspiration anomaly quantification for different durations is a good way of finding anomalies in the climatic data if a sufficient amount of years is used. The study is novel because, this type of low flow analysis is not carried out before on this scale with this many catchments. Although, the used method is not exceptional or novel, it gives a good view on the separate and combined effect of precipitation and evapotranspiration on low flow occurrence. Also, the used visualization method shows the effects of extreme evens well (like figure 6 in the manuscript). The visualization of enhanced PET under drought conditions is also nicely visualized. These figures help by understanding and supporting the text and give the reader a clear first glance at the study and the results.

Low flow seasonality is a relevant topic with the expected increase in weather anomalies by climate change. Understanding the climatic drivers and their impacts on low flow genesis can help understanding the processes leading to low flows and help in managing discharge. This becomes more and more relevant after the recent drought years. In my opinion, there are no mayor points that limit acceptation of this paper. However, there are a few major comments that could be useful to take into consideration.

Major comments:

There are three major comments that can be raised while reading this work. The first comment is about the exclusion of 2% of the low flows, this is not supported in the current context. Low flows in years with high precipitation will still be caused by anomalies in precipitation and evaporation. Therefore, the exclusion of the low flows that are above the value of three standard deviations from the mean seems not logical. In this way the method with which the low flows are estimated is in conflict with the actual used method where 2% of the data will be excluded because it is above 3 standard deviations. This means that the definition of a low flow has to be changed or that all data has to be used in this study (including the 2%). Including this data should not influence the results in a negative way because the drivers of low flow are likely the same. Another way to clarify this could be to do the analysis with including the 2% of low flows and then conclude that the abnormally high low flows are not significant or hinder the analysis. After the analysis the high low flows can be excluded with a good reason.

The second major comment involves the used potential evapotranspiration estimation method. This is only addressed very briefly via referencing to the paper of Hargreaves and Samani (1985). The used calibration parameter value and the source of solar radiation are not given in this way. The calibration value can influence the results for the Hargreaves PET estimation significantly. Especially in humid areas the PET can be overestimated when using Hargreaves (Trajkovic, 2007). A method that uses observed radiation can therefore result in less evaporation which influences the overall PET anomalies. The estimated PET can possibly be validated with the lysimeter used by (Seneviratne et al. 2012b). A clearer description of the used PET method by including the formula and the used values for the parameters will help by giving insight in the uncertainty of the PET estimation. This will also help making the methodology more clear and improve the possibility to apply this framework elsewhere.

The last major comment is on figure 4 (page 9) of the manuscript. In this figure the timing of annual low flows in Switzerland is shown. The text that refers to this figure states: *'Within the Swiss Plateau, low-flow timing is more spatially consistent during some (non-extreme-drought) years (e.g. 2009, 2013, 2016), than during others (e.g. 2000, 2002, 2004, 2010, 2017)'*.
However, there is no reason given for this difference for each year. Is this caused by SWE, other drivers of low flows or P and PET? If it is more related to P and PET it is useful to include this in the text to further clarify the contribution of these drivers on more local scale. Also, if it is caused by SWE the results of the paper of Jenicek et al. (2016) can be related to this in non-drought years. Therefore, I suggest to get a better understanding of the variability of streamflow in non-drought years. This can be done by looking more closely at the relation between SWE, PET and P on streamflow during these years. This can also help by putting the studied drivers (P and PET) in context to other drivers of low flow like anthropogenic activity.

Specific comments:

- In part 3.2 (line 140-142) of the manuscript the graph of figure 1 is used to explain the contribution of P and ET to low flow occurrence: *'However, again distinct regional differences exist: at low elevations, almost all annual low flows occur after periods of anomalously high potential evapotranspiration and anomalously low precipitation (Fig. 1a&b)'*. However in this figure only the occurrence of low flows per month related to the elevation level of the catchment is shown. In figure 2 the differences explained in the text of 3.2 are shown and therefore this reference should be changed to figure 2a,b.

- By implementation of this framework in another study area, can be stated that the PET estimation method maybe has to change depending on the climatic conditions of the new study area. It could be that they have to switch to radiation based methods (see major comment 2) depending on the local climate.

- The description for figure 2 and 3  is quite large and maybe can be shortened by putting more explanation in the text or by making the figures clearer with a main and sub-title. Especially, the part about the percentages of low flows that are caused by combinations of drivers (figure 3), is already mentioned in the text.

- The line in figure 2 seems higher than 1200 meter (even with the non-linear y-axis). The data seems to be more in agreement with Jenicek et al. (2016) on a separation between low and high elevation catchments around 1350 meter above mean sea level. This can also be a part of the discrepancy between Jenicek et al. (2016) and this manuscript on SWE relation to low flows.

- The reason for choosing the spearman correlation instead of for example the Pearson correlation is not given. This can easily be done by stating that the data is non-linear and the spearman correlation will result in a better fit with this data. See D. R. Legates & G. J. McCabe (1999).

- Link Seneviratne 2012a to the IPCC report does not work in the references

References:

Hargreaves, G. H. and Samani, Z. A.: Reference Crop Evapotranspiration from Temperature, Appl. Eng. Agric., 1(2), 96, doi:10.13031/2013.26773, 1985.

Jenicek, M., Seibert, J., Zappa, M., Staudinger, M. and Jonas, T.: Importance of maximum snow accumulation for summer low flows in humid catchments, Hydrol Earth Syst Sci, 20(2), 859–874, doi:10.5194/hess-20-859-2016, 2016.

Legates, D. R., & McCabe Jr, G. J. (1999). Evaluating the use of "goodness-of-fit" measures in hydrologic and hydroclimatic model validation. *Water resources research*, *35*(1), 233-241.

Smakhtin, V. U.: Low flow hydrology: a review, J. Hydrol., 240(3–4), 147–186, doi:10.1016/S0022-1694(00)00340-1, 2001.

Trajkovic, S. (2007). Hargreaves versus Penman-Monteith under humid conditions. *Journal of Irrigation and Drainage Engineering*, *133*(1), 38-42.

---

## Author Response (AR1)

**Dear Editor,**

**We hereby resubmit our revised manuscript entitled "Effects of climatic anomalies on low flows in Switzerland". We revised the manuscript according to the recommendations, we addressed all comments of the reviewers and the editor, and we hope that our revised version is now suitable for publication.**

**We thank you for the detailed explanation of your decision and addressed the points of concern. Below we list all editor and reviewer comments** *(in italic)* **and our answers (in bold).**

**Thank you for considering our revised manuscript, and we hope you now consider it suitable for publication. We appreciate your time and look forward to your response.**

**With best regards**
**Marius Floriancic**

**(on behalf of the co-authors Wouter R. Berghuijs, James W. Kirchner, Tobias Jonas, and Peter Molnar)**
* * *
*Editor Decision: Reconsider after major revisions (further review by editor and referees)*
*(26 Nov 2019) by Kerstin Stahl*

*Dear Marius and co-authors,*
*Thanks for your online-replies to the reviews. The manuscript has received differing reviews, recommending minor revision, two times major revision and one rejection. Based on the details of the reviews, I invite you to submit a substantially revised version of the manuscript for further consideration. The revised manuscript will then be reviewed again by at least two reviewers in order to decide further.*

*My editorial assessment based on the reviews in particular identified the following necessary major revisions:*

*1) Working out the sufficiently original contribution: Three reviews contain the phrase 'not surprising' when describing the results of the study. That itself does not hinder publication, but the added value of the findings has not been sufficiently worked out yet. R4 finds the hypotheses too weak, an assessment that is also reflected in R2 and illustrated by the summary written by Ryan Teuling's students. R2, R3 and R4 find the 'initial analysis' on seasonality particularly redundant with previous work, confusing, or at least not original enough to warrant the space it takes in the manuscript. This suggests that the necessary conclusion to continue with summer low flow may also have been based on a thorough review of existing work on low flow seasonality. In what way does the study go beyond previous work, e.g. the maps in HADES, the regime stability classification for the "Modulstufenkonzept" and all the underlying research? The reviewers do make some general suggestions, where to look for the added value, for example that "results [of merely measuring whether there is correlation] could benefit from an actual quantification of the relative contributions" or to "better use the added value of the large dataset to explain controls" or really take up the 'shaping' idea from the research questions. Please take the combination of these reviewer comments seriously into account in the revisions to prove progress by and original contribution of the study.*

**We now better emphasize the novelty of the work by the following main changes:**

**1) We removed Fig.1 and the results on "low-flow timing" and only report the timing based on existing literature in the introduction. We think that this change comes with two advantages. First, it will introduce the international HESS readership to the German literature that they probably are unaware of but that contains potentially interesting information. Second, it avoids that our analysis repeats the analysis of low-flow patterns that (mostly) also appears elsewhere in work by ourselves and others.**

**2) We now better explain in the introduction that previous studies have shown that low flows can be driven both by PET and P. In our opinion, the submitted revised manuscript provides additional deeper data-driven insight into the durations, magnitudes, and timings of the climatic anomalies that drive low flows, and how these vary across hundreds of catchments situated in diverse landscapes (topographies, soils, etc.). These more detailed insights about low-flow generation reveal aspects that are not systematically covered by the existing literature (which focuses on other dimensions of P and PET, or studies a smaller number of sites). Therefore, we believe that the provided results (and data) may be**

**useful for the hydrological community, as also already acknowledged by reviewers 1, 2, 3 and Ryan Teuling's student.**

**3) We now quantify the relative importance of P and PET in driving low flows (see section 3.5) using a multivariate regression between the low-flow magnitudes and the P and PET anomalies. This analysis is done for all years and for the years with the lowest low flows, and it highlights how PET increases in importance in the most severe dry years in our dataset. We also included an analysis of SWE in the revised manuscript, to better represent the effect of winter precipitation on summer low flows.**

**4) We now also created a new discussion section (4) in which we further discuss the novelty and implications of our results.**

*2) The reviews also raised concerns about terminology, in particular the use of 'extreme'. I agree that this terminology requires a specification, what frequency this refers to and that 'low flow' and 'streamflow drought' need to be distinguished. Regarding terminology issues pointed out by the reviewers, please make sure your terminology meets international conventions or clearly explain and reference any Swiss terminology used. For example consider the WMO Manual on low flow estimation (Gustard and Demuth 2005) or the textbook on low flow and drought estimation methods by Tallaksen and van Lanen (2004) . Also I suggest to draw parallels to recent research on drought propagation that correlated SPI and SPEI (essentially not much different from your non-standardized precip accumulations) as well as studies using basin properties/catchment characteristics for regionalization in general.*

**We addressed this issue by only referring to low flow conditions in this paper (and not droughts, deficits, and extremes), and by providing a definition of low flow for any of the analyses we performed. Some insights from past drought studies are still relevant to consider (e.g., in the introduction) so some of these studies are cited. We also included a more complete reference to the work on drought propagation (see lines 64-68).**

*3) Two reviewers (and I) also I find the mixed results and discussion confusing as it is not easy to distinguish plain results against wider interpretation with relation to other studies and current debate. The reviewers and I will review this very carefully again and I would highly recommend changing to a clear separation of*

*the two as part of the revisions. It may in fact help demonstrate the added value of the analysis (see comment 1).*

**We have now separated the results and discussion sections.**

*4) Subjective choices made, incl. the >3 sd exclusion, widow sizes for P accumulation, or the 1200 m.a.s.l., are not acceptable without good argument by physical reason or empirical proof. I expect the revised manuscript will either provide sound reasons for these decisions or test the results' sensitivities.*

**We understand that these choices look (and are) to some extent subjective.**

**First, we removed former Figure 1 (the part where we use 1200 m a.s.l. threshold was only chosen for showing different low-flow timings). Thus, this threshold is not relevant anymore.**

**Second, it is important to note that none of the (overall) results would change if these thresholds were modified. For example, while the 3-sigma rule is a widely-used tool in statistics to remove outliers from a dataset (Pukelsheim, 1994), using a bigger (e.g. 4-sigma) or smaller (e.g. 2-sigma) yields a similar overall outcome.**

**Third, we calculate the P and PET anomalies for time-windows ranging from 7 to 182 days (i.e., 7, 14, 30, 60, 90, 120, 182 days) which spans the whole time-period that seems physically most relevant when discussing the potential effects of P and PET departures from the norm on low-flow generation. In this sense the time windows are not arbitrarily chosen and are an integral part of our analysis. We better explain this now, and acknowledge that some very long-term memory effects may not be fully captured by this approach.**

*5) I take the liberty to add one missing consideration that has not been picked up, but that I find crucial to be addressed for the credibility of the study: magnitude and seasonality of observed low flows can be altered by water management operations, river regulation, sewage treatment plant return flows, the filling of reservoirs and minimum flow release from them, hydropeaking etc...(e.g. Pfaundler and Wüthrich, 2006). All these issues are very prominent in Switzerland and metadata on e.g. minimum flow releases and water transfers is available and can and should be used in any analysis. At the moment, the study considers*

*only natural climatic causes of low flows. It also does not consider the effect of trends and jumps due to human interventions over the study period. The minimum work to do is to carefully test for those, remove any records that show signs of human impact and carefully discuss how common water use and regulations may affect (intensify? compensate?) the identified climate-low flow relations.*

**We agree that human activity often influences low-flow magnitudes, especially in a place like Switzerland. We also point out that no comprehensive database exists with meaningful metadata of human influence on low flows for all of the 380 Swiss catchments. To avoid that human impacts alter our results we did the following additional checks and report them in a new discussion section in the revised manuscript (section 4.3):**

- **When compiling the dataset, we removed all catchments where there was obvious alteration of the flow regime (e.g., by screening the hydrographs). We removed for example all stations with hydropeaking or large reservoirs. That is also one of the reasons why the Alpine areas of Switzerland are less represented in our dataset (as they are e.g., within the report of the "Modulstufenkonzept" that is referenced by the editor).**

- **The absolute values of low-flow magnitudes are influenced by human activity. We do report and analyze the absolute magnitude of climate anomalies throughout our work, but only in the multivariate GLM model do we use the absolute magnitudes of low flows. In this case the predictive power is low, which shows that the absolute magnitude of low flow is poorly predicted by climate anomalies alone, and is more dependent on catchment attributes, and yes perhaps regulation. In the non-parametric correlations of low-flow magnitudes with P and PET we use the rank order of annual low-flows in the nineteen-year study period. There are good reasons to believe that the rank order of low flows is less influenced by human activities (i.e. the driest years will likely still have the lowest low flows, even when flows are partly managed). While individual cases can (and will) have some human imprint, we emphasize that we never establish results based on individual connections between anomalies and low flows (which can be highly affected by human influences) but rather focus and infer findings from repeating patterns across diverse conditions (which are less likely to be impacted by humans).**

- **In addition, in the revised version of the manuscript we also tested the sensitivity of our results by calculating the correspondence of low-flow magnitudes and climate-anomaly magnitudes for the 20% of catchments with the most human influence and the 20% of catchments with the least human influence (estimated using CORINE Landcover data as a proxy of water use/regulation). This analysis shows that the overall results do not change substantially, whether a catchment is near-pristine or heavily human-impacted.**

*Technical editorial comments*

*For the revision, please consult the manuscript preparation instructions - symbols and mathematical notation and avoid multi-letter variable names, in particular see suggestion of not using "ET" (or here "PET") in the HESS instructions.*

**We agree that in general using multi letter abbreviations can be confusing. However, PET seems to be the exceptions to this rule, since it is also widely used in literature (including HESS: Gu et al., 2020 - https://doi.org/10.5194/hess-24-451-2020; Jansen & Teuling, 2020 - https://doi.org/10.5194/hess-24-1055-2020; Callow et al., 2020 - https://doi.org/10.5194/hess-24-717-2020; Weerasinghe et al., 2020 - https://doi.org/10.5194/hess-24-1565-2020; Alam et al., 2020 - https://doi.org/10.5194/hess-24-735-2020; Qiu et al., 2020 - https://doi.org/10.5194/hess-24-581-2020; Jiang et al., 2020 - https://doi.org/10.5194/hess-24-1251-2020; Therefore, we will make the suggested change to alternative symbols, if the editor still insists on this.**

*Please mitigate some of the excessive citations (parentheses with eight references are not useful). Ideally be more specific about the relevance of the individual references to this study.*

**We considered this comment in the revision.**

*Please note that manuscripts "in review" cannot be cited and will have to be removed from the references.*

**We removed this citation.**

*Figure captions: As also noted by reviewers, add legends to figures rather than writing long descriptive caption texts and in particular shorten the captions by removing duplicate legend/caption text and long explanations.*

**We now have legends for Figures 2a, 3, 5 and 8; all other figures show always the same two things in blue and red (precipitation and PET). For clarity we still use extended figure captions for a good reason: experience has shown that readers who are browsing an article will find it much easier to understand if the main point of each figure is clearly stated in the caption, not just in the main text (which may appear several pages earlier or later, and thus will not be found unless the reader actually reads the whole paper from front to back). We also find that it is helpful to give detailed explanations directly in the caption, rather than expecting the reader to flip back and forth between the figure and wherever it is discussed in the main text. We will consider shortening the figure captions, by removing parts that are not essential.**

*First letter of axes labels should be capitalized.*

**We changed this.**

*Regarding R1's comment on the data statement and your reply: that's fine - please make sure that a list of station IDs and the address/who to contact for each station ID is included with the published dataset and/or as supplement.*

**Thank you.**

*References*

*Gustard A., Demuth S. 2008: Manual on Low-flow Estimation and Prediction. Operational Hydrology Report No. 50. WMO-No. 1029. Geneva.*

*Pfaundler M., Wüthrich T. 2006: Die Saisonalität hydrologischer Extreme. Das zeitliche Auftreten von Hoch- und Niedrigwasser in der Schweiz. Wasser Energie Luft 98: 77–82.*

*Tallaksen L.M., van Lanen H.A.J. 2004: Hydrological Drought: Processes and Estimation Methods for Streamflow and Groundwater. Developments in Water Science 48. Elsevier, Amsterdam / Oxford.*

We adjusted the manuscript according to the reviewer comments below. We already responded to these reviewer comments during the online discussion, but we repeat those responses here for the sake of clarity and completeness. However, some of our answers do not reflect the changes we ultimately made, therefore we report the original reviewer comments *in italic*, our original answers in bold with the modifications indicated by tracked changes.

**Reply to Anonymous Referee #1**

**We thank Anonymous Referee #1 for the positive and constructive feedback. We appreciate the suggested corrections and will address them in our revised version. Below we list our response (in bold) to the reviewer's comment (*in italics*).**

*Floriancic et al. explore how anomalies in precipitation and potential evapotranspiration shape the occurrence and magnitude of annual low flows across 380 Swiss catchments. The varying time period for the precip and PET anomaly calculation, with the end point being the day of the low flow, is a novel method for completing the joint analysis of climate drivers on annual low flows. I found the conclusions to be well-supported by the data. I particularly like how Figure 6 illustrates the role of long periods of PET in development of extreme low flows. The paper is well-written, and the methods are clearly outlined. I find this manuscript to be a significant contribution to the field, and I recommend it for publication in HESS. I have the following few minor/technical comments that should be easy to address:*

**Thank you.**

*L100: change "There were years whose lowest" to "There were years when the lowest"*

**We have corrected this.**

*L140-142: Sentence starting with "However," incorrect figure reference at end of sentence – should be Fig. 2a&b.*

**We have corrected this.**

*Figure 4: Suggest changing the color-scheme to something that is color-blind friendly.*

**We have changed the color options for Figure 4 to  make it colorblind friendly.**

*L308-311: Based on the winter precipitation versus annual low flow analysis completed in this study, I don't think this statement is sufficiently supported. As stated earlier in the paragraph, winter precipitation does not always accurately represent SWE. With such a range of catchment elevations (and thus climate conditions), a more detailed analysis would be needed to determine the impact of SWE on summer low flows.*

**We agree that the role of SWE in summer low flows cannot be directly inferred with the available dataset. In the revised manuscript we  have conducted an extra analysis in which we related the average SWE on 1 March over all catchments to the magnitude and timing of summer low flows. This extra analysis showed that also SWE, like winter precipitation, is very weakly related to low-flow magnitudes in catchments that experience annual low flows in summer. Therefore, although snow cover may affect low flows in higher altitude catchments (as some previous research has shown), we find little evidence of this effect in low altitude catchments in Switzerland.**

*L314: "most work has discussed individual drivers" – statement suggests that some work has analyzed multiple drivers of low flows, but no studies are referenced here. Section should reference the relevant studies listed in the introduction on L68-70.*

**In the revised version, we added appropriate citations here.**

*L319-321: I struggled to directly relate these broader implications statements to the results. How will the impacts be different between spring and autumn? What are the different implications of PET anomalies in May versus September? These implications are likely obvious to the authors, but on the first read through – I did not make the connection.*

**In the revised version, we  emphasize how antecedent conditions with regard to soil moisture state and subsurface water availability, and the water demand by vegetation in a catchment, matter. It is not sufficient to look at (the combination of) anomalies only, as the same combinations may occur throughout the year with different results, depending on soil moisture and vegetation state. Therefore, it is also necessary to include the timing of these anomalies together with the general climatology of a basin (see new discussion chapters 4.1 & 4.3)**

*L350: Data availability – Rather than making the data available "upon request", I would encourage the authors to provide open access to the compiled data used in their analysis (streamflow, catchment-averaged weather and climate conditions, PET, etc.) through an archiving medium such as figshare.com. While not essential, it would be beneficial.*

**We will publish the dataset in the "open access" ETH library collection. However, unfortunately we cannot supply the full daily mean streamflow dataset as they are only available at the Swiss cantonal authorities and the Swiss Federal Office of the Environment upon request. Nevertheless, our dataset will include the date of low flow occurrence (2000 – 2018) and the magnitude of the annual lowest flow ($Q_{min}$), and we will include a file with all contact information for the relevant organizations where the streamflow time series can be obtained.**

**Reply to Anonymous Referee #2**

We thank Referee #2 for the constructive feedback. We appreciate the suggested improvements and will address them in our revised version. Below we list our response (in bold) to the reviewer's comments **(**in italic**).**

*In this work, the authors assess how anomalies in precipitation and potential evapotranspiration shape occurrence and magnitude of annual low flows for 380 Swiss catchments comparing preceding precipitation and evapotranspiration for different periods to the annual minimum flow. After an initial analysis of annual low flows, the authors decided to focus on summer low flows for the rest of the analysis. The paper is clearly outlined and easy to read. I agree with reviewer 1 that I particularly like Figure 6 illustrating the increasing role of PET during development of extreme low flows. However, different to reviewer 1 I have some major comments that I find important to address before publishing:*

**Thank you. Below we address the comments point-by-point.**

*Major comments:*

*Summer and winter low flows*

*After an initial analysis of annual low flows, the authors decided to focus on summer low flows for the rest of the analysis. This analysis takes relatively much of the full paper both in text and in Figures (1, 2 and 4). While I agree that it is important to differ between summer and winter low flows particularly when analyzing the drivers, I do not see that the results of general occurrence are new (e.g. Smakhtin 2001 and references therein, Fiala et al. 2010 and basically all runoff regime literature for Switzerland in particular e.g., Weingartner and Aschwanden, 1992) nor that they deserve this weight in the article. I suggest to minimize this to the introduction refereeing to the relevant references and remove Figure 1 and focus in Figure 2 on only the summer events or split in to summer and winter low flows at the beginning and then assess both for winter how snow (or precipitation and temperature) shapes low flow occurrence and magnitude and for summer how precipitation and evapotranspiration shape low flow occurrence and magnitude*

We have edited the introduction regarding summer vs. winter low flows and included further literature as requested by Referee 2.

We also removed Fig.1 from the manuscript to avoid repetition of previous findings. We furthermore use (the new) Fig. 1a and b to argue why we relate summer low flows to precipitation / PET anomalies, and expect these relations to hold for a wide range of low-elevation catchments

*More focus on the shaping*

*Instead of counting and presenting summer and winter low flows representing the annual low flow, I think it would be more interesting to try to add on the shaping of the low flows caused by precipitation and evapotranspiration. For instance, in addition to correlation between precipitation and PET separately on extreme or less extreme low flows it would be interesting to look at their combined effect. And then how much of the combined effect could be attributed to precipitation and to PET. This would allow a better relative quantification and really add to the values of this study.*

We emphasize that Fig. 2 and Fig. 5 are studying the combined effects of P and PET on low flows already. In the revision we partitioned the effects of the two drivers more explicitly, by comparing the predictive skill of bivariate regressions between the individual climatic  anomalies. and $Q_{min}$, and a multivariate stepwise GLM regression between a both anomalies at all durations and $Q_{min}$. We show with this analysis that while precipitation explains most the variability in $Q_{min}$ overall across all years, in very dry years the situation is opposite and PET becomes a more important predictor than P, especially at shorter durations. However, it has to be understood that the predictive power of the multivariate stepwise GLM model for Qmin is low overall.

*Choice of summer months*

*The authors focus at extreme low flows and the preceding conditions and chose 2003, 2011, 2015 and 2018 as the relevant years. The drought in 2011 was finished for most catchments before July, I urge the authors to look into the data and if so, adjust the analysis by either treating 2011 differently, i.e. not*

*considering it a summer but a spring low flow and or change the analysis period for all years that was defined by the authors Jul-Nov.*

**We agree that the drought in 2011 is predominantly occurring in spring throughout the Swiss catchments (i.e. only 143 of 380 catchments have their low flows in July through November in 2011).** ~~We choose the 6-month period July-November because most low flows occur during this period. However, in some cases low flows do occur outside of this window (e.g., 2011). When we calculate the statistics of the main figures that use the 6-month window (e.g. Fig. 2 c&d, 3, etc.) the results do not change significantly because our chosen period captures almost all low flows. In the revised version we will more clearly acknowledge that not all low flows are captured by this window, but also that this choice does not affect the results significantly.~~ We changed the period that we consider for warm-season low flows in the revised version of the manuscript to May through November.

*Terminology*

*There is a seamless transition from "drought" to "low flow" and mixed use of "droughts" and "deficits" while the citations support both. I find this mix critical since already deficit (even when regularly/seasonally occurring) causes low flows but meteorological droughts are larger deficits than normal, i.e. the regularly/seasonally occurring deficit. Please, revise the introduction to distinguish clearly between these. This would help the reader in the analysis that follows.*

**In the revised version we will better distinguish between drought, deficit and low flow. In short, we will minimize the use of the term (and references to) drought, since low flows are not necessarily droughts.**

*Likewise, the authors use often only the term "low flow" when actually referring to "extreme low flow". This can result in wrong statements (e.g. L72 "low flows are exceptional flow conditions" or L186 "triples the chance of an annual low flow"). And I would ask the authors to revise and correct the usage throughout the manuscript.*

**In the revised version of the manuscript we emphasize that we  only discuss the annual 7-day lowest flows throughout the manuscript and we fully avoid the term "extreme" to enhance clarity. We agree that not being explicit about this could have led to misinterpretation (e.g. L72 in the original manuscript).**

*Mixed results and discussion section*

*In my opinion results and discussion should be separated. This allows to focus on the results. Only then what we can learn from the results and where we might to be a bit more careful, then also relate and compare to what was done elsewhere and where limitations and possibilities lay. In the present form the manuscript mixes these aspects and is more difficult to search for s specific result /argument this way.*

We have separated these two sections in the revised manuscript.

*Minor comments:*

*L25-27 Remove this sentence for the abstract*

This sentence may sound trivial but is needed for the logical flow of argument in the sentence following it. Therefore, we prefer to keep this sentence. We edited the abstract to improve its clarity.

*L40/41 is "landscape" only including surface features? Maybe use rather catchment properties".*

We use landscape because we thought it would make clear that it does not include all catchment properties (such as its climatic conditions). Landscapes extend into the subsurface.

*L72 this statement is not correct they occur every year. Make clear that it is about extreme low flows here!*

In the revision we chose more precise wording which reflects that the low flows we are studying are "annual lowest flows" and not  necessarily  extremes in the long-term record. We avoid using the term "extreme low flow" throughout the revised version of the manuscript.

*L92-94 a summary table (maybe only in the supplementary material) would be helpful*

All data (and a summary) will be provided through the "open access" platform of the ETH library.

*L121 1200m asl, why this threshold? Why not a range?*

We removed this part from the revised version of the manuscript.

*L130 remove "However, this remains to be tested".*

OK.

*L147-150 For these low flows, people are usually prepared. Here it would be interesting how much more extreme are others. If it was due to a lack of precipitation, the signal should be visible in spring melt.*

We agree that such winter precipitation deficits can have effects on flows later in the snowmelt season, and will likely be visible in the data. However, the aim of this paper was not to explore all these hydrological connections, but rather focus only on the climatic conditions leading to the lowest flow in the year. This remains an interesting suggestion for further research.

*L154 "suggesting…" could be also formulated that lower elevation Swiss catchments could be representative sample for global summer low flow? (and maybe not even global but for humid regions with seasons?); This part would better fit in the introduction or methodology/catchment section.*

This part was removed from the revised version of the manuscript.

*L161 altitudinal variation in 30-day anomalies: could that be influenced by catchment size? A large catchment might not react on such an anomaly a small catchment not anymore if the driving anomaly is at the beginning of the period.*

We tested if catchment size affected the altitudinal variation. While such effects can be expected, no clear signal was found, probably because the catchments are relatively small (< 519km$^2$, with a median of 74km$^2$.

*L174 "substantial site to site variability" can this be quantified?*

To clarify what this variability refers to, we now explicitly refer to Fig. 1c and 1d for the reader to look at the spread. We also add the range in the text.

*L181 can these 8L189-196 These results are not surprising (the authors refer even to studies that found the same) but nicely illustrated and supported by the data. However, it would add to the value of this study to quantify the contribution of precipitation and evapotranspiration.*

We agree that these results are maybe not surprising, as they have been shown for individual cases (as referenced earlier in the paper). Our work improves past studies by (a) providing a large dataset which shows the variability and consistency in low flow-climate relations among basins; (b) quantifying the effect of duration of the climatic anomalies required to generate the extreme low flow events; and (c) separating the effects of precipitation and PET. We believe this provides a more robust picture of otherwise intuitive relations.

*L189-196 Consider also to compare to Stahl et al. 2010*

We have put our results in context of the findings of Stahl et al. 2010.

*L196-197 delete sentence*

It is unclear to us why this sentence should be deleted.

*L203-204 it is possible to avoid that by the study design (see also my major comment on seasonal split)*

We agree that it is possible to select data such that only the drier years are kept. However, that is not the purpose of our study. We rather discuss that not all annual low flows are created equal.

*L222-228 that depends on how one looks at the drought: is the same scale as in the references used? The effect was also found in low flows but maybe not in the metric "annual low flow", again distinguish between drought and low flows (see also major comment on terminology above) Figure 4: Looking at the figure makes me wonder how/if the regulated catchments might influence the pattern presented. Could that be picked up in the discussion?*

**In the revised version we avoid the term "drought" and only refer to "low flows". We  discuss the influence of flow regulation on low  flows in the revised manuscript and provide an additional analysis in the new discussion section 4.3.**

*L241 How brief since the study is about anomalies?*

**We will remove this.**

*Technical comments*

*While I find that active voice generally a good choice, I would avoid starting every sentence with "we" (e.g. 2.1 but also elsewhere), please revise.*

**We have tried to reduce the prevalence of sentences beginning with "we" (although they are usually the most compact, clear and direct way of expressing things).**

*L22 "dry years saw" please rephrase*

**We have changed this.**

*L28 redundant, delete either "could" or "potentially"*

**We have deleted "could".**

*L44 "(PET)" -> ", PET"*

**We have changed this.**

*L44 remove "should" and "usually"*

We have deleted "should" and "usually".

*L46 remove "made"; split sentence: ": : :to a catchment. Hence a sustained: : :"*

We have changed it accordingly.

*L49 Make a new paragraph*

Ok.

*L64 "smaller" = "lower"?*

We have changed this.

*L66 "comes" -> "occurs"; remove "the" before summer and before winter*

We have changed this.

*L79 "useful" = "suitable"?*

We have changed this.

*L94 "quantified" -> "estimated"*

Ok.

*L100 "years whose lowest annual flows were much" -> "years with lowest annual flows much"*

We have changed this.

*L103 "low flows" -> "extreme low flows"*

We have changed this.

*L224 remove "the" before summer and before winter*

**We** **have changed** this.

*L238 "more strongly" -> "higher" (also in L244)*

**We** **have changed** this.

*L259-262 rephrase to make more concise*

**We** **have rephrased** this sentence.

**Reply to Anonymous Referee #3**
* * *
**We thank Anonymous Referee #3 for the detailed, constructive feedback. We appreciate the suggested corrections and will address them in our revised version. Below we list our response (in bold) to the reviewer's comments (***in italic***).**

*The objective of presented study is to investigate how precipitation (both summer and winter) and PET anomalies influence low flows across Switzerland both in typical and exceptionally dry years. In my opinion, authors provided detailed and important insight into climatic drivers controlling low flows based on data assessment from 380 catchments in Switzerland. In general, I found the results interesting, although the methods used are not novel. I found the main contribution in assessing a large number of catchments which may help us to better understand why catchments sometimes behaves differently, which are main controls and thus what may happen in the future in a warming climate. Thanks to a large number of catchments covering different elevations, I think the results can by generalized to other regions, at least to those located in similar climates. In this respect, the results have an international value and may be very useful for hydrological community. Therefore, the results are important and certainly appropriate for HESS. However, I have some comments listed below, which need to be addressed before I can recommend the manuscript for publication. These comments are mainly related to methods and results interpretation. I hope that these comments will help authors to improve the manuscript.*

**Thank you.**

*Major comments:*

*Authors used winter precipitation to show how winter and snow conditions are important for summer low flows. Although this is an important aspect especially for higher elevation catchments, I am not sure to which degree authors were able to capture the snow effect by selecting just winter precipitation as a single variable. The winter precipitation does not tell us whether the precipitation is falling as rain or snow. This is, in my opinion, very important since snow contributes to runoff much later than rain and thus influence the seasonality of groundwater recharge and potentially summer low flows. Therefore, I am not sure whether the winter precipitation could correctly capture this issue well enough to make any general conclusion. Using some snow-related metrics (snowfall fraction, snowfall water equivalent, annual maximum SWE or similar) would be perhaps better to show whether there is (or is not) any relation. Therefore, I would be careful with interpretation going towards the role of snow. I do not see much*

*evidence in authors results to make some conclusion, although several previous studies quantified this effect at different elevations.*

**We agree that winter precipitation is not an ideal proxy for snow.** **In the revised version we added a new analysis with SWE on 1 March as a proxy for snowmelt potential affecting low flows. This analysis confirmed that also SWE, like winter precipitation, is weakly related to low-flow magnitudes in catchments that experience annual low flows in summer. Therefore, although snow cover may affect low flows in higher-altitude catchments (as some previous research has shown), we find little evidence of this effect in lower-altitude catchments in Switzerland.**

*I am not fully convinced that assessing both winter and summer low flows is a good approach since the meteorological drivers are different for both of low flows types (see e.g. Harpold et al. (2017) for general overview). I found the mixing of both types throughout the manuscript sometimes a bit confusing. Authors did first analysis (Fig. 1 and Fig. 2) using annual low flows from all catchments (regardless whether they were summer or winter) and later they decided to further analyse only summer low flows. Although I would maybe prefer to focus only on summer low flows in the study, I accept the authors' decision to make first some results related to both winter and summer low flows together and later focus just on summer low flows. However, I am a bit confused how authors exactly proceeded to select catchments and years for summer low flows analysis (but maybe I only missed something). First, it seems that authors analysed all seasonal low flows occurred in the warm period for all study years (even in case that annual low flow occurred in winter). However, later (L306-307) it seems that authors completely excluded catchments/years in case that annual low flow occurred during winter. The latter approach could result in excluding many of the highest elevation catchments from the analysis (and thus it might lead to the conclusion that winter precipitation is not an important signature to influence summer low flows as noted in the previous comment). Therefore, please clarify how you proceeded. I would think that the first mentioned approach is more appropriate and should be used in the analysis (especially in case you are focusing on the role of snow or winter precipitation in addition to role of previous precipitation and PET).*

**This is a very good point, and indeed in the main part of the paper looking at the effect of climatic anomalies we focus exclusively on catchments that have the lowest annual flow in summer and autumn, thereby excluding most high-altitude catchments. This is one of the reasons why winter precipitation (and SWE) do not affect low-flow magnitude. We explain this in the revised paper more clearly.**

*Regarding to the comment above, I think that mixing the summer and winter low flows in Fig. 2 (top panels) is not a suitable approach since the climatic controls are different for both type of low flows. As it is now, you are losing a lot of information, especially in higher elevation catchments, because you are trying to describe (mostly) winter low flows in these catchments using variables, which are not much relevant. Therefore, I would suggest to make the Fig. 2 just for June (or July) to November low flows. Then you would see, whether the precipitation and PET are important drivers for summer low flows even at highest elevations or whether the figure would suggest that there might by also something else (e.g. snow from preceding winter). In case you decide to keep Fig. 2 as is, please consider to split it into two figure, since the mixing of annual and summer low flows in one figure (top panels vs. bottom panels) is, in my opinion, confusing.*

**The purpose of Fig. 1a and 1b is to show that the importance of P and PET as drivers for annual lowest flows systematically changes with elevation. This is in our opinion a useful result that we also want to show in the revised version of this manuscript. We now realize that using a subset of these data for Fig. 1c and 1d may confuse the reader. Therefore, we will better emphasize the different samples used to create the upper and lower panels of Fig.1.**

*Authors calculated preceding PET as one of the main climate drivers. However, physically correct way is to use actual evapotranspiration (AET) instead of PET. I am aware, that calculating AET would not be such easy. Nevertheless, the relation between PET and AET is not always straightforward since higher PET do not necessarily mean higher AET (especially in lower elevation catchments with lower precipitation, higher water demand and thus lower water availability). Therefore, I would appreciate more discussion related to PET vs. AET interactions.*

**Obviously, we agree that AET is the physical process by which water leaves the catchment, that may lead to low flows (and could thus be considered a driver). However, the purpose of our paper is to infer the climatic drivers of low flows. AET is not a climate driver of low flow, it is the outcome of how climate interacts with the soil and vegetation in the catchment. Therefore, we choose (high) PET as a driver because PET is the climatic condition that drives AET (and subsequently low flows). In the revised version we  better emphasize our choice of PET as a driver over AET as a driver, and we discuss its limitations especially regarding the complementary relationship between AET and PET. Using AET would furthermore require an additional soil water balance model which adds uncertainty to the analysis.**

*Minor comments:*

*One of the conclusions is that low flows are controlled by either low precipitation or high PET or combination of both. This is not surprising since there are not many other options (at least for summer low flows in near-natural catchments). Therefore, I would rather highlight implications which arose from results, but which are not such trivial. In this respect, I would recommend to slightly reformulate the respective part of abstract, short summary and perhaps also hypotheses (line 75) to better highlight the novelty of your work.*

**Indeed, it is no surprise that PET and P drive most low flows. However, the purpose of our manuscript is to show to what extent, and which characteristics of P and PET drive low flows, and how these vary spatially. These more detailed pictures of low-flow drivers are nontrivial and we  try to better express their value in the revised paper.**

*Authors used winter precipitation, but this signature is not mentioned and explained in the methods section.*

**In the revised version, we have added the description of how we obtained winter precipitation also in the methods (rather than just in the later stages of the paper). We also added a new analysis with SWE on 1 March to quantify the potential effect of winter precipitation and snowmelt on $Q_{min}$.**

*I suggest to create a map showing the location of catchments. I think that just a simple map of Switzerland with shaded DEM with points showing the position of catchments outlets would help those readers not*

*familiar with Swiss hydrology. Maybe, catchment points might be coloured according to catchment elevation (or something similar).*

**We have added such a map.  in the  supplementary material.**

*It is a bit questionable to describe the previous precipitation just using the sum of precipitation from the defined preceding period. The reason is that the importance of precipitation for the low flow at the specific date changes when going back in time (precipitation closer to the day with the low flow is more important than that occurred earlier). Did you also consider applying some kind of continuous precipitation index, e.g. current precipitation index CPI (Smakhtin and Masse, 2000)? I would appreciate more discussion on this issue.*

**We choose a time-window to reflect that low flows are typically not generated instantly, but are generated over longer time spans. We agree that precipitation during times closer to the actual low flow will probably often impact the flow more than precipitation during a longer time prior to a low flow. Alternative metrics such as CPI may account for this fact (to some extent) and their merit . However, precipitation in the periods immediately preceding low flows often does not significantly refill groundwater stores, and thus may have very little impact on the low flows themselves (they only result in a short peak in the hydrograph). Such effects are not captured by CPI.**

*L101: Please, add a brief information why you remove "unusually high annual low flows". I see the point, but it would be good to clarify it.*

**This is explained in more detail in the revised version of the manuscript.**

*L147-149: Could you please add some references regarding this statement? Just to avoid speculations, since you cannot prove this based on your results.*

**The appropriate references were added.**

*Fig. 2: Here it is nicely shown that PET anomalies are relatively less important compared to precipitation anomalies (inter-annual variations up to 40 mm for PET, but up to -200 mm for precipitation).*

**Thank you.**

*L201-202: Maybe I missed something, but I do not see the described effect (wet years)*
*from Fig. 3b*

**Wet years refers to years with higher low flows (Fig. 3b). We now realize this may be unclear to the reader and therefore we edited this statement in the revised manuscript.**

*Fig. 3a: Why there are still some years when low flows are preceded by above-average precipitation and below-average PET (bottom-right quadrant)? Would the figure looks similar also for other than 30 days time-windows (e.g. 60 and longer)? You suggested some explanation in lines 201-205, but could you be more specific?*

**We have extended our description of why it makes sense that above-average P and below-average PET anomalies are observed when (i) the seasonality of the flow regime outweighs the effects of shorter-term weather, and (ii) when very wet years occur (with high low flows).**

*Fig. 4: This figure shows both summer and winter low flows. However, earlier you stated that only summer low flows are analysed starting from Fig. 3 (L155-156). Therefore, could consider putting this figure as a Fig. 3 to be consistent. Additionally, I would maybe change the colour scale by using "cold" colours for cold months and "warm" colours for warm months.*

**As stated before, in the revised version we  better emphasize when and why we use summer vs. all low flows in various parts of the paper. **

*L237: Please, specify the thresholds for "below-threshold" precipitation and "above-threshold" PET (maybe in methods as well).*

**We specified these thresholds in lines 251-252 in the original manuscript. In the revision we have also included this description in the caption of Fig. 4.**

*L286-287: This hypothesis would be correct only in case that most of winter precipitation would occur as snow. Therefore, not higher winter precipitation in general, but higher snowfall (snow storages) should*

*lead to larger and later summer low flows (but only at high elevations with high snow storages). Please, consider reformulation. In this respect, you correctly pointed to the fact that winter precipitation sums do not accurately represents SWE (L299).*

**We refer to our earlier more detailed comment on how we addressed this limitation.**

*L325. I would maybe add "winter" to describe the precipitation in California. In contrast to humid catchments in Central Europe, the previous summer droughts in California were mostly driven by lack of winter precipitation (and snowpack).*

**We have added "winter".**

*Technical corrections*

*L95: Please, add the reference to RhiresD and TabsD products*

**We have added a Meteoswiss reference.**

*L106: I would not use the term "long-term" for time period of 18 years. Instead, I would directly specify from which time period the average has been calculated.*

**We have changed this.**

*L142: I think you wanted to refer to Fig. 2a&b*

**Thank you, we have changed this.**

*Figure captions: A lot of text in figure captions (Fig. 2, 3, 5 and 6) is related to figure interpretation rather than figure description. In my opinion, these parts would better fit directly to the main text*

**We tried to include the main message of the figure in every caption. We believe this is informative to the reader.**

*L177: "Our previous results ...". I would remove "previous" since it implies something you did in some previous study. Alternatively, be more specific instead (e.g. "results shown in figure/section no. ...").*

**We** **have changed** this.

*Fig. 6: Please, consider larger axis captions to increase the readability.*

**OK.**

**Reply to Anonymous Referee #4**
* * *
**We thank Anonymous Referee #4 for the feedback. Below we list our response response (in bold) to the reviewer's comment (***in italic***).**

*In this paper, the authors assess to what extend precipitation and PET anomalies trigger summer low flow events in Switzerland. The assessment employs Spearman correlations between low flow magnitude and climate anomalies in P and PET aggregated over varying lead times before the peak of the low flow event. The correlations are overall weak, but still indicate that most low flows arise from the compound effects of precipitation and PET anomalies, with longer and larger anomalies related to more extreme low flow events, as one would expect. The assessment of lead times before the peak of the event is not new (e.g. Fangmann and Haberlandt, 2019 on a monthly time scale), but indeed appropriate to assess the genesis of events.*

**Indeed, drivers of low flows have been studied before (as reflected by the citations, including Fangmann and Haberlandt, 2019), and the result that both PET and P are important for low flows may not be a big surprise. However, the paper provides insight into the durations, magnitudes, and timings of the anomalies that drive low flows, and how these vary across hundreds of catchments situated in diverse landscape conditions. These more detailed insights about low-flow generation reveal aspects that cannot just be derived from intuition. In addition, the aspects of low flows that we discuss are not captured by previous studies, because they study other regions, and/or they study different aspects of low flows. Therefore, we believe that the provided results (and data) may be useful for the hydrological community.**

*While the paper is generally easy to follow, the paper appears to suffer from weakly formulated research question and consequently from a limited scope of the study. The results remain superficial and do not provide sufficiently new insights in low flow generation in Switzerland. I therefore cannot recommend the paper for publication in its present form. I will provide detailed feedback below, which I hope to be useful for the authors for further elaborating the paper.*

**In the revised version we have sharpened the research questions, to make them more specific and clearer. Since the detailed comments on this issue are discussed below, we refer to our detailed responses there on how this was done.**

*Weakly formulated research questions and and consequently from a limited scope of the study. The results remain superficial and do not provide sufficiently new insights in low flow generation in Switzerland. I therefore cannot recommend the paper for publication in its present form.*

**Below we respond to the detailed comments that refer to this concern.**

*Science question*

*Science questions (or hypotheses) of this paper (Line 73-75) are formulated in a way that everybody would immediately agree: There is little doubt that low flows will typically occur after anomalous weather conditions, and that most extreme low flows will be associated with the most extreme weather conditions. This leads directly into a quite superficial analysis and weak conclusions. I urge the authors to sharpen the science questions and, accordingly, the study design, in order to gain more significant insights in how precipitation and evaporation together generate low flow events in Switzerland. I agree with Referee 2 that the focus of the paper should be much more on the interplay of the two meteorological drivers. And their relative importance for events with different time of occurrence within the summer/fall low flow season.*

**We point out that lines 73-75 in the original manuscript do not represent our science questions. However, we have rephrased the text to make this distinction between "introductory text" and "science questions" clearer.**

**In the revised version we have sharpened the actual science "questions"  in ines 113ff "We investigate (a) how precipitation and *PET* anomalies separately and jointly shape the occurrence and magnitude of  low flows across Switzerland, (b) which durations of  these anomalies have the **

**strongest impact on low-flow occurrence and magnitude, both in typical and in exceptionally dry years, and (c) how winter precipitation and snow packs influence the magnitude and timing of summer low flows."**

**None of the above questions is answered in previous studies for our study region. In addition, we have updated all of the results and discussion paragraphs to better present the results and their implications. In particular we have added a new analysis in which we objectively compare how much of the total predictability of $Q_{min}$ comes from precipitation and how much from PET. With this analysis we show that while anomalous precipitation is the overall dominant climatic driver, in the most driest years of the record, the relation switches and PET becomes the dominant driver for explaining variability in Qmin. This addresses the interplay of the drivers that the referee is referring to.**

*Methods*

*The paper also suffers from weakly defined analyses. The methods section does not provide all necessary methodological details; they pop-up in a mixed results and discussion section. This makes analysis rather ad-hoc and hampers a well-structured assessment of the research question. I strongly advocate organizing the paper into clearly separated methods, results and discussion sections to foster a transparent, indepth assessment.*

**Apologies for the confusion. In the revised version we made sure that all methodological aspects are already explicitly mentioned in the methods section.**

*In the following I review the used methods found in the results section.*

*In Section 3.2, the purpose of this "first correlation analysis" is not clearly defined (ref. also to the vague section title). The section assesses the correlation of 30- day-anomalies. For what purpose the time window has been chosen, and what may analysing 30 days before the event tell us has not been indicated.*

**The purpose of section 3.1 is to reveal what magnitude of climate anomalies are typical for low flows, how this varies between P and PET anomalies,  and whether P or PET appears to be more important. In the revised version we have introduced these purposes more clearly, both in the methods section, and the results section. We have also provided a more quantitative perspective on the P and PET partitioning (see above).**

The purpose of choosing a 30-day window is to reflect that low flows are generated during a prolonged period of anomalous climate. We show the results for 30 days, but emphasize that other time- windows (from 1 week to 120 days) yield broadly consistent results.  We choose 30 days as the result to present, because 30 days (as later shown) is the time window which explains most typical low flows (Section 3.3).

*Section 3.4 Duration of climatic anomalies – The analysis of durations of anomalies before the peak of the event is largely depending on very short interruptions of the climate anomaly that have no effect on streamflows. Some pooling would be necessary to filter out disturbances in this type of analysis. The second analysis based on various time windows is more robust, and the most insightful analysis of the study.*

We acknowledge that short (irrelevant) interruptions may affect the determined length of a climate anomaly. To address this we  use a 10-day moving average of time series to filter out short duration interruptions.

*Section 3.5 The role of winter precipitation is not a pertinent research question, it is well-known that in an Alpine environment it is rather snow-storage than accumulated precipitation that shapes summer low flow with respect to timing and magnitude. Analysing winter precipitation (instead of snow storage and snow melt) has not the potential to lead new insights in the low flow generation process in Switzerland.*

As pointed out by previous reviews, winter precipitation is not always a robust proxy for winter snowpacks. We have changed our discussion of the role of winter precipitation for summer low flows accordingly. We would like to emphasize that it remains largely unquantified how winter conditions (either snow specifically, or both snow and winter rain) affect low flows across the Alps. This is clearly important information as we all agree that in many high Alpine catchments winter conditions can shape summer low flows. In addition, in the revision we use  SWE on 1 March in all catchments to show that also stored water in the snowpack ready for melt in spring is not a strong predictor of

low flows in catchments where the  lowest annual flow is in summer (low-altitude catchments in our dataset). See also answers to previous referees on this topic.

*Specific comments:*

*L 41: It is not either climate, or catchment, but the combined effect of meteorological drivers and catchment functioning that determines streamflow.*

We obviously agree that both climate and the catchment itself shape low flows (as we tried to convey in the original text). We have rephrased this to avoid confusion.

*L68: Contradiction to "the effects of evapotranspiration on low-flow occurrence and magnitude have received relatively little study" (two paragraphs above).*

In the revised version we have rephrased this to make clear what aspects of ET have not received much attention to date.

*L72 Sentence does not make much sense.*

In this sentence we aim to explain why focusing on climate anomalies makes sense. We have rephrased it to avoid confusion.

*L73 ff: Please revise hypothesis (better formulate them as science question(s) and objectives of the study. Avoid duplication of the overall aim into one objective (currently objective a).*

As stated earlier, this is not the hypothesis we test in the paper. We now realize that this confusion can arise (probably because we used the word "hypothesize" in this sentence). We have reformulated this statement to reduce the chance of this confusion.

*L114: One sentence methodology, apart from the definition of the anomaly measures, is definitely too short.*

In the revised paper we have added text that explains the rationale of this analysis.

*Section 3.1: This is prior knowledge and should go into the introduction*

**We agree. We removed Section 3.1 from the revised manuscript and we state the most important points in the introduction.**

*L222: Statement is not true. What the cited papers say is that large parts of Europe were affected by the drought events of 2003 and 2015. But papers also show how different timing and magnitude of events were across Europe.*

**We now realize that we oversimplified the spatial coherence reported in previous studies. We change the interpretation of our results accordingly, by not stating that Switzerland is necessarily in contrast with other regions of Europe. However, we would like to point out that the spatial gradients in low-flow timing in Switzerland appear stronger than in some other parts of Europe. We have reformulated this in the revised manuscript.**

*L226: ditto*

**See response to previous point.**

*L239: Citation needed. What do you mean by erratic?*

**By "erratic", we mean that daily precipitation is more irregular in time (compared to PET). We now use the word "irregular" to be clearer. We are unsure where a citation is needed in line 239?**

*L285: No, snowpack is not the same as precipitation sum, snowmelt is precipitation redistributed over time.*

**We forgot to add an additional line of logic in our statement that connects winter precipitation as a (weak) proxy for snow for our study region. We will add this to avoid confusion. In addition, we rephrased the implications of the results as we earlier discussed that winter P and snow are not identical. We also supported the results of this part of the analysis with SWE data.**

*L300 ff: "if SWE is important, we expect to see stronger correlations between winter precipitation and summer low flows at higher elevations – see my previous comment (L222). The following analyses are wrongly motivated and results overinterpreted.*

**We have revised the text to reflect that winter P and snow (pack/fall) are not identical, and we have added supporting analyses with SWE (see above and responses to previous referees)**

*L327: Remove sentence, as the paper does not represent a novel methodological Framework*

**We agree that no real framework is provided, and have removed this statement accordingly.**

**Reply to the Student Review**

**We thank Robert Lubben (and Ryan Teuling) for their feedback. We very much appreciate the suggested improvements and will address them in our revised version. Below we list our response (in bold) to the reviewer's comments (***in italic***).**

Short summary:

The reviewed paper is about the effects of climatic anomalies on low flow occurrence in 380 Swiss catchments for the period 2000-2018. The low flows are defined by the annual 7-day lowest flows. The anomalies in precipitation and evapotranspiration are calculated for several time periods before the annual low flow occurrence (7 up to 182 days). With this data two hypotheses are tested 1) low flow occurs after anomalous weather conditions and 2) that the most extreme flows will be associated with the most extreme weather anomalies. The results of the study are that the low flows mostly occur after anomalous precipitation and evapotranspiration events. Most of the low flows (92\%) are influenced by below average precipitation and 70\% is influenced by above average evapotranspiration. Also, the extreme weather anomalies, or meteorological droughts, tend to generate extremely low flows. Winter precipitation, as SWE, was less important for the low flow seasonality than the climatic anomalies in this study.

General comments:

Overall, this paper is well written and of high quality. No textual errors could be found. The two hypotheses and their origins are stated well. The need for having an answer on the hypotheses has a good relation with former research. The conclusions clearly give an answer to the hypotheses and the discussion about the effects of SWE on low flow seasonality is helpful for placing the results of this study in context and linking this study to former research. The figures of the results help to give a better understanding and can easily be linked with the corresponding text. The overall structure of the paper really shows the reader how the research is conducted and which methods are used. The methodology clear and can (almost) be reproduced with the given description. However, the method for estimating potential evapotranspiration needs a bit more clarification, see major comments. The data that has been used for the analysis is of high quality and has a high spatial density. The estimation of low flows is a good method that is widely used in low flow analysis and it identifies the occurrence of a low flow in such a way that

errors in measurements and can be filtered out (Smakhtin, 2000). The precipitation and evapotranspiration anomaly quantification for different durations is a good way of finding anomalies in the climatic data if a sufficient amount of years is used. The study is novel because, this type of low flow analysis is not carried out before on this scale with this many catchments. Although, the used method is not exceptional or novel, it gives a good view on the separate and combined effect of precipitation and evapotranspiration on low flow occurrence. Also, the used visualization method shows the effects of extreme evens well (like figure 6 in the manuscript). The visualization of enhanced PET under drought conditions is also nicely visualized. These figures help by understanding and supporting the text and give the reader a clear first glance at the study and the results.

Low flow seasonality is a relevant topic with the expected increase in weather anomalies by climate change. Understanding the climatic drivers and their impacts on low flow genesis can help understanding the processes leading to low flows and help in managing discharge. This becomes more and more relevant after the recent drought years. In my opinion, there are no mayor points that limit acceptation of this paper. However, there are a few major comments that could be useful to take into consideration.

**Thank you.**

Major comments:

There are three major comments that can be raised while reading this work. The first comment is about the exclusion of 2\% of the low flows, this is not supported in the current context. Low flows in years with high precipitation will still be caused by anomalies in precipitation and evaporation. Therefore, the exclusion of the low flows that are above the value of three standard deviations from the mean seems not logical. In this way the method with which the low flows are estimated is in conflict with the actual used method where 2\% of the data will be excluded because it is above 3 standard deviations. This means that the definition of a low flow has to be changed or that all data has to be used in this study (including the 2\%). Including this data should not influence the results in a negative way because the drivers of low flow are likely the same. Another way to clarify this could be to do the analysis with including the 2\% of low flows and then conclude that the abnormally high low flows are not significant or hinder the analysis. After the analysis the high low flows can be excluded with a good reason.

**The exclusion of 2\%% of the low flows does not change the overall results. The reason for the exclusion is not to get rid of "wet years" but rather to remove outliers in our dataset. Measurements of low-flow magnitudes are critical, and with the 3-sigma rule we exclude outliers that might result from faulty measurements. To avoid the possibility that these data points distort our analysis we remove the upper 2\%% of our data points. This is similar to many robust estimation methods in statistics, in which the extreme tails of a distribution are trimmed off before statistical parameters (means, variances, etc.) are calculated. We have made that clearer in our revised version.**

The second major comment involves the used potential evapotranspiration estimation method. This is only addressed very briefly via referencing to the paper of Hargreaves and Samani (1985). The used calibration parameter value and the source of solar radiation are not given in this way. The calibration value can influence the results for the Hargreaves PET estimation significantly. Especially in humid areas the PET can be overestimated when using Hargreaves (Trajkovic, 2007). A method that uses observed radiation can therefore result in less evaporation which influences the overall PET anomalies. The estimated PET can possibly be validated with the lysimeter used by (Seneviratne et al. 2012b). A clearer description of the used PET method by including the formula and the used values for the parameters will help by giving insight in the uncertainty of the PET estimation. This will also help making the methodology more clear and improve the possibility to apply this framework elsewhere.

**Our analysis only depends on the relative values of the PET estimates, not on the absolute PET values themselves. Because the Hargreaves calibration parameter only re-scales the PET values, it has no effect on their relative magnitudes and therefore will have no effect on our result. We can of course present the Hargreaves formula and the values of the coefficients for readers who are unfamiliar with the particulars.**

The last major comment is on figure 4 (page 9) of the manuscript. In this figure the timing of annual low flows in Switzerland is shown. The text that refers to this figure states: 'Within the Swiss Plateau, low-flow timing is more spatially consistent during some (non-extreme-drought) years (e.g. 2009, 2013, 2016), than during others (e.g. 2000, 2002, 2004, 2010, 2017)'. However, there is no reason given for this difference for each year. Is this caused by SWE, other drivers of low flows or P and PET? If it is more related to P and PET it is useful to include this in the text to further clarify the contribution of these drivers on more local scale. Also, if it is caused by SWE the results of the paper of Jenicek et al. (2016) can be related to this in non-drought years. Therefore, I suggest to get a better understanding of the variability of streamflow in

non-drought years. This can be done by looking more closely at the relation between SWE, PET and P on streamflow during these years. This can also help by putting the studied drivers (P and PET) in context to other drivers of low flow like anthropogenic activity.

**We have extended the discussion of different low-flow timings throughout the years. The differences in low-flow timing are most clearly related to when the climatic anomalies (of precipitation and PET) occur, as most of the low flows (no matter when they occur) are related to these climatic anomalies. We show (and  discuss) how these P and PET anomalies, but less so winter precipitation, relate to the timing of low flows.**

Specific comments:

In part 3.2 (line 140-142) of the manuscript the graph of figure 1 is used to explain the contribution of P and ET to low flow occurrence: 'However, again distinct regional differences exist: at low elevations, almost all annual low flows occur after periods of anomalously high potential evapotranspiration and anomalously low precipitation (Fig. 1a&b)'. However in this figure only the occurrence of low flows per month related to the elevation level of the catchment is shown. In figure 2 the differences explained in the text of 3.2 are shown and therefore this reference should be changed to figure 2a,b.

**We  realize that this can be misleading. We have changed this sentence.**

By implementation of this framework in another study area, can be stated that the PET estimation method maybe has to change depending on the climatic conditions of the new study area. It could be that they have to switch to radiation based methods (see major comment 2) depending on the local climate.

**We agree that other PET methods may be more suitable in other places. It is important to remember that we do not intend to find the best PET estimation method in this paper, but it is more important for us to use a consistent method which takes advantage of the unique gridded air temperature data that we have and that is robust for all the 380 catchments in the different geoclimatic regions of Switzerland. It is also important to remember that our analysis only depends on the relative differences in the PET estimates (from year to year and day to day, but not from site to site), and not on their absolute values.**

The description for figure 2 and 3 is quite large and maybe can be shortened by putting more explanation in the text or by making the figures clearer with a main and sub-title. Especially, the part about the

percentages of low flows that are caused by combinations of drivers (figure 3), is already mentioned in the text.

**We agree that the captions of Figures 2&3 are long. This is on purpose: experience has shown that readers who are browsing an article will find it much easier to understand if the main point of each figure is clearly stated in the caption, not just in the main text (which may appear several pages earlier or later, and thus will not be found unless the reader actually reads the whole paper from front to back). We also find that it is helpful to give detailed explanations directly in the caption, rather than expecting the reader to flip back and forth between the figure and wherever it is discussed in the main text. We will consider shortening the figure captions, by removing parts that are not essential.**

The line in figure 2 seems higher than 1200 meter (even with the non-linear y-axis). The data seems to be more in agreement with Jenicek et al. (2016) on a separation between low and high elevation catchments around 1350 meter above mean sea level. This can also be a part of the discrepancy between Jenicek et al. (2016) and this manuscript on SWE relation to low flows.

**The non-linearity of the y-axis is necessary to avoid a very uneven distribution of catchments along it. We  removed the line and do not  use that threshold anymore in the revised version on the manuscript.**

The reason for choosing the spearman correlation instead of for example the Pearson correlation is not given. This can easily be done by stating that the data is non-linear and the spearman correlation will result in a better fit with this data. See D. R. Legates & G. J. McCabe (1999).

**Thank you, we have added an explanation and the corresponding reference.**

Link Seneviratne 2012a to the IPCC report does not work in the references

**Thank you, we have corrected this.**

References:

Hargreaves, G. H. and Samani, Z. A.: Reference Crop Evapotranspiration from Temperature, Appl. Eng. Agric., 1(2), 96, doi:10.13031/2013.26773, 1985.

Jenicek, M., Seibert, J., Zappa, M., Staudinger, M. and Jonas, T.: Importance of maximum snow accumulation for summer low flows in humid catchments, Hydrol Earth Syst Sci, 20(2), 859–874, doi:10.5194/hess-20-859-2016, 2016.

Legates, D. R., & McCabe Jr, G. J. (1999). Evaluating the use of "goodness-of-fit" measures in hydrologic and hydroclimatic model validation. Water resources research, 35(1), 233-241.

Smakhtin, V. U.: Low flow hydrology: a review, J. Hydrol., 240(3–4), 147–186, doi:10.1016/S0022-1694(00)00340-1, 2001.

Trajkovic, S. (2007). Hargreaves versus Penman-Monteith under humid conditions. Journal of Irrigation and Drainage Engineering, 133(1), 38-42.

---

## Referee Report (RR1)

**A review of "Effects of climate anomalies on low flows in Switzerland" by Floriancic et al. (2nd round of reviews)**

In my opinion, authors have considerably improved the manuscript. New methods and results have been added to support the study results (e.g. the GLM model or assessment of human impacts). The clarity of the text has been improved by several text modifications and by splitting results and discussion sections. Additionally, further information has been added to the text to better explain methods and results. Nevertheless, some of the previous comments were not fully addressed and thus I would like to ask for further clarification. Although, my comments below are relatively large in extent, I think that it should not be time demanding to implement suggested changes.

**General comments**

I am still not convinced about the fact that authors used only catchments where summer low flows are also annual low flows. This way, they excluded most of high elevation catchments from the analysis. Therefore, it is not much surprising that SWE (or winter precipitation) are not important indicators for summer low flows since snow dominated catchments were (probably) not analysed. Nevertheless, I am accepting authors decision to present the results in this way.

However, the fact that only a subset of 380 study catchments was used for most of the analyses is (in my opinion) not fully clear from the methods and results sections. I think that most of readers might be confused about how exactly you proceeded. For example, in Section 3.1 one would conclude that you analysed all 380 catchments and showed the results in Fig. 1 (a-d). However, this would be not fully true since all catchments are shown only in Fig. 1a and 1b, while Fig. 1c a 1d show only those catchments for which the annual low flow occurred in summer (as I understood from your response). I think that most of readers cannot infer this important limitation from the text, despite the fact that you mentioned that Fig. 1c and 1d show May-November low flows (which is mentioned only in the Figure caption, but not in the main text). For the reader this would not be clear since two possible interpretation exists (at least to me); 1) you considered all catchments, but only warm period low flows, or, 2) you considered only those catchments where annual low flows occurred in the warm period. Without knowing your response, I would (wrongly) assume that (1) is how you proceeded. Similar notice, which might be a bit confusing is given in Fig. 2 caption ("winter low flows were excluded"). A clear statement that two different subsets of catchments were used for presented analyses is also missing in methods. I partly found it in Section 2.3 (L 161-163), but, again, I think that the formulation here is not fully clear and do not explicitly mention that this procedure caused exclusion of several snow dominated catchments from analysis.

A clear statement, how you proceeded is given only in discussion Section 4.2 (L 401-406). I would recommend to provide the reader with a clear information already in methods (and results) about the catchment reduction since it widely affects your interpretation and conclusions regarding the role of SWE and winter precipitation. Also maybe add the information how many catchments were excluded in the end. Besides, consider to reformulate the abstract as well which (wrongly) implies that your results regarding winter precipitation and SWE can be related to all selected 380 catchments across Switzerland.

Additional to the above, I think that some interpretation regarding the role of snow or winter precipitation is oversimplified. The reaction of individual catchments to climatic anomalies and thus low flows is also a matter of catchment storage, which is usually longer than one season. Therefore,

the winter conditions most likely influence the summer streamflow (and low flows), although the importance of such influence may be minor (as shown by your results for lower elevation catchments) and it certainly differs from catchment to catchment. I am aware that this goes much beyond the scope of the paper, but I would suggest reflecting the issue of catchment storage in discussion (beyond the sentence on L 370-371).

**Specific comments and technical corrections**

Authors did not consider a comment to describe (in methods section) the procedure how they analysed the role of winter precipitation (although they declared in the response that they added the description to methods section). Similarly, the newly used predictor (SWE) is not mentioned in methods (there is only the information about source of SWE data).

Regarding the comment of the Reviewer 3 on L237 (original manuscript). All specific terminology ("below-threshold" and "above-threshold" in this case), should be defined at the place, where it is firstly used. This is not the case in the revised version. Additionally, the explanation needs to be in the main text, not only in the Figure caption.

L 169: Perhaps, you wanted to rename Section 3 to "3. Results" since the discussion is newly included as Section 4.

Please use term "elevation" instead of "altitude" consistently in the paper.

Technical note: For the future, it would be great if you would be more specific in the response, specifically, to indicate where one could find the changes you made (e.g. by referring to line numbers in the response). Additionally, to submit a "tracked changes" version of the revised manuscript (as requested by HESS and which was missing here) really helps the reviewers with orientation.

---

## Author Response (AR2)

Dear Editor,

We hereby resubmit our revised manuscript entitled "Effects of climatic anomalies on low flows in Switzerland". We revised the manuscript according to the recommendations by addressing all comments of the reviewers and the editor, and we hope that our revised version is now suitable for publication.

We thank you for the detailed explanation of your decision and addressed the points of concern. Below we list all editor and reviewer comments *(in italic)* and our answers (in bold).

Thank you for considering our revised manuscript. We appreciate your work and look forward to your response.

With best regards Marius Floriancic

(on behalf of the co-authors Wouter R. Berghuijs, Tobias Jonas, James W. Kirchner, and Peter Molnar)

Dear Marius et al.,

thanks for resubmitting this revised version of the manuscript. In general the two reviewers that assessed this new version found that many of the critical aspects addressed. However, some concerns remain and one reviewer has requested further revisions that require reconsideration before a publication decision is made. Hence I would like to invite you to resubmit a revised version for reconsideration.

In summary, both reviewers still raise concerns particularly about

a) lacking clarity of the data selection and analysis of sub-samples (streamflow records used and seasonal windows analysed) and

b) about choice and terminology of the metrics analysed, questioning in particular the choices made to analyse an impact of SWE.

I re-read the Methods Section 2.3 to assess these concerns and also did not find it sufficiently clear. Unless the methodology is properly written out stating for each analysis: the data sampling, the variable used, a hypothesis for the test applied and the assumptions on the data this test requires, and for statistical models ideally also the equation, I see no possibility to accept the manuscript for publication.

For illustration perhaps, I try to explain my sources of confusion while reading section 2.3: after the at-site time series correlations are stated as a method, line 156 then suddenly introduces a "sign test for the distribution of the individual rs values". So there is a switch from at-site to a regional test? But what is the (regional/spatial?) hypothesis tested here, which test exactly is used? Then, the text jumps back to at-site correlations, then again to a regional sub-selection, which is suddenly introduced (another one?)... Then the GLM? Is this a regional model again or a number of at-site models (predict Qmin spatially or temporally)? How exactly is this 'fraction of the R2' determined - as far as I know there are many different methods to do this, again making certain assumptions. Finally, I can understand what has been done only by looking at the results and that is not sufficient. Some analysis (like the human influenced catchments) are even not at all introduced in the Methods section. All (!) sampling choices and analyses carried out need to be clearly understandable from reading the methods section alone. I think this revision is easy to implement.

We now realize that the methods were not sufficiently presented and therefore we improved the methods section. We now provide a detailed step-by-step description of all analyses and statistical procedures are more carefully described and explained. See the substantial (track) changes in Section 2.3. and Section 2.1 in the revised version of the manuscript.

Please also thoroughly consider the concerns of R2 regarding climate anomaly vs absolute low flow. Line 60 of the ms states: "The annual lowest flow is (for a particular year) an exceptional flow condition, so we expect low flows to occur after weather conditions that are atypical (for that same year)." This sentence is very hard to read, but as I understand it, the assumption is that any annual minmum flow must be caused by seasonal climate anomaly. For very stable glacial and nival regimes, the Qmin of which is almost in the same week-month, I doubt this is correct. Please check if the statement is generally applicable like this and make it more clear. We reformulated the sentence to make the sentence easier to understand. In addition, we state that (most) catchments tend to have year-to-year variations in low-flow timing (e.g., Fig 1 revised paper). Many catchments with relatively stable low-flow timings (e.g., some high alpine sites) still tend to have substantial year-to-year variability in their low-flow timing (e.g., Fig 1 revised paper). We therefore believe it is logical to expect that annual low flows occur after weather conditions that deviate from their seasonal norm. We now reformulate and extended the sentence:

"Because the annual lowest flow is an atypical flow condition, we expect it to follow atypical weather conditions, rather than reflecting climate seasonality alone. Therefore, we hypothesize that annual lowest flows will typically occur after anomalous weather conditions, that is, weather conditions that deviate from the seasonal average."

I think we don't doubt the result of the SWE impact, but it will only be credible if the assumptions are well justified. I agree with the R's concerns that the reduction to using 1 March snowpack as an indicator may be a lost chance and certainly will require better argument for this choice. The rest of the mountain-world uses April 1st snowpack for water resources planning! I strongly suggest that you add some proof (could be literature) that for Switzerland March 1st is the 'consensus' peak-of-snow-accumulation and really reflects the 'end-of-winter' storage of snowpack+groundwater well in CH.

We understand that 1st of April is in many cases used as the standard reference for (northern hemisphere) studies that consider SWE. However, in Switzerland many lower elevations locations have no substantial snow left by this date, whereas 1st of April snowpack depths are likely strongly correlated with snowpack depths a month prior. Therefore March 1st snowpack better serves the particular purpose of our study, focused on warm-season low flows, but April 1st snowpack would yield comparable conclusions. We now also state this in section 2.1 in line 144 – 146, and added references to three papers presenting data evidence for this across Switzerland (Steger et al., 2013 – Figure 3; (Lüthi et al., 2019 – Figure 4 and Winstral et al., 2019 – Figure 3).

A concern not yet sufficiently well responded to is the one-sided outlier removal. The 'three-sigma rule' intends to detect statistical outliers: on both ends of the distribution. Removing them from analysis is another step that needs argument. The underlying assumption of the statistical view is that such values are unlikely/physically impossible and must be wrong due to the measuring technique. Eliminating

subjectively the outlies in (only) one tail of the distribution because (?) they are inconvenient (?) represents a non-acceptable data selection. The lowest flows will likely also be not well-gauged! I cannot accept the simple addition of a reference to the general three sigma rule paper as a response to this concern. I also don't understand why the removal would be useful. Having higher flow values and hence more variability in the sample that is correlated with preceding climate anomaly (which will for sure be not a deficit when there is high flow - unless it's caused by snowmelt and then that is relevant!) should only be beneficial to finding a higher correlation. If for some reason high flow values are not wanted, another sampling strategy up front is necessary (like annual series vs partial duration series in flood stats - "peak over threshold", for low flows there is a reason why normally events under threshold are commonly used in low flow statistics).

To eliminate the problems of the three-sigma rule we decided to now – instead of removing the high annual low flows by the 3-sigma rule – remove all annual lowest flows that are higher than the 25th percentile of all flows, so above 2.5 mm/d. This choice is made because we still believe that setting an upper threshold is important to ensure we characterize actual low-flow conditions (similar to how flood studies use peaks over threshold in frequency analysis). We now state this in section 2.1 of the revised version.

**Additional editorial comments:**

I am afraid the abstract could better reflect the work - maybe a result of the revisions?. It has redundancy and contradictory statements in several sentences. The repeated focus on the comparison to larger scale Europe does not match title nor the real contribution that this study now makes. I suggest to revise it to bring out the contribution better and attract the right attention.

We edited the abstract to better reflect the results presented in our manuscript. We did this by rewording several sentences, by removing any reference to wider patterns across Europe, and we do not mention spatial patterns anymore. Together this should attract the attention to the right places, as all results mentioned in the abstract are at the heart of our manuscript.

Yes, I will insist in changing PET to the likewise common Ep for example if it is used as a variable name. If it's only used in text and as acronym, please do as you like. In any case, just because others have done it wrong is not a reason to do it wrong also.

**Thank you for the clarification. We now changed PET to Ep throughout all text and figures.**

Regarding figure legends and captions: HESS house style asks for "concise figure captions". There are different styles for figure captions, yes, and that can be accommodated. However, in the current version of the manuscript the same information on figures is sometimes repeated 3-4 times: in a header over the figure (really unnecessary!) and exact same text in the caption text under a) and b) for example, then in the following explanatory sentence again, and in the text of the manuscript. Please revise this for conciseness - in particular do not repeat colors if there are legends. Repetition does not make anything more clear but is instead confusing for readers.

We reviewed all captions and adjusted them accordingly:

- Fig 1: No changes made because it was already concise.
- Fig 2: This caption is shortened by removing some repetitive statements.
- Fig 3 & 5: The captions might still look long, but they describe multiple subfigures. The only part we could reduce is removing the statement on the main finding from this figure. However, we really believe that we do the reader a service by including this statement as it provides guidance and only marginally extends the caption.
- Fig 4: This caption is shortened and concise.
- Figs 6-9: We reduced the number of words used in the caption and the figures (i.e. headers).

**Anonymous Referee #2**

Review of "Effects of climate anomalies on low flows in Switzerland" by Floriancic et al.

Many of my previous comments were acknowledged and I find the focus shift towards the assessment of the shaping of the low flows worthwhile.

I went through the revised version and found still some points that the authors might consider before publishing:

**Main comments:**

• Now the shaping of low flows focuses on summer low flows, which excludes all the catchments that have winter low flows. While I find it important to explicitly distinguish between summer and winter low flow due to the processes driving, I would suggest looking also at the development and drivers of the low flows in the cold season. Here as well there might be a pattern occurring from simply lack of precipitation and the built up of snowpack or a combination of the two. This analysis could be done in a similar manner as the one for summer low flows using PET and lack of precipitation, for instance using low air temperature (or since available estimates of SWE).

We changed the title (and various other parts) of the paper to better emphasize our work is about warm-season low flows. We agree that analyses of cold season low flows can also be interesting. However, from the start the focus of this study was on warm-season low flows only, because these are the low flows that are driven by summer climate anomalies and are a source of severe river droughts in lowland catchments in recent years. Cold-season low flows may also be related to climate anomalies, but less strongly than summer low flows. Including winter low flows would therefore go beyond the original scope of the paper. However to reach out to the reviewer, we show results for the 30-day climate anomaly analysis (Figure 2c&d) also for cold-season low flows (occurring in December through April) in the supplementary materials (Figure S2). This emphasizes that climate anomalies are important for summer low flows, but maybe less so for winter low flows.

• While the authors made efforts in the revision to make the terminology used less ambiguous and more consistent, I am not convinced of every choice made here. To me anomaly for the cumulated sum of differences between actual an average value the preceding period is not a good idea. It suggests already that the variable (PET or PRECIP) was anomalous before even looking at it, at the start of the analysis only the streamflow is low, but it might not be at all. I disagree also on the definition of anomaly for low flows with the authors arguing that within the year the streamflow deviates from the "norm". In my opinion on the contrary it most of the times is not anomalous since it occurs very reliable in the same season of the year.

We improved the textual description and introduction to "climate anomalies". Using deviations from the seasonal norm comes with the advantage that can reveal how short-term weather anomalies affect low flows, rather than seasonal climatological patterns. These short-term weather anomalies obviously do not fully explain all low-flow characteristics, but clearly explain a significant part of it. We also emphasize that this definition has been used from the first time this paper was send out to review.

• The authors have with their data set the chance to show which of the preceding periods for which driver or combinations of drivers was anomalous but the study at state does not take that chance. It would require rewording and calling only anomalous conditions anomalies and then second it would make the study much stronger if there was a quantification of the drivers and clear communication of what a combination causes in terms of low flow or not.

Our paper characterizes which anomalies lead to annual low flows highlighting several aspects of low flows, and the anomalies that drive them. The provided suggestion is an interesting alternative suggestion to look at the data, but we do not see any chance of including this without changing the entirety of the paper or by extending the paper with more data methods and results, without a clearly defined connection to the current version.

• Rather than only arguing with correlation coefficients it would be very interesting to see a quantification *f* the effects. To keep it comparable between the catchments that could be expresses in percentage. But

precipitation deficit in the preceding period as well as PET compared to mean PET in the season could be compared to Qmin. This would make a stronger argument for the study.

Perhaps we do not understand what the referee is pointing to, but Figure 6 shows an objective measure of the bivariate non-parametric correlation between P and PET deviations from their means and annual Qmin. All catchments and years are put together for this analysis so that we get a regional robust signal. Figure 7 instead shows the fraction of the best R2 in a multivariate linear model obtained by each variable P and PET. We do not follow which stronger arguments can be made.

• Snow in the preceding winter was found to be only weakly related to summer low flow, this was based on the SWE of the 1st of March. I am not convinced of taking the SWE of a specific date, that might be already in the melting season for many lower elevated catchments and may not be the maximum amount for the highest elevations. I would suggest testing another metric maybe the maximum SWE in the winter period or something else that can be considered representative for the snowpack that could (or not) contribute to summer low flow for each catchment.

See comment to the editor earlier. We considered and discussed a suite of snow-related variables, including SWEmax. Finally we chose a SWE at a fixed time in spring as the best measure because this in our opinion captures the lasting snow still available for melt into the summer and feeding baseflow. SWEmax, which will occur much earlier in the winter season, may be depleted rapidly during warm periods and not contribute to summer low-flows directly.

**Minor comments**

• Most results show only % of the low flow of the catchments show correlation with anomaly xy. I suggest to add more comparison between the catchments (where is the correlation stronger, weaker? can this be attributed to a region, geology, elevation range?) for more than only the map of low flow occurrence, such material could help follow up studies and they could be for instance be placed in a supplemental material.

We now provide maps of the spatial distribution of the rank correlations between the 30-day anomalies and low flows in the supplementary material (Figure S3).

• Statistical tests and quantification of process importance: Why the original data set was reduced first to selected warm season flow? Why not first select this period and then determine Qmin for all years and all catchments in this period? Were there really some that have only 5 years available? Are these then only the drought years?

As now more clearly stated (and explained above) our study aims to report the climate drivers of warm-season low flows only. In addition, we only select events that can be considered low flows (i.e. by setting an upper threshold). Just over 15% of the sites had less than 10 years of data available after setting these criteria, which typically occurred in the drought years. The nature of our analysis is that we do not emphasize the results of individual catchments, but rather infer behaviors from patterns and behaviors that are revealed across many sites.

• The expression "annual late summer and autumn low flow period" could be simply introduced as warm season/period, as is also used already sometimes by the authors (why would May be late summer?).

We now consistently use the term "warm-season low flows" throughout the whole manuscript and fully avoid the term "summer and autumn low flow period".

• The colors of the figures are not readable in black and white print. While I am aware these are also electronically available at least the figures with blue and red lines should be distinguishable in b&w, that could be by choosing a different line type or a different color choice. Best would be if also the colors of the maps could be distinguishable then e.g. Jun and Oct would not look the same. There are palettes that have 12 colors that can be also distinguished in b&w e.g. viridis.

We changed the blue colors throughout the manuscript to make the figures readable in black and white in Figures 2, 4, 6, 7 & 9.

Detailed comments line by line

L18 In the abstract 2011 is still mentioned as summer low flow despite occurring in spring

We changed this to "years".

L21 what is meant by characteristics and where is that picked up in the study?

By characteristics we refer to duration and magnitude of the anomalies. We slightly reworded the abstract, explain it better in the introduction, and remind the reviewer that these analyses can be found in Section 3.2., 3.3., 3.4 (and figures 2, 3, 4, 5).

L35 I am not convinced that event is the right term for low flow; for an event I would expect that there is a threshold involved defining he start and end of it. The reference discussed droughts and I guess the authors intend to write also here "drought" as suggested by the follow-up sentence

We changed it to "droughts".

L37 One key reference here is Price (2011)

We included a reference to Price et al, 2011.

L41 I guess here drought is intended again (at least the references point there)

We changed it to "droughts and their effect on low flows".

L53 add Staudinger et al. (2017) for storage in Swiss catchments

**We included a reference to Staudinger et al, 2017.**

L55-62 see general comment above: I am not convinced of the terminology and would not call every low flow preceding period "climate anomaly", I would rather call the deviating periods among the preceding periods anomalies.

We refer to our earlier response above.

**L65 "likely" sounds like it could be also not true? Why would that be?**

We changed the wording to "expected to be" as there are several scenarios possible where this may not be true (although such scenarios are unlikely to happen consistently, except in for example completely human controlled flow environments).

L70 add something on that the lag time is dependent on the storage behavior that is different for each catchment; limited-substantially they express the opposite please clarify

In the revised version, the sentence before makes the connection with storage.

L73 But there is also snow involved in these changes which could cause a less seasonal than we are used to effect anyhow?

We now added "In addition, anticipated changes in snowfall and snow packs may also alter river flows (CH2018, 2018)."

L75-76 please revise and be more precise: Temperature is not depleting soil moisture storage. PET is also influenced by wind and not solely by temperature. T is a good indicator for PET, but AET will be very much dependent on how much a storage (soil) is wet or dry. E.g. with high PET but zero water in the soil, the soil will not be depleted and AET accordingly will be low, same PET on a wet soil will cause the depletion described and AET is high. We have revised this to be more precise. The passage now says, "High temperatures can be an indicator of high Ep, and thus high potential for depletion of soil moisture storage, reducing aquifer recharge and streamflow (e.g., Jaeger & Seneviratne, 2011; Vidal et al., 2010). Temperature extremes can be amplified when low soil moisture limits evapotranspiration, leading to lower relative humidity and higher air temperatures, which further increase Ep (Granger, 1989)."

**L103 which are the few studies? Please add.**

To our knowledge there are no studies that look at low flows the way we do in our manuscript. We changed it to "To our knowledge there are no studies..."

L194 What is "tend" intended to express? The majority? And after which climate anomalies this is not the case? Please, clarify or reformulate.

We reformulated this sentence.

L196 remove "tend"

We removed it.

L197 affect -> affects

"affect" relates to "...magnitudes of both precipitation and PET anomalies..."

L203 at the same time? or in the same period in series or in parallel -> rephrase

We changed it to "in the same time period".

L213 remove Particularly

We removed "particularly".

L245 "below-threshold precipitation ... above-threshold PET" which threshold is that? From the methods I got that it is about the entire cumulated variable (difference actual value and average) preceding the low flow period. Please, clarify

We now explicitly explain this in the methods section (line 174-177).

L243-248 I would argue that it is not only because of the duration but when the "anomaly" starts, because together they form somehow an intensity of the anomaly. The authors argue that duration of high PET is stronger correlated than duration of low precipitation. This might be because of the seasonal character rather than the duration implicitly considered. This is for precipitation much lower than for PET. And this might change again whether the anomaly starts say in May and lasts for a month or whether it starts in September and lasts for a month.

We mention that different dimensions of climate anomalies (i.e., magnitudes, timings, and durations) are important in low-flow formation in the discussion section in lines 481 – 486. Indeed the reviewer is right, the "intensity" of the anomaly is precisely a combined effect of the departures from mean and the duration. This is exactly what we try to capture. It is also true that seasonality plays a role in this intensity, and this is especially true for Ep due to temperature seasonality.

L249/250 If a single precipitation event can exceed the "threshold" maybe the definition of anomaly is illposed?

This specific analysis is using above threshold P and Ep rather than climate anomalies. We updated the text accordingly to make this distinction clearer.

L286/287 exhibit clear growth -> clearly increase

We changed this.

L317 "explains most of the predictability" should that be variability? Please clarify

This was a typo: explains most of the variability in Qmin correct.

L362 What does "relatively comparable" mean?

We changed it to "similar".

L412 Virtually?

We changed it to "Almost".

**Figures**

Generally, all figures have long captions including partly interpretation that is already in the text. Please, remove the redundant parts and only leave the necessary elements in the captions. E.g. Figure 1 drop the last sentence, that is literally repeated in the main text.

We removed the last part of Figure caption 1 and shortened the other Figure captions as well. See comments to the editor above.

Figure6 in black and white the lines cannot be distinguished either use different line type or colors with different hue. Some methodological questions: 1) are the anomalies in fact cumulated sums over the preceding period for PET and precipitation differences, respectively 2) if these are anomalies shown, does each point consider a different number of data pairs 3) are here all catchments included, i.e. also the high elevation catchments?

We changed the colors throughout the manuscript (i.e., in Figs. 2, 4, 6, 7, 9 of the revised manuscript) to make the lines (and bars) distinguishable in black and white. Anomalies are cumulated departures of P and PET from the mean for periods preceding the annual low flow. Non-parametric correlations (markers) shown in Fig 6 are computed from a combined dataset of all stations and years together. Each marker has exactly the same number of station-years in the correlation. We did not analyze station-wise correlations because for most sites the record would not be long enough. With the combined record and using variables that are depths over the catchment area (mm) we believe we obtain a robust regional estimate of the strength and direction of the relationship between P/PET departures and Qmin. High-elevation catchments are included in our dataset only if they have warm-season annual low flows.

Figure 7 b) it would be good to see next to an average how different that is for the dry years, please add the lines for each single year

The data in Figure 7b are not an average, it is the fraction of the R2 obtained by a multivariate GLM model by P and PET for all catchments in the 4 dry years. The data are insufficient to get robust estimates of the bivariate P or PET correlations for each year individually, so this fraction cannot be computed for every drought year.

In my opinion, authors have considerably improved the manuscript. New methods and results have been added to support the study results (e.g. the GLM model or assessment of human impacts). The clarity of the text has been improved by several text modifications and by splitting results and discussion sections. Additionally, further information has been added to the text to better explain methods and results. Nevertheless, some of the previous comments were not fully addressed and thus I would like to ask for further clarification. Although, my comments below are relatively large in extent, I think that it should not be time demanding to implement suggested changes.

**Thank you for acknowledging the changes we made in the revised manuscript.**

**General comments**

I am still not convinced about the fact that authors used only catchments where summer low flows are also annual low flows. This way, they excluded most of high elevation catchments from the analysis. Therefore, it is not much surprising that SWE (or winter precipitation) are not important indicators for summer low flows since snow dominated catchments were (probably) not analysed. Nevertheless, I am accepting authors decision to present the results in this way.

**Thank you for accepting our choice to focus on warm-season low flows.**

However, the fact that only a subset of 380 study catchments was used for most of the analyses is (in my opinion) not fully clear from the methods and results sections. I think that most of readers might be confused about how exactly you proceeded. For example, in Section 3.1 one would conclude that you analysed all 380 catchments and showed the results in Fig. 1 (a-d). However, this would be not fully true

since all catchments are shown only in Fig. 1a and 1b, while Fig. 1c a 1d show only those catchments for which the annual low flow occurred in summer (as I understood from your response). I think that most of readers cannot infer this important limitation from the text, despite the fact that you mentioned that Fig. 1c and 1d show May-November low flows (which is mentioned only in the Figure caption, but not in the main text). For the reader this would not be clear since two possible interpretation exists (at least to me); 1) you considered all catchments, but only warm period low flows, or, 2) you considered only those catchments where annual low flows occurred in the warm period. Without knowing your response, I would (wrongly) assume that (1) is how you proceeded. Similar notice, which might be a bit confusing is given in Fig. 2 caption ("winter low flows were excluded"). A clear statement that two different subsets of catchments were used for presented analyses is also missing in methods. I partly found it in Section 2.3 (L 161-163), but, again, I think that the formulation here is not fully clear and do not explicitly mention that this procedure caused exclusion of several snow dominated catchments from analysis.

A clear statement, how you proceeded is given only in discussion Section 4.2 (L 401-406). I would recommend to provide the reader with a clear information already in methods (and results) about the catchment reduction since it widely affects your interpretation and conclusions regarding the role of SWE and winter precipitation. Also maybe add the information how many catchments were excluded in the end. Besides, consider to reformulate the abstract as well which (wrongly) implies that your results regarding winter precipitation and SWE can be related to all selected 380 catchments across Switzerland.

We now fully revised the methodology section to provide a step-by-step explanation of when and where we used subsets of the data (see chapter 2.3. – lines 164 - 204). We also better emphasize that the main motivation of the manuscript is to better explore warm-season low flows, by adapting the title and also the wording throughout the revised version of the manuscript. We'd like to point out, that the main focus of the paper was on warm-season low flows in lowland catchments that are sensitive to climate anomalies and have the greatest adverse economic impacts (agriculture, shipping, etc.).

Additional to the above, I think that some interpretation regarding the role of snow or winter precipitation is oversimplified. The reaction of individual catchments to climatic anomalies and thus low flows is also a matter of catchment storage, which is usually longer than one season. Therefore, the winter conditions most likely influence the summer streamflow (and low flows), although the importance of such influence may be minor (as shown by your results for lower elevation catchments) and it certainly differs from catchment to catchment. I am aware that this goes much beyond the scope of the paper, but I would suggest reflecting the issue of catchment storage in discussion (beyond the sentence on L 370-371).

As correctly observed, by looking at warm-season low flows only, we exclude most of the high-elevation catchments. In the lower elevation catchments, snow only represents a small fraction of total annual precipitation, thus snow has almost no importance for (long term) storage in low elevation catchments, where the annual lowest flows occur in the warm season.

**Specific comments and technical corrections**

Authors did not consider a comment to describe (in methods section) the procedure how they analysed the role of winter precipitation (although they declared in the response that they added the description to methods section). Similarly, the newly used predictor (SWE) is not mentioned in methods (there is only the information about source of SWE data).

We now updated the methods section accordingly and describe how we calculate the winter precipitation sums and also SWE. See section 2.3 at lines 191-195.

Regarding the comment of the Reviewer 3 on L237 (original manuscript). All specific terminology ("belowthreshold" and "above-threshold" in this case), should be defined at the place, where it is firstly used. This is not the case in the revised version. Additionally, the explanation needs to be in the main text, not only in the Figure caption.

**We now included this in the methods section 2.3 at lines 174-177.**

L 169: Perhaps, you wanted to rename Section 3 to "3. Results" since the discussion is newly included as Section 4.

Thank you, we changed this in the revised version.

*Please use term "elevation" instead of "altitude" consistently in the paper.*

We now consistently use "elevation" throughout the manuscript.

Technical note: For the future, it would be great if you would be more specific in the response, specifically, to indicate where one could find the changes you made (e.g. by referring to line numbers in the response). Additionally, to submit a "tracked changes" version of the revised manuscript (as requested by HESS and which was missing here) really helps the reviewers with orientation.

We apologize for this inconvenience and now (try to) provide more specific response including section number and line numbers, and also provide a track-changes manuscript.

**Effects of climate anomalies on warm-season low flows in Switzerland**

Marius G. Floriancic1,2, Wouter R. Berghuijs2, Tobias Jonas3, James W. Kirchner2,4, Peter Molnar1

1 Institute of Environmental Engineering, ETH Zurich, 8093 Zürich, Switzerland

- 2 Department of Environmental Systems Science, ETH Zurich, 8092 Zürich, Switzerland
- 3 WSL Institute for Snow and Avalanche Research SLF, 7260 Davos Dorf, Switzerland

4 Swiss Federal Research Institute WSL, 8903 Birmensdorf, Switzerland

Corresponding author: Marius G. Floriancic (floriancic@ifu.baug.ethz.ch)

10

5

Keywords: low flow, hydrological drought, precipitation, evapotranspiration

Short summary: Low river flows affect societies and ecosystems. Here we study how precipitation and potential evapotranspiration shape annual warm-season low flows across a network of 380 Swiss catchments. Low flows in these rivers
typically result from below-average precipitation and above-average potential evapotranspiration. The lowest low flows result from long periods of the combined effects of both drivers.

Abstract. Large parts of Europe haveSwitzerland has faced extended periods of low river flows in recent summersyears (2003, 2011, 2015, and 2018), with major economic and environmental consequences. Understanding the origins of events like these

- 20 is important for water resources management. While precipitationIn this work we provide data illustrating the individual and potential evapotranspiration obviously impact summer low flows, it remains largely unquantified which characteristicsjoint contributions of precipitation and potential evapotranspiration are related to low flow magnitude, flows in both typical and how these relationships may vary regionally.dry years. To revealquantify how weather drives low flows, we explore how deviations from mean seasonal climate conditions (i.e., climate anomalies) of precipitation and potential evapotranspiration
- 25 shapedcorrelate with the occurrence and magnitude of the-annual 7-day lowest flows (Qmin) during the warm season (May through November) across 380 Swiss catchments from 2000 through 2018. Most annualwarm-season low flows followed periods of below-average precipitation and above-average potential evapotranspiration, and the lowest low flows resulted from both of these drivers acting together. In the driest years, low flow conditions occurred simultaneously across large parts of Europe, but lowLow-flow timing during these years-was still-spatially variable across Switzerland. Low flows in theall years
- 30 , including the driest (2003, 2011, 2015, and 2018-). Low flows in these driest years were associated with much longer-lasting climate anomalies compared tothan the maximum two≤2-month anomalies which causedpreceded typical warm-season low flows in other years. Across Switzerland, weWe found that precipitation totals in winter and snow water equivalent and winter precipitation totals only slightly influenced the magnitude and timing of summer and autumnwarm-season low flows.-in low-elevation catchments across Switzerland. Our results provide insight into how precipitation and potential
- 35 evapotranspiration jointly shape warm-season low flows across Switzerland, and potentially aid in assessing low-flow risks in similar mountain regions using seasonal weather forecasts.

**1. Introduction**

In recent decades, Europe has experienced several severe low flow eventsdroughts (Van Lanen et al., 2016). Their impacts, such as dry river reaches and high water temperatures, have a range of adverse effects on society and river ecology (e.g.,
Poff et al., 1997; Bradford & Heinonen, 2008; Rolls et al., 2012Price et al., 2011; Rolls et al., 2012; van Vliet et al., 2012). Severe low flows in the years 2003, 2011, 2015 and 2018 led to substantial economic losses by limiting water availability for households, industry, irrigation and hydropower, as well as impacting river transportation (Stahl et al., 2016; Munich Re, 2019). Such effects are expected to become more severe and frequent as water demand rises, and as droughts are anticipated to increase in frequency and intensity in the future (e.g., De Stefano et al., 2012; Wada et al., 2013), leading to calls for
improved understanding and management of droughts and their effects on low flows across Europe (e.g., Seneviratne et al., 2012a; Van Lanen et al., 2016; WMO, 2008).

In temperate climates, annual low flows typically occur in two distinct seasons, i.e., in. during late summer and autumn in warmer regions and during winter in colder regions-during winter (Fiala et al., 2010; Smakhtin, 2001). This typical low-flow seasonality has been reported for many regions of the world, including, for example, Austria (Laaha & Blöschl, 2006; Van Loon & Laaha, 2015), the Rhine river basin (Demirel et al., 2013; Tongal et al., 2013), and North America (Cooper et al., 2018; Dierauer et al., 2018; Wang, 2019). Switzerland also has two-distinct low-flow seasons, where the distinction between warm-season low flows and winter low flows is strongly connected to elevation (Wehren et al., 2010; Weingartner & Aschwanden, 1992). Low flows in low elevation Swiss catchments tend to occur in late summer and early autumn (August through October), whereas) in highlow-elevation Swiss catchments most low flows occur, and during the winter (January

through March).) in high-elevation catchments.

Catchment properties shape low flows by controlling the storage and release of water ((e.g., Stoelzle et al., 2014; Van Lanen et al., 2013; Van Loon & Laaha, 2015<del>), but the landscape itself does not cause low flows.; Staudinger et al., 2017),</del>

60 but the landscape itself does not cause low flows. Instead, the drivers of low flows are meteorological conditions that dry out catchments (e.g., Fleig et al., 2006; Haslinger et al., 2014 ; Smakhtin, 2001). Two distinct low flow seasons exist throughout Switzerland (and many other regions), suggesting that different weather conditions drive low flows during these two seasons: warmWarm-season low flows are typically caused by sustained periods of high evapotranspiration and low precipitation, whereas winter low flows often follow sustained periods of sub-freezing temperatures (e.g.; Laaha et al., 2013); Van Loon,

- 65 2015). Thus, low flows are not created instantaneously, but result from The duration of these anomalous weather conditions acting over longer periods. The is critical in shaping the annual lowest flow is (for a particular year) an exceptional flow condition, so we expect low flows to occur after flows. Their timing varies between years and is largely driven by climate seasonality. In this paper we refer to weather conditions that are atypical (for that same year). From now on, we refer to atypical weather conditionsdeviate from the seasonal norm as 'climate anomalies'- regardless of the magnitude of this departure.
- 70

The two main climatic factors controlling water storage and release in a catchment are precipitation and temperature (through controllingits influence on snow processes and evapotranspiration). It is therefore likely that Therefore precipitation (P) and potential evapotranspiration ( $PETE_p$ ) anomalies are expected to be important drivers of warm-season low flows across Switzerland. For example, precipitation Precipitation controls the amount of water that is made-available to for runoff in a

[revised manuscript text omitted]